# Control points for design of taxonomic composition in synthetic human gut communities

## Graphical abstract

## Authors

Bryce M. Connors, Jaron Thompson, Sarah Ertmer, Ryan L. Clark, Brian F. Pfleger, Ophelia S. Venturelli

## Correspondence

venturelli@wisc.edu

## In brief

Microbial communities have tremendous potential for applications ranging from healthcare to agriculture, but systematic approaches for controlling these complex systems are lacking. Connors et al. develop a two-stage model-guided approach to precisely manipulate community composition as a function of media components and initial species abundances.

## Highlights

- Media and inoculation conditions are control points for community composition

- To design community temporal behaviors, we used a dynamic computational model

- Monoculture information expedites community design goals

- Statistical design of experiments and models navigate high-dimensional design space

Connors et al., 2023, Cell Systems *14*, 1044–1058
December 20, 2023 © 2023 The Authors. Published by Elsevier Inc.

## Cell Systems

**CellPress**

# Control points for design of taxonomic composition in synthetic human gut communities

Bryce M. Connors,[1,2] Jaron Thompson,[1,2] Sarah Ertmer,[1,2] Ryan L. Clark,[1] Brian F. Pfleger,[2] and Ophelia S. Venturelli[1,2,3,4,*]

[1]Department of Biochemistry, University of Wisconsin-Madison, Madison, WI 53706, USA
[2]Department of Chemical & Biological Engineering, University of Wisconsin-Madison, Madison, WI 53706, USA
[3]Department of Bacteriology, University of Wisconsin-Madison, Madison, WI 53706, USA
[4]Lead contact
*Correspondence: venturelli@wisc.edu

## SUMMARY

Microbial communities offer vast potential across numerous sectors but remain challenging to systematically control. We develop a two-stage approach to guide the taxonomic composition of synthetic microbiomes by precisely manipulating media components and initial species abundances. By combining high-throughput experiments and computational modeling, we demonstrate the ability to predict and design the diversity of a 10-member synthetic human gut community. We reveal that critical environmental factors governing monoculture growth can be leveraged to steer microbial communities to desired states. Furthermore, systematically varied initial abundances drive variation in community assembly and enable inference of pairwise inter-species interactions via a dynamic ecological model. These interactions are overall consistent with conditioned media experiments, demonstrating that specific perturbations to a high-richness community can provide rich information for building dynamic ecological models. This model is subsequently used to design low-richness communities that display low or high temporal taxonomic variability over an extended period. A record of this paper's transparent peer review process is included in the supplemental information.

## INTRODUCTION

Microbiome engineering holds tremendous potential for applications spanning human health to environmental remediation.[1] A key challenge toward realizing this potential is developing the capability to steer communities toward desired states.[2] Microbial communities are complex and dynamic systems, shaped by nonlinear interactions and feedback.[3,4] Control of such nonlinear dynamical systems toward desired states is a fundamental question that lies at the heart of many problems encountered in the field of engineering and is critical to harnessing the functional capabilities of microbiomes for a wide range of potential applications.[5] Precise control of microbiomes requires the ability to elucidate influential control parameters and predict temporal behavior as a function of these different inputs.[6]

The ability to control microbial community dynamics could facilitate the development of defined combinations of human-associated intestinal isolates as next-generation bacterial therapeutics.[7,8] The beneficial properties of well-characterized mixtures of commensal strains could be optimized while simultaneously avoiding the drawbacks of fecal microbiota transplantation.[9,10] A key challenge toward this goal is the economical production of defined, therapeutic communities that span the phylogenetic and functional diversities of the healthy adult microbiome.[11] The current strain culturing process contributes substantially to this

challenge, as constituent organisms of a community are typically grown as separate monocultures, and then subsequently mixed to a desired species composition.[12] This "monoculture-then-mix" process is complicated, costly, and scales poorly for communities with large numbers of organisms.[12] Implications of this economic barrier extend beyond commercial-scale profit margins: medical progress is slowed when pilot-scale drug supply bottlenecks clinical trials or global health applications become cost prohibitive.[11–13] Culturing communities in a single-vessel could circumvent high costs associated with the monoculture-then-mix approach but presents new challenges stemming from the fundamental question of community control.

Leveraging model-guided approaches to predict community growth as a function of specific control inputs would greatly enhance our ability to manipulate community composition toward a desired state.[14] The Monod equation, and its ecological counterpart MacArthur's consumer-resource model, is, in theory, well suited to describe the growth and metabolite dynamics likely to govern community assembly.[15,16] However, it is not trivial to identify and quantify many unknown metabolites driving constituent community member growth in an experimental system.[17] Even if all key resources were known and measurable, modeling these metabolites in addition to microbial species may lead to an intractable state space for parameter inference from experimental data in larger communities.[18] Finally, the consumer-resource model

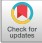

structure for a given system may not easily generalize to other environmental conditions or communities.

Optimal design of experiments (DoE) is an approach that leverages informative experiments and statistical modeling to map key input-output relationships and has been increasingly used to engineer biological systems.[19] For example, DoE has been used to explore regulatory sequence space for modulating protein translation and tuning enzyme expression to optimize production of a target metabolite.[19–21] In addition, DoE was used to formulate a chemically defined media by optimizing microbial growth as a function of various media components.[22,23] Statistical modeling, an integral part of the DoE workflow, has been used to predict specific community-level functions from species abundance and community composition as a function of dietary inputs in the murine gut.[24–26] Dynamic ecological models, such as the generalized Lotka-Volterra (gLV) model, have been shown to be predictive of microbial community assembly in a particular media environment.[18,27] These studies suggest that statistical and ecological models can be integrated with experimental data to predict and design biological system behaviors.

We investigate and exploit control points for microbial community assembly by combining high-throughput experiments and computational modeling. We develop a two-stage, model-guided approach for systematically tuning key media components and initial species densities to maximize the diversity of a 10-member human gut community in batch culture. Based on the hypothesis that monoculture growth kinetics contribute substantially to community assembly, we use a model-guided approach to design a culture medium to promote similarity in the growth responses of single species. This high-throughput, monoculture-based optimization procedure yields a concomitant improvement in community diversity in the designed medium. We then use a design-test-learn (DTL) cycle to systematically modulate individual species' initial population sizes (i.e., inoculation densities) to further optimize community diversity in the new medium. Finally, we build a dynamic ecological model informed by our DoE-generated dataset that captures pairwise inter-species interactions. We demonstrate that pairwise interactions can be inferred from complex community data and are largely consistent with spent media experiments. This model is used to guide the design of communities with distinct classes of dynamic behaviors. Our generalizable framework provides a foundation for the data-driven control of defined microbial communities toward target compositional states.

## RESULTS

### Manipulating media components to enhance community Shannon diversity

Diverse and defined communities have immense therapeutic potential. An even strain composition is a frequent target for defined bacterial therapeutics, evidenced by the use of this formulation in a recent clinical trial.[28] Therefore, as our target composition, we aimed to maximize the Shannon diversity of a synthetic human gut community.[9,10,28] Shannon diversity is an ecological metric used to quantify the evenness and number of species in a community (STAR Methods; Equation 1).[29] We designed a representative 10-member community that spans the phylogenetic and metabolic diversities of the healthy human gut microbiome

(Figure 1A).[31] This community consisted of *Blautia hydrogenotrophica* (BH), *Bifidobacterium longum* (BL), *Bacteroides uniformis* (BU), *Collinsella aerofaciens* (CA), *Dorea longicatena* (DL), *Eggerthella lenta* (EL), *Eubacterium rectale* (ER), *Faecalibacterium prausnitzii* (FP), *Prevotella copri* (PC), and *Parabacteroides johnsonii* (PJ) (Table S1). Several of these species, including FP, have been demonstrated as critical to the recovery of a healthy microbiome after childhood malnutrition and thus hold promise as bacterial therapeutics for global health applications.[32,33]

The metabolic niches of individual species, which dictate their ability to grow in a given environment, are major determinants of community assembly.[34,35] We characterized the growth of individual species (monocultures) using time series measurements of optical density at 600 nm ($OD_{600}$) in a baseline-defined medium that can support the growth of diverse human gut species (STAR Methods; Table S2).[24] The monocultures displayed a wide range of growth rates and population sizes at steady state (i.e., carrying capacities) (Figure S1A, medium 7), suggesting that species with low monoculture fitness may be outcompeted in the community. Previous studies have shown that monoculture growth is a key determinant of community interactions and assembly.[36] Therefore, we hypothesized that promoting similarity in the growth responses of single species (i.e., balancing monoculture growth) would enhance the community Shannon diversity. To this end, we used a high-throughput workflow to design a medium that supported the balanced growth of all species in monoculture.

We exploited concentrations of key media components as metabolic control points to manipulate species growth responses.[37,38] Sugars and amino acids represented the main fermentable substrates.[39] In addition, pH is a major environmental factor and can differentially modify bacterial growth.[40] Yeast extract (Y.E.) is a complex digest containing vitamins, peptides, and other resources and can enhance the growth of FP.[17] Since the design space of media components is very large, we used statistical DoE to identify an optimal concentration profile of these components by manipulating the following four key variables: (1) a mixture of three sugars, (2) a defined mixture of amino acids, (3) Y.E., and (4) pH (Figure 1B; STAR Methods). The DoE workflow involves (1) the identification of (independent) variables and (dependent) outputs of the system, (2) the construction of an experimental design matrix of combinations of levels of each variable that satisfies an optimality criterion, (3) experimental implementation, (4) statistical modeling of the experimental data, and (5) the use of optimization techniques to determine the values of the variables that are predicted to yield a desired system output. We used this workflow to maximize the similarity between the carrying capacities of the monocultures as a function of media components while also supporting sufficient growth.

We performed time series $OD_{600}$ measurements for each monoculture in each media condition and fit a logistic growth model to these data (Figures S1A–S1C). The carrying capacity parameter ($K_i$) of this model indicates the population size at steady state (Figure 1C). To quantify the similarity among the growth responses of individual species as a function of media components, we determined the Shannon diversity of the normalized carrying capacities in a particular medium. Normalization was performed by dividing by the sum of the inferred carrying capacities in a particular medium, mirroring the calculation of

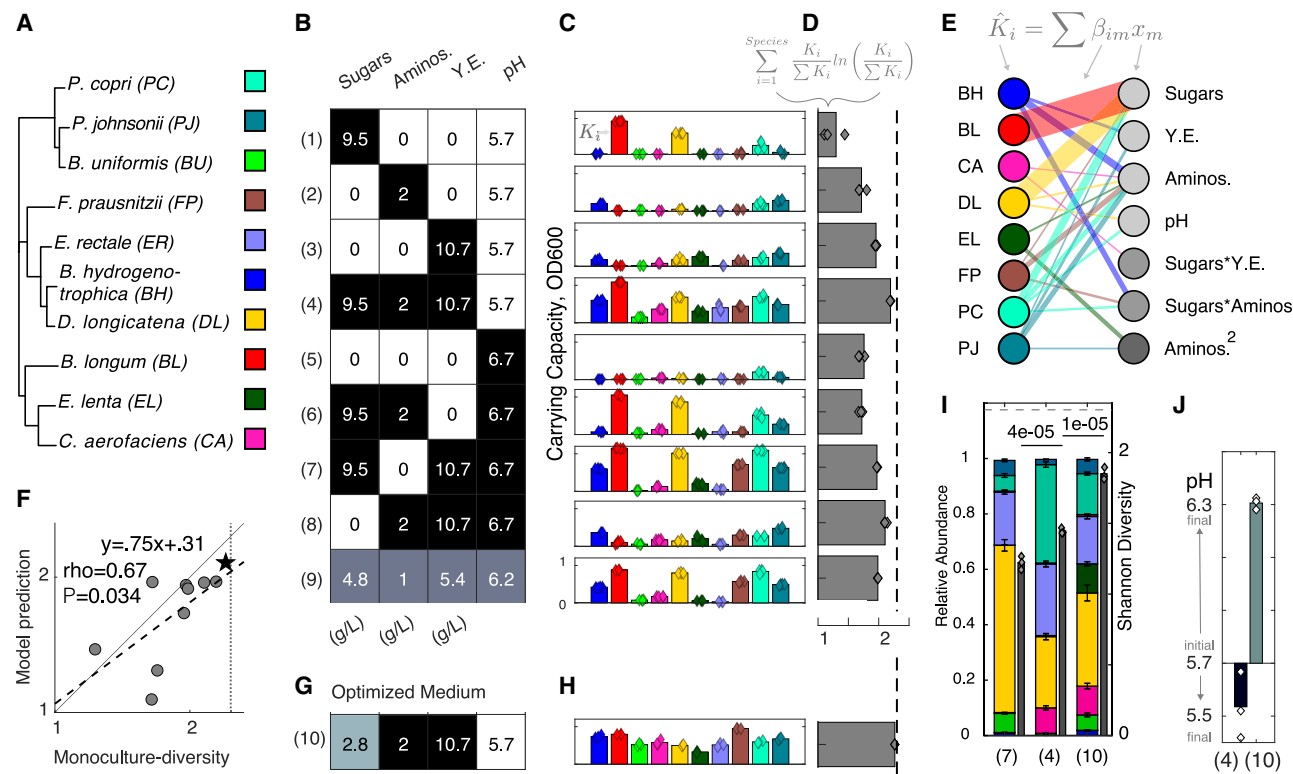

**Figure 1. Model-guided design of media composition to enhance community Shannon diversity**

(A) Phylogenetic tree of the 10-member synthetic human gut community: Bacteroidetes (upper branch), Firmicutes (middle branch), and Actinobacteria (lower branch). Phylogenetic analysis was performed using a concatenated alignment of 37 single-copy marker genes in Phylosift (Darling et al.[30]). Strain information is provided in Table S1.

(B) Heatmap of fractional factorial experimental design matrix that varies the concentration of a sugar mixture (glucose, arabinose, and maltose), yeast extract (Y.E.), defined amino acid mixture (Aminos.), and pH in a common base medium (STAR Methods). Shading indicates design levels: "high" (black), "intermediate" (gray), and "low" (white), with concentration values labeled in units of g/L or pH.

(C) Bar plots of the steady-state abundance (carrying capacity, $K_i$) of each species determined by fitting a logistic differential equation model to the time-series measurements of absorbance at 600 nm ($OD_{600}$) in each media condition (STAR Methods; Equation 3; Figure S1A). Bar height denotes the mean carrying capacity and data points denote biological replicates (n = 4 with outlier detection; STAR Methods).

(D) Bar plot of monoculture diversity (STAR Methods; Equation 4 and 5) based on the mean carrying capacities for each medium (STAR Methods). Mirroring the calculation of Shannon diversity from community absolute abundance data, monoculture diversity is calculated from the set of normalized carrying capacities in a particular medium. Normalization is performed by dividing each species carrying capacity (endpoint growth) in that medium by the sum of carrying capacities for all species in that medium, analogous to the calculation of relative abundance from absolute abundance. Data points denote monoculture-diversities calculated from each biological replicate. Dashed line indicates maximum possible monoculture diversity for ten species.

(E) Bipartite network representation of linear regression growth models, where edge thickness is scaled by mean parameter value across cross-validated parameter sets. Models predict the carrying capacity ($\hat{K}_i$) of the species $i$ as a function of media component concentrations ($x_m$) according to the experimental design (STAR Methods; Equation 6). Model parameters ($\beta_{im}$) represent the inferred effect of variable media component $m$ on the growth of species $i$. Left and right nodes denote species and media components, respectively. Light gray nodes denote linear terms, medium gray nodes denote interaction terms, and dark gray nodes denote quadratic terms. Parameters with mean values of less than 0.05 are not shown, and all parameters are shown in Figures S2A–S2J.

(F) Scatter plot of monoculture diversity calculated from logistic model fitted carrying capacities (x axis) vs. predicted monoculture diversity from media regression model on held-out media conditions (y axis; STAR Methods). Pearson correlation (rho) and p value (p) are indicated for regression line (dashed), whereas solid line indicates x = y. Star denotes the optimized medium.

(G) Heatmap of media component concentrations that maximized monoculture diversity (STAR Methods; Equation 7).

(H) Bar plot of the inferred carrying capacities based on the logistic model of each species on the optimized medium (Figure S1B). Bar height denotes the mean carrying capacity and data points denote biological replicates (n = 3).

(I) Stacked bar plot of community compositions from an even inoculum in the baseline medium (7), the highest monoculture diversity screened medium (4), and the optimized medium (10). Bar height indicates mean of three biological replicates, error bars indicate 1 SD from the mean. Shannon diversity of mean community composition (n = 3 biological replicates; STAR Methods; Equation 1) is indicated as gray solid bars (right bars). Shannon diversities as calculated from each set of biological replicates are overlaid as diamonds. p values from one-way ANOVA with multiple groupwise comparisons (MATLAB's "anova1" and "multcompare" functions) are indicated above respective Shannon diversities. Biological replicates are plotted individually in Figure S6A.

(J) Bar plot of the change in media pH for community cultures in the best screened (4) and optimized medium (10). Bar height indicates mean of biological replicates (diamonds, n = 3).

Shannon diversity from community absolute abundance data (STAR Methods; Equation 4). This quantity, hereafter referred to as "monoculture diversity," varied widely as a function of media composition (Figure 1D).

Although we identified a medium that enabled high monoculture diversity in the screening experiment (Figure 1D, medium 4), we turned to model-guided optimization for further improvement. We fit linear regression models with quadratic and interaction terms to predict the carrying capacity of each species from the concentrations of the media component variables (Figure 1E; STAR Methods; Equation 6). The media regression model parameters provide an interpretable, quantitative relationship between the concentration of a given media component and the growth of a given organism (Figures S2A–S2J). For instance, the linear term corresponding to sugars was large in the BL and DL models, consistent with a substantial growth improvement in the presence of the sugar mixture (Figures 1C, S2B, and S2E). Interaction terms in the regression models indicate where the growth effect of one media variable is dependent on another and thus are particularly useful for characterizing the complex environmental dependencies of fastidious anaerobes.[37,41] For example, BH has a linear term for amino acids and an interaction term for sugar and amino acids but lacks a linear term for sugar (Figure S2A). From the perspective of community controllability, the model parameters indicate that sugars are a useful control point for BH only if amino acids are also available. Overall, the regression parameters reveal a diverse network of growth effects, wherein sugars are the strongest, amino acids are most frequent, and interaction effects are relevant (Figure 1E). This inferred media component network predicting species growth suggests that combinations of media components could be manipulated to produce desired community compositions.

To reduce the overfitting of model parameters and avoid biasing of hyperparameters, we implemented elastic net regularization with nested leave-one-out cross-validation (STAR Methods). The goodness of fit was high for all species, whereas validation predictions on the data outside the training set (out-of-fold) ranged in accuracy (Figures S2K–S2M). Despite the sparse sampling of the design space using a half-factorial experimental design, the models were predictive of withheld monoculture diversity data (Pearson rho = 0.67, p = 0.034), although they were variably predictive of the carrying capacities of the constituent species (Figures 1F and S2N).

Using media factors as a control point, an optimization procedure (STAR Methods; Equation 7) identified a profile of media factors that maximized the predicted monoculture diversity (STAR Methods). The predicted media factor concentrations were similar to medium 4 but contained 3-fold less sugar (Figure 1G). To test this prediction, individual species were grown in the optimized medium. The monoculture diversity for the optimized medium was close to the maximum possible value, consistent with the model prediction (Figure 1H).

To determine if optimizing monoculture diversity yielded an increase in Shannon diversity of the community, we cultured the 10-member community from even initial species proportions in the baseline medium 7, best-screened medium 4, and optimized medium (Figures 1B and 1G). The model-guided, monoculture-based optimization process yielded a concomitant improvement in community Shannon diversity (Figure 1I). The community

cultured in the optimized medium 10 displayed a higher endpoint pH than the acidified environment of medium 4 (Figure 1J). These results suggest that the reduced sugar concentration in the optimized medium, compared with the otherwise identical medium 4, mitigated the production of high levels of inhibitory organic acids by fast-growing sugar fermenters. Although most high-density microbial bioprocesses require sophisticated pH control systems (i.e., bioreactors), a community culture that maintains non-inhibitory pH levels autonomously could be produced in simple vessels by precisely manipulating media components.

Our model-guided, high-throughput, monoculture-based approach identified a single medium in which all species were capable of similar endpoint growth. Compared with the baseline medium, Shannon diversity of the 10-member community was increased from 53% to 80% of its maximum possible value. These results demonstrate that a moderate number of key media variables were effective control points for manipulating the composition of a complex microbial community.

### Building a constrained dynamic growth model to narrow the inoculum design space

The initial population densities of the constituent members of a microbial community have been shown to impact community assembly.[18,42,43] Therefore, we used a DoE approach to systematically tune species inoculation densities to further improve the endpoint Shannon diversity of the community culture. However, searching for an optimal set of inoculation densities involves traversing a ten-dimensional design space wherein each inoculation value can vary over many orders of magnitude. To address this challenge, we derived a dynamic population model that incorporated high-throughput monoculture kinetic data and a total growth constraint to predict a region of the inoculum design space that was expected to yield high-diversity endpoint compositions (Figure 2A).

The total growth of communities comprising combinations of 25 human gut isolates was shown to be a saturating function of the number of species in each community, suggesting an upper limit on community growth in the medium.[24] In agreement, our 10-member community displayed substantially lower total growth than the sum of the independent monocultures (Figure 2B). This implies that negative inter-species interactions, such as resource competition, dominated the ecological network of the community. We reasoned that enforcing an upper limit on total community growth (independent of species composition) could serve as a useful null hypothesis governing community assembly, given unknown, but largely negative, inter-species interactions. We further reasoned that when subjected to this growth constraint, species with higher monoculture growth rates and larger carrying capacities would tend to outcompete lower-fitness species.

We derived a system of ordinary differential equations (ODE) in which species monoculture logistic models are coupled via an empirical total growth limit (STAR Methods). This model referred to as a constrained system of logistic equations (CSLE) reaches equilibrium for any community composition in which species abundances sum to the total growth limit ($\sum x_j(t) = K_{comm}$). The steady-state abundance of a species in the CSLE model is a continuous function of initial conditions (provided $\sum K_j > K_{comm}$). This enables the use of optimization techniques to find a set of

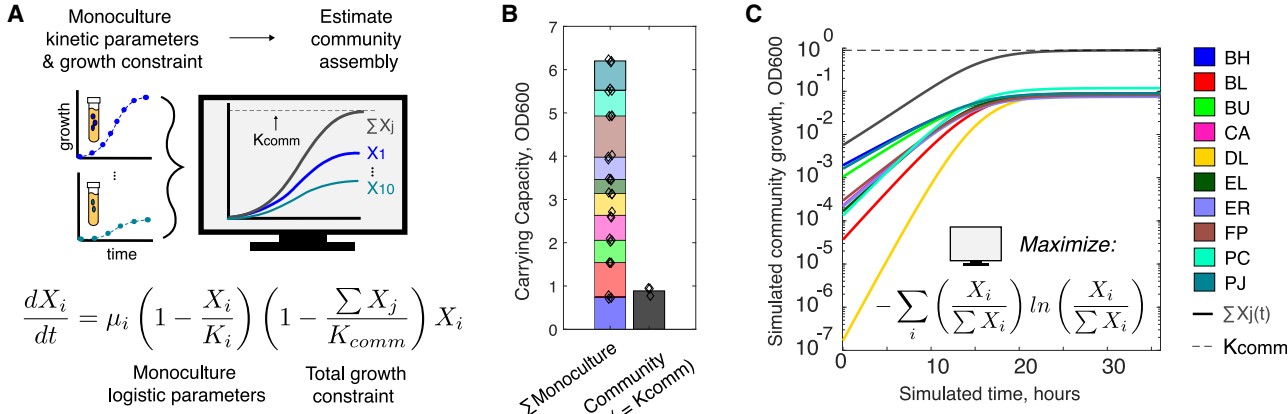

**Figure 2. Exploiting high-throughput monoculture kinetic data and a total growth constraint to narrow inoculum design space**
(A) Schematic of experimental approach and model equation to predict community assembly as a set of monoculture logistic models coupled under a total growth limit, referred to as a "constrained system of logistic equations" or "CSLE" (STAR Methods; Equation 9). Parameters: monoculture logistic growth rates ($\mu_i$), monoculture carrying capacities ($K_i$), and total community growth limit ($K_{comm}$, community carrying capacity).
(B) Bar plot of the stationary phase growth of a 10-member community culture vs. the sum of the inferred carrying capacities of logistic growth models for the 10 monocultures (bar height indicates mean, diamonds show biological replicates, n = 3, y axis units are $OD_{600}$). In this case, the $K_{comm}$ parameter value is taken as the mean endpoint $OD_{600}$ of the 10-member community culture (n = 3 biological replicates).
(C) Line plot of CSLE simulation of community assembly. Optimization techniques are used to maximize the predicted steady-state Shannon diversity as a function of initial conditions (STAR Methods; Equation 10). This model was then used to guide the experimental design for the first community inoculum experiment (Figure 3A).

initial population densities that maximize steady-state Shannon diversity using monoculture data (Figure 2C; STAR Methods). This set of initial conditions was used as a starting point for designing the first community inoculation density experiment (Figure 3A).

**Optimizing community diversity as a function of inoculation density using a DTL cycle**
We used a model-guided design-test-learn (DTL) cycle to tune inoculation densities toward achieving a target endpoint species composition (Figure 3A). The iterative DTL approach uses regression models, trained on community composition data, to guide the design of experimental conditions for subsequent independent cycles.[24] The design step consists of generating a multi-level, multi-factor definitive screening design.[44] The test step uses automated liquid handling to array inocula according to the experimental design (STAR Methods). The learn step infers interpretable model parameters from experimental data and evaluates the model's predictive capabilities.

Guided by the CSLE model, we selected a range of inoculation densities for each species to efficiently explore the design space around the best-predicted inoculum (Figure 3B; STAR Methods). The designed inocula communities were cultured anaerobically for 28 h (approximately early stationary phase). The resulting community diversity and composition varied widely as a function of the designed inocula (Figures 3C and 3D, DTL 1), confirming that inoculation density was a viable control point for manipulating community assembly. Despite a modest monoculture growth rate and carrying capacity (Figure S1C), ER overgrew in many conditions (Figure 3D, light purple), suggesting that it benefited from positive inter-species interactions that were not captured in the CSLE model.

Regression models with linear, quadratic, and interaction terms were trained to predict the absolute abundance of each species in the community from the inoculum values (STAR Methods). After the first DTL cycle (DTL 1), the inoculum regression models accurately predicted half of the species relative abundances (Pearson rho > 0.7, p < 1e−6; Figures S3A–S3C). However, three species (CA, EL, and PC) with predictive models displayed low overall growth (average relative abundance less than 2.5% of the community across the design conditions; Figure 3D). As such, these models may not be able to accurately extrapolate beyond these ranges to achieve a high Shannon diversity.

Throughout the DTL cycle, we used prior information to choose the set of inoculation density values predicted to yield the highest diversity community and used these values as the "center point" condition for the new design. The center point of an experimental design represents a unique condition in which all variables are set halfway between their high and low levels. The high and low levels were then assigned to explore the design space around the best-predicted condition or center point (STAR Methods). In DTL 2, the new center point inoculation density value for species that were poorly predicted or displayed low overall growth was determined based on a qualitative interpretation of the community data. If a species tended to overgrow (e.g., ER) in the previous cycle, the new center point value was set at the previous cycle's low value. Conversely, if a species tended to undergrow (BL, CA, DL, EL, PC, and PJ), the new center point value was set to the previous high value (Figures S3D and S3E; STAR Methods). For species that were accurately predicted by the models and displayed substantial growth, we performed model-guided optimization of the inoculation density values, targeting a community composition that maximized the predicted Shannon diversity (STAR Methods).

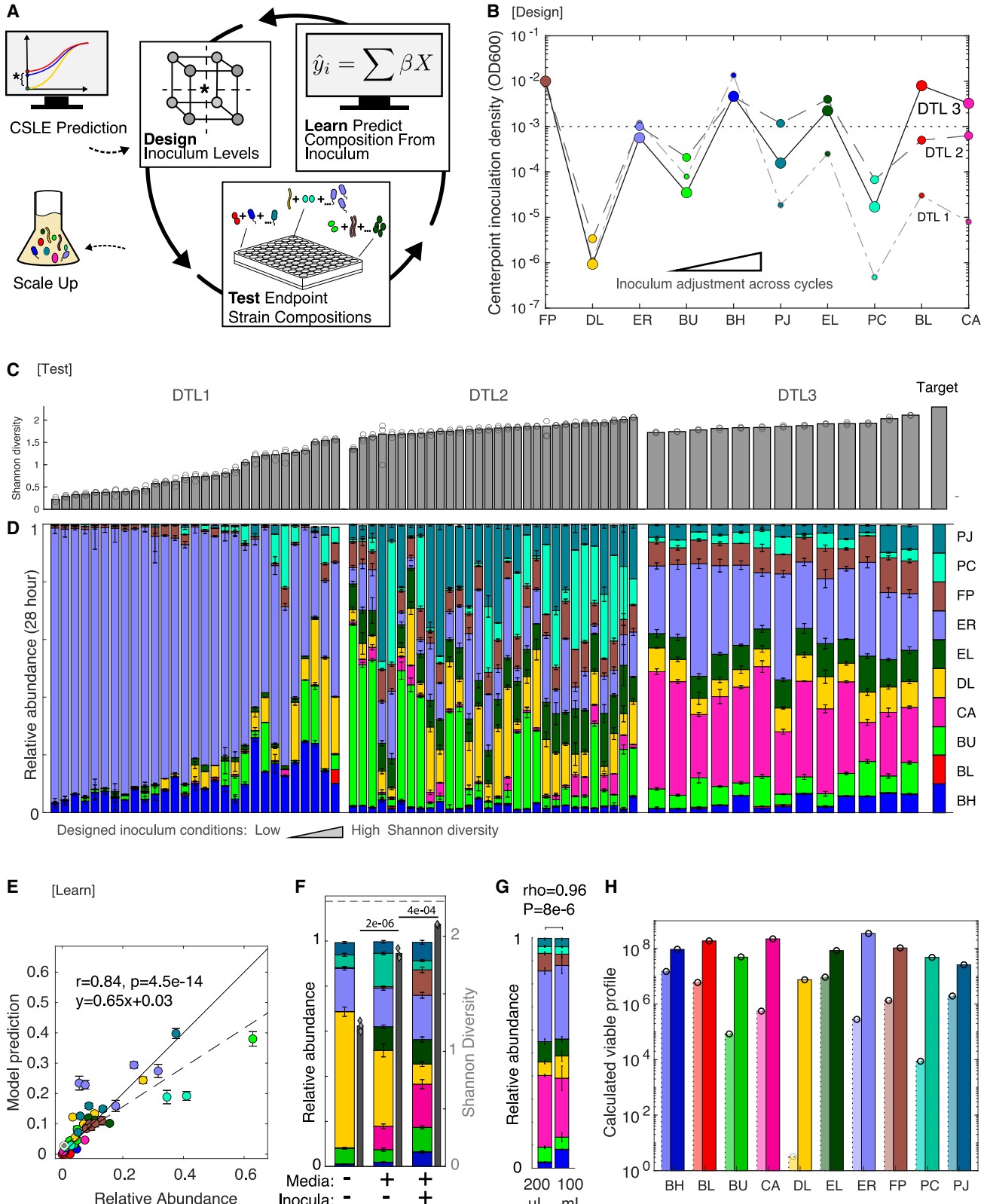

**Figure 3. Tuning species inoculation densities to optimize community Shannon diversity using a design-test-learn cycle**

(A) Schematic illustrating the design-test-learn (DTL) cycle for maximizing community diversity as a function of species inoculation density. The center point of each experimental design corresponds to the inoculum predicted to yield the highest community Shannon diversity. DTL 1 center point is predicted with the CSLE

*(legend continued on next page)*

The median community diversity was substantially higher in DTL 2 than in DTL 1, indicating that data obtained in DTL 1 were informative for enhancing Shannon diversity (Figure S3F). In addition, ER was present at a lower abundance in the community than in DTL 1 (Figure 3D). For further improvement, inoculum regression models were trained on community composition data from both DTL 1 and 2. The models accurately predicted the absolute abundance of all species except BL during cross-validation (Pearson rho > 0.70, p < 1e−8; Figures S3B and S3C). Furthermore, the models accurately predicted the abundance of each species in test communities that were entirely withheld from the training and cross-validation processes (Figure 3E; Pearson rho = 0.84, p = 4.5e−14).

To validate the accuracy of the improved models, we repeated the optimization procedure with the nine predictive regression models for all species excluding BL to determine the center point for DTL 3 (STAR Methods). The high and low levels probed a smaller design space than previous cycles, reflecting higher confidence based on the substantial improvement in model performance. Despite having the largest inoculum and high monoculture fitness, BL remained low in abundance in DTL 3, suggesting that it was negatively affected by other members of the community in these conditions (Figure 3D). Notably, the health-beneficial species FP was higher abundance in DTL 3 communities than in the even inoculum baseline condition (Figure 3F).[45–47] Overall, the highest Shannon diversity condition was identified in DTL 3, representing 91% of the maximum possible value for a 10-member community (Figure 3F). This was a substantial improvement from the already high 80% maximum diversity achieved by medium optimization alone (Figure 3F). Since diversity metrics differentially weight species evenness and could potentially yield different results, we compared Shannon with Inverse Simpson diversity and confirmed similar trends (Figures S3F–S3H). Our overall workflow enhanced the Inverse Simpson diversity index from 23% to 75% of the maximum for the 10-member community.

The CSLE model informed by monoculture data displayed an informative relationship with species relative abundance for 7 of the 10 species, demonstrating that monoculture growth can drive substantial variance in community assembly (Figure S3I). Notably, a significant correlation (Pearson rho = 0.66; p = 0.039) was observed between the log-transformed best inoculation densities predicted by the CSLE model to inform DTL 1 and the identified inoculation densities that yielded the highest community diversity in DTL 3 (Figure S3J). These results indicate the CSLE model effectively leverages high-throughput monoculture data and a total growth constraint to narrow the inoculum design space for achieving a high-diversity community composition.

Biomanufacturing of microbial communities in a real-world setting requires (1) robustness of endpoint community composition to technical variability in species inocula, (2) translation to production-scale volumes, and (3) viability of organisms harvested at the endpoint. Despite the 4-fold variation in inoculation density levels in DTL 3, the coefficient of variation (CV) of the endpoint Shannon diversity across design conditions was less than 6% (Figure 3C). This demonstrates that our process was robust to moderate variations in species inocula, whereas changing inoculum densities over orders of magnitude was a viable control point. The community relative abundances in 200 μL and 100 mL batch cultures were highly correlated (Pearson rho = 0.96, p = 8e−6), demonstrating that a 500-fold difference in batch culture scale did not substantially alter community assembly (Figure 3G).

A statistically robust estimate of the number of colony-forming units (CFUs) per mL for each species would require a large number of 16S rRNA gene-sequenced colony picks. In addition, selective plating of each species in the community is not feasible. Therefore, we used different approaches to provide information about the viability of species in the endpoint community cultures. To this end, we transferred a small aliquot (25-fold volume/volume dilution) of the communities measured at the

model, thereby exploiting high-throughput monoculture kinetic data to narrow the inoculum design space. Subsequent DTL cycle center points are predicted according to inoculum regression models, which are trained on community data collected during the DTL process.

(B) Categorial scatter plot of center point inoculum conditions for each DTL cycle, which are informed by model predictions when possible (STAR Methods). Species are sorted by the magnitude of the difference between the log transformed inoculation densities of the first and last DTL cycles: inoculation densities of species to the left were better predicted by the CSLE model. The dotted line indicates the even proportion inoculum (baseline condition). Increasing marker size corresponds to DTL cycle number as indicated in the righthand side of the panel. Full design matrices are shown in Table S3.

(C) Bar plots of Shannon diversity of mean community compositions shown in (D). Axis ticks are 0.5 units, target condition (right) is the maximum for a 10-member community (Shannon diversity equals 2.3). Gray circles indicate Shannon diversities calculated from sets of n = 3 biological replicates.

(D) Stacked bar plots of endpoint community compositions (28 h, approximately stationary phase) for each DTL cycle. Stacked bars and error bars represent mean and 1 SD. from the mean of biological replicates (n = 3), respectively, for each condition. The target endpoint community composition (even species proportions) is indicated on the far right of the panel. Biological replicates are plotted individually in Figures S6B–S6D.

(E) Scatter plot of the experimentally measured relative abundance of each species versus inoculum linear regression model prediction. The model was trained on community composition measurements from the first two DTL cycles and used to predict five held-out inoculum conditions from DTL 1 and DTL 2. Pearson correlation (rho) and p value (p) are indicated for regression line (dashed), whereas solid line indicates x = y. The cross validation (out-of-fold) predictions with species-specific correlation coefficients are shown in Figures S3A and S3B.

(F) Stacked bar plot of community compositions from media optimization (Figure 1) and inoculum optimization (this figure). Species composition (stacked bars, left axis) and Shannon diversities as calculated from mean of species abundances (gray solid bar); diamonds show diversities calculated from individual sets of biological replicates (right axis). Even inoculum and baseline (pre-optimization) are indicated with (−), whereas (+) indicates that the community resulted from media and/or inoculum optimization in this study. p values from one-way ANOVA with multiple groupwise comparisons (MATLAB's anova1 and multcompare functions) are indicated above respective Shannon diversities. Biological replicates are plotted individually in Figure S6E.

(G) Stacked bar plot of species relative abundance of the 10-member community cultured from DTL 3 center point condition in a 200 μL microtiter plate versus a 100 mL flask. Bar height and error bars represent mean and 1 SD of n = 3 biological replicates. ρ and p indicate Pearson correlation and p value, respectively, between the relative abundances in the two communities. Biological replicates are plotted individually in Figure S6F.

(H) Bar plot of calculated viable profile (calculated CFU/mL) for inoculation (light bars with dotted outline) and endpoint for community cultured from best inoculum condition at 100 mL scale, n = 1 biological replicate indicated by gray circle (STAR Methods).

CellPress

endpoint into fresh media and grew them to an approximately stationary phase (Figure S4A; STAR Methods). All species in all conditions yielded greater than 3-fold increase in abundance during the second passage, suggesting that these species had viable populations in the DTL 3 endpoint measurements (Figures S4B and S4C). Measurements of $OD_{600}$ quantify total biomass and not the number of cells, which may be important for certain applications. To further investigate the viability of the community, we performed CFU plating of the best inoculation condition DTL 3 condition 6 (Figure S4D). We calculated the viable profile of species in the community as the product of relative abundance and total CFU (Figure 3H). This quantity (Figure 3H, solid outline) was higher than the viable inoculation density (dotted outline), suggesting that the population size of each species had increased by approximately an order of magnitude (STAR Methods). In addition, colonies from the community CFU plates were pooled, and all species were detected except PC using 16S rRNA gene sequencing (Figure S4E). Notably, BL was well represented in the community culture inoculated with the best inoculum condition (Figure 3H). This result was consistent with the dominance of BL observed in later passages of DTL 1 and coincided with a fresh lot of BL freezer stocks that perhaps yielded precultures better primed for community growth (Figures S4F–S4H). In sum, these results demonstrate that our two-stage culturing design process yielded a diverse community of viable organisms, even though we did not train the model on species viability data.

## A dynamic computational model of community assembly from designed inoculation conditions

The gLV model (STAR Methods; Equation 13) is a set of coupled ODE that describe the growth dynamics of each species in a community as a function of basal growth rate and pairwise interactions with constituent community members. This model has accurately predicted complex synthetic community dynamics, and its interpretable parameters have revealed inter-species interactions.[18,24,27] These studies largely employ combinatorics-based approaches using subsets of the community to infer model parameters from experimental data (e.g., combinations of species presence/absence at even inoculation proportions). We hypothesized that the gLV model could also capture trends in community assembly in our DTL dataset wherein all species were initially present but varied in inoculation densities. Since the optimal experimental design structures used in the DTL cycle aim to constrain parameters for a linear model, we hypothesized that these data could also be informative for parameter inference using an ODE model.[44]

We trained the gLV model on monoculture time series growth and community stationary phase measurements (including three additional passaging time points of DTL 1 and one additional passaging time point of DTL 3) to characterize the communities over longer timescales (Figures 3D, S4C, and S4F–S4H; STAR Methods).[13,18,48] Parameters were inferred from absolute abundance estimates taken as the product of relative abundance and total $OD_{600}$. The model predictions were subsequently normalized to relative abundance prior to statistical evaluation to eliminate potential error propagation in the absolute abundance calculation (STAR Methods). To reduce the overfitting of model parameters to noise in the training data, we implemented L1 reg-

ularization and used cross-validation to find the optimal regularization coefficient value (STAR Methods).

Algorithms to identify the best parameter estimate seek to balance a trade-off in parameter bias due to regularization and sensitivity of the parameters to noise in the training data (i.e., variance in parameters to small perturbations in the training set data).[49] A model whose parameters display high variance to small perturbations in the training data (under-regularization) or biased by over-regularization will both display poor predictive capability on new data outside the training set. Therefore, the best parameter estimate would map to a regularization coefficient value that maximizes model prediction performance on held-out data. Using the full dataset for cross-validation, the model's average predictive capability on held-out partitions (Pearson rho > 0.85) occurred for a wide range of regularization coefficient values, suggesting that it was not prone to overfitting (Figure S4I). In contrast, the model's predictive capability was sensitive to the regularization coefficient value and displayed a clear peak in model prediction performance when fit to a randomly sampled subset (25%) of the full dataset. This indicates that the model fit to a smaller dataset was prone to overfitting in the absence of a sufficient regularization penalty (Figure S4J). These results demonstrate the full dataset derived from DoE-driven inoculation densities was sufficiently informative to constrain a gLV model whose predictive capability was not sensitive to a wide range of regularization coefficient values.

The gLV model trained on a randomly sampled 90% of the dataset was highly predictive of test data consisting of the remaining 10% (Pearson rho = 0.92, p = 8.3e−88; Figure 4B). Analyzing correlations on a per-species basis revealed that all except FP were accurately predicted (Pearson rho > 0.7, p < 1e−4 vs. rho = 0.46, p = 0.04; Figure 4B). The inoculation density of FP was fixed at a high value for all conditions in the DTL cycle. In contrast, inoculation density conditions for the other nine species varied widely according to the experimental designs, suggesting that structured sampling of the inoculation density design space via DoE was informative for the model.

We next explored whether interspecies-interaction parameters inferred from 10-member data (Figure 4C) were consistent with recipient species monoculture growth in the presence of each donor species supernatant. To this end, we calculated the "supernatant growth effect" by subtracting the total growth (i.e., area under the monoculture growth curve) of the recipient species cultured in the donor species supernatant from that of the recipient species cultured in fresh media. We evaluated whether the supernatant effects were consistent with the sign of the inferred pairwise gLV inter-species interaction coefficients. (STAR Methods). To allow visualization on similar scales, we normalized the supernatant effects and gLV parameters by dividing by the respective largest magnitude value (Figure 4D). We determined the sign of normalized interaction parameters and supernatant effects as positive, negative, or neutral (zero plus or minus a small threshold corresponding to 5% of the largest parameter value or supernatant effect, respectively) (STAR Methods). Overall, 69% of the inter-species interaction parameters displayed consistency in sign with the supernatant growth effect, with negative interactions being more frequent and thus more frequently consistent (Figure 4D; STAR Methods). In addition, a highly significant Spearman

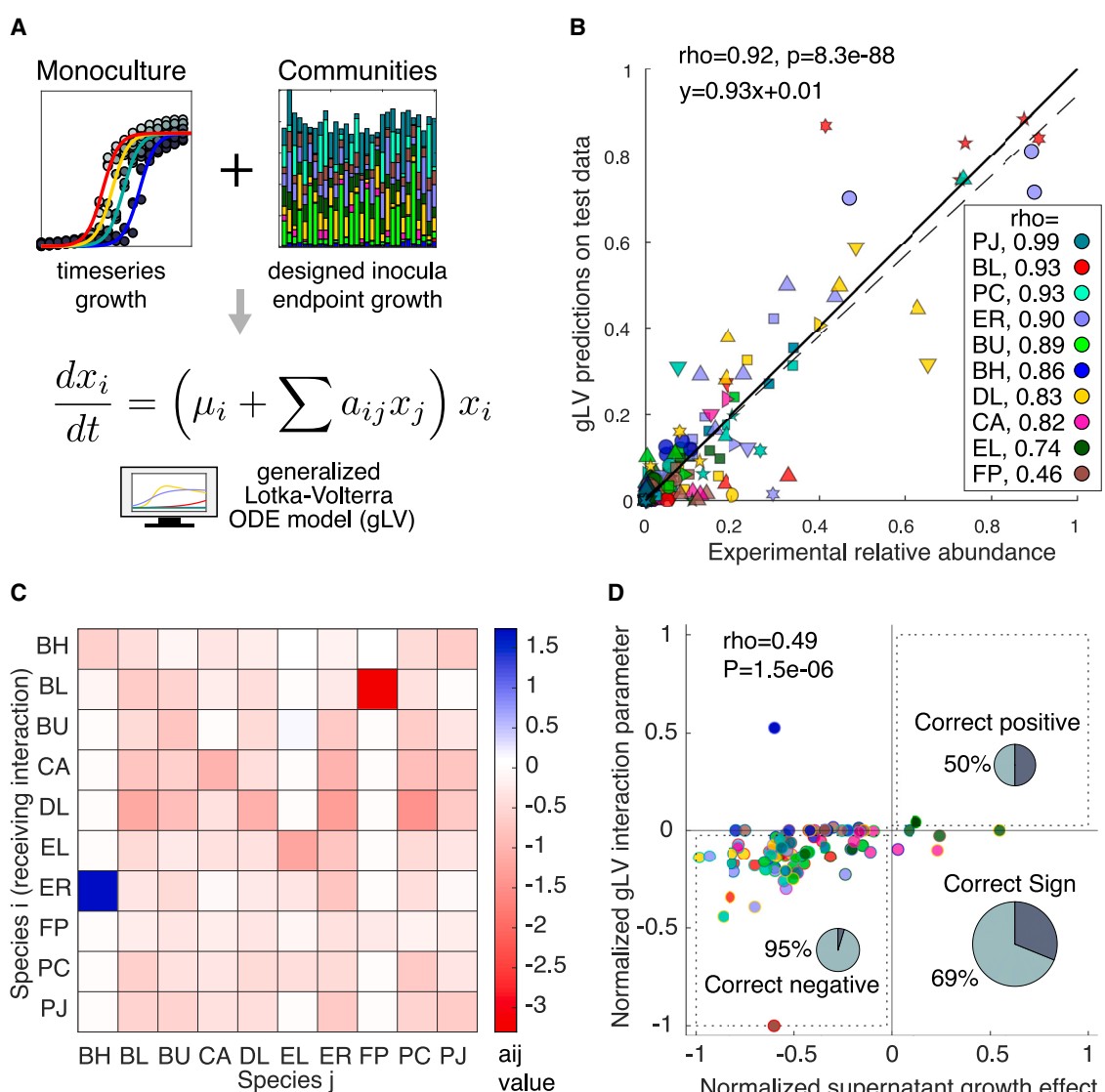

**Figure 4. Dynamic computational model of community assembly from designed inoculation conditions**

(A) Generalized Lotka-Volterra (gLV) ordinary differential equation model is trained on monoculture time series and 10-member communities from designed inoculation conditions (STAR Methods).

(B) Scatter plot of gLV predictions vs. measured relative abundance with Pearson correlation (rho) and p value (p) for all test data indicated above the plot. Correlations for individual species data are shown in the legend. "Test" signifies that data are randomly sampled and withheld from parameter inference and hyperparameter tuning (STAR Methods). Marker x-position corresponds to mean of three biological replicates and marker shape indicates experiment: circle, DTL 1; square, DTL 2; triangle, passage 2 of DTL 1; downward triangle, passage 3 of DTL 1; diamond, DTL 3; 5-sided star, passage 4 of DTL 1; 6-sided star, passage 5 of DTL 1; and right-pointing triangle, passage 2 of DTL 3.

(C) Bar plot of median gLV interaction parameter values ($a_{ij}$ matrix) for 5-fold cross validation approach with full dataset. Color scale indicates value of interaction parameter ($a_{ij}$).

(D) Scatter plot of gLV inter-species interaction parameters (C) vs. supernatant growth effect, normalized by respective largest magnitude values (STAR Methods). Supernatant growth effect is calculated by subtracting the area under the monoculture growth curve of recipient-species-in-donor-supernatant from recipient-species-in-fresh-media. Marker edge color corresponds to recipient species (i.e., receiving interaction or cultured in supernatant). Face color indicates donor species (from which interaction is received or in whose supernatant the recipient is cultured). Self-interaction parameters and effects are omitted, "rho" indicates Spearman rank order correlation coefficient, and "p" indicates p value. Normalized values are categorized as positive ($x > 0.05$), negative ($x < -0.05$), and zero ($-0.05 < x < 0.05$) with percentage of gLV parameters consistent with supernatant growth effect sign indicated as the light region of the inlayed pie chart.

correlation (Spearman rho = 0.49; p = 1.5e−6) was observed between the normalized supernatant growth effects and the inferred gLV inter-species interaction parameters. This implies that the rank order of inferred gLV inter-species interaction parameters displayed an informative relationship with the pairwise supernatant effects (Figure 4D). Differences between the

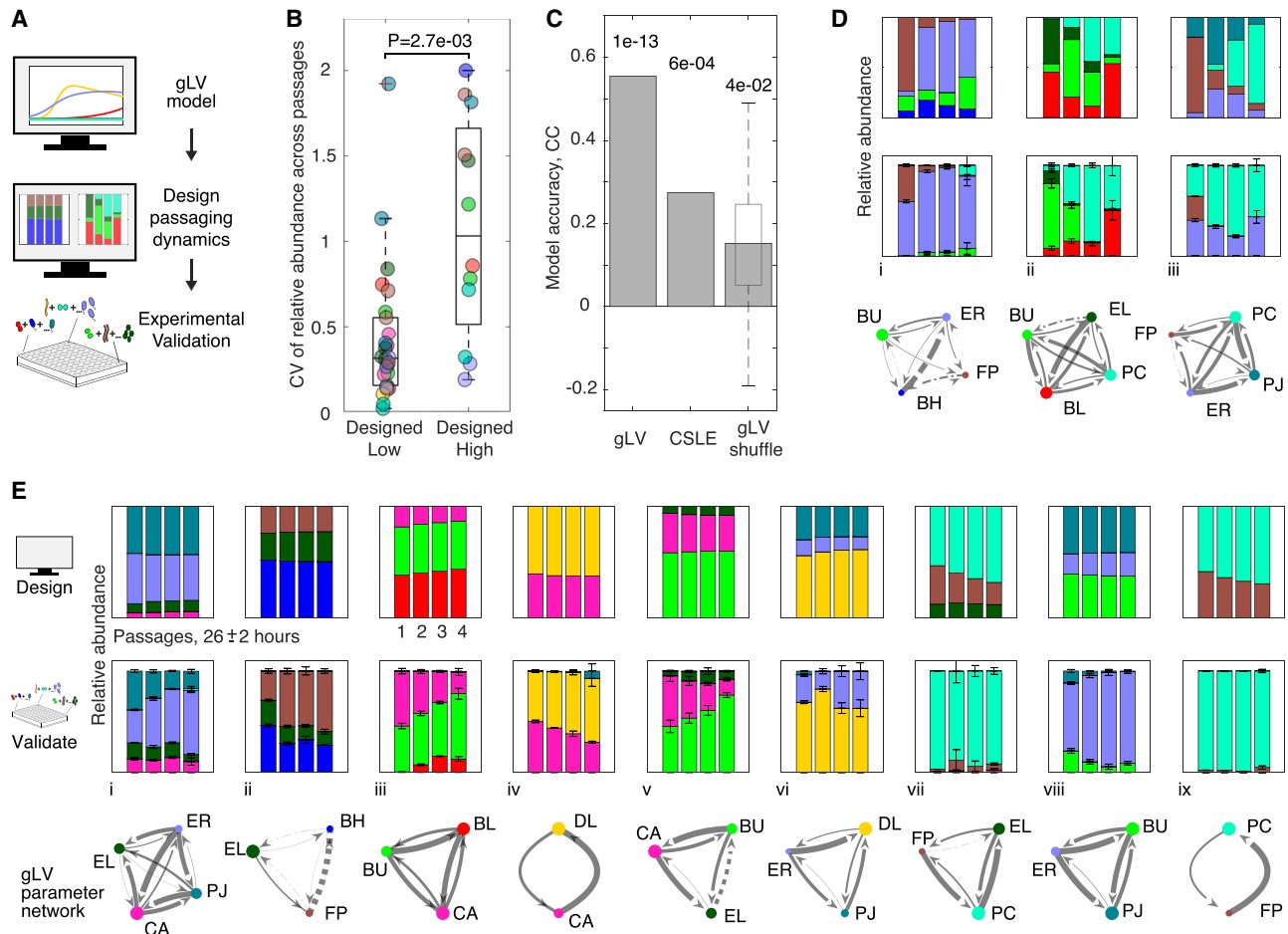

**Figure 5. Model-guided design of communities that display high or low temporal variability in community composition**

(A) A generalized Lotka-Volterra model with interactions inferred from 10-member community data, described in the previous section, guides design of subset communities (2–4 members) that display low and high variability of species composition over four simulated passages. Experimental validation is performed on a set of communities in which each species is present in at least twice.

(B) Categorial scatter plot of the coefficients of variation of relative abundance across passages for species in designed communities (STAR Methods). A low coefficient of variation across passages would indicate low temporal variability of species composition. Box plot central line indicates median, upper and lower edges denote 75th and 25th percentiles, respectively, whiskers denote range of non-outlier datapoints, and "+" denotes outlier. The p value from a statistically significant Mann-Whitney U test is shown.

(C) Bar plot of Pearson correlation (CC) between model predictions and experimental relative abundances of designed communities for gLV and two null models with p values shown above bars. CSLE refers to our competition null model described in Figure 2 and "gLV shuffle" refers to an ensemble null model wherein the original gLV inter-species interaction parameters are randomly assigned to new positions in the interaction matrix (STAR Methods). The bar height indicates the median of the distribution and box plot elements are as described in (B).

(D and E) Stacked bar plots of gLV model predictions of stationary phase community composition (top row), experimental measurements (middle row), and inferred gLV inter-species interaction networks (bottom row) for a set of (D) high and (E) low temporal variability communities. Low temporal variability communities are sorted by Shannon diversity of final passage (experimental). Species color legend follows node labels. Each subplot denotes the relative species abundance at stationary phase of the four passages; for experimental data bar height and error bars denote mean and 1 SD of biological replicates (n = 3 with outlier detection; STAR Methods). Solid and dashed edges indicate negative and positive gLV inter-species interaction parameters, respectively. Edge width is proportional to the magnitude of the inter-species interaction parameter and node size is proportional to the specific growth rate parameters in the gLV model. Biological replicates are plotted individually in Figures S6G and S6H.

inferred inter-species interactions and monoculture growth in another species supernatant could be attributed to higher-order interactions present in the 10-member community environment or insufficient information to infer inter-species interactions using the training dataset. In sum, inferred gLV inter-species interaction coefficients from inocula-designed 10-member community measurements are largely consistent with the pairwise spent media growth effects.

## Model-guided design of microbial community dynamics

Maintaining species coexistence over time can be influential in achieving a target community function for industrial and therapeutic applications.[50–52] We leveraged our gLV model to design communities composed of subsets of the full community that display low and high compositional variabilities over the timescale of four passages (Figure 5A). Low temporal variability communities were identified in our gLV model by tuning simulated

inoculation densities of all 2- to 9-member communities to maximize an objective function that favored diverse communities with constant composition across passages (STAR Methods; Equation 15). We selected a set of nine of the highest diversity communities that represented all 10 species for experimental validation. To determine if the model could distinguish between low and high temporal variability communities, we included three representative communities with predicted high temporal variability (STAR Methods).

We computed the CV of the relative abundance of each species over time to evaluate the degree of temporal variability. Consistent with the *in silico* design objective, species had significantly lower CV across passages in communities designed for low temporal variability (Mann-Whitney U test, p = 2.7e−3; Figure 5B). The gLV model trained on 10-member community data was moderately predictive of relative abundances in the 2–4 member designed communities (Figure 5C; Pearson rho = 0.55, p = 1e−13). Furthermore, the gLV model far outperformed both the CSLE model and null gLV models, where the inter-species interaction parameters were randomly shuffled. These results indicate that the inferred inter-species interaction parameters based on measurements of the 10-member community enable the prediction and design of lower species richness community dynamics (Figure 5C).

The gLV predictions captured several qualitative characteristics of the high temporal variability communities, including the highest abundance species at each endpoint and exclusion of FP (Figure 5D). Five of the nine low temporal variability communities were well predicted by the model, whereas four were not accurately predicted (Figure 5E, i–v and vi–ix, respectively). In particular, the model incorrectly predicted the persistence of FP and PJ in several communities (Figure 5E, communities vii and ix, vi and viii, respectively). The challenge in accurately predicting FP abundance was consistent with the gLV model's disproportionately low prediction accuracy for this species on the original 10-member test data (Figure 4B; Pearson rho = 0.46). By contrast, PJ was very accurately predicted in the 10-member test set (Pearson rho = 0.99).

We performed an analysis of model uncertainty to provide deeper insights into poor predictions for communities vi–ix given that (1) biological data are usually noisy and (2) design of the 2–4 member communities represents an extrapolation (prediction of new conditions that differ substantially from training data) from the 10-member community training data. Indeed, training data noise and extrapolation beyond the training data are common challenges for building predictive mathematical models.[53] To this end, we analyzed model uncertainty with bootstrapping, which uses random sampling of the data with replacement to generate many training datasets from which an ensemble of models is inferred.[54] The resulting distributions of parameters across the ensemble provide insights into parameter uncertainty. For example, a sharply peaked distribution (i.e., low interquartile range) indicates low uncertainty about a given parameter, whereas a wide distribution (i.e., high interquartile range) indicates high parameter uncertainty. Parameter distributions are well constrained for the species in the poorly predicted communities (Figure S5A; FP and PJ indicated with asterisks).

We evaluated model predictive uncertainty by using the bootstrapped ensemble of models to generate predictive distributions for the poorly predicted communities (Figures 5Evi–5Eix). The predictions of the ensemble of models derived from bootstrapping were consistent with the measured relative abundance of FP in these communities with high frequency (i.e., the largest histogram bins aligned with the measured relative abundances) (Figures S5B–S5E). The ensemble of models predicted a multimodal distribution of PJ abundance in these communities, with one mode corresponding to the gLV model's best parameter estimate and a second mode corresponding to the measured relative abundance (Figures S5D and S5E). Therefore, the model either predicted the persistence or exclusion of PJ in the 2–4 member communities depending on variations in the training data. The inaccurate predictions of FP and PJ in communities vi–ix were made with large uncertainty. In contrast, the ensemble of models predicted CA and DL with higher confidence (community iv), consistent with the experimental data (Figure S5F). This implies that certain species were predicted with higher or lower confidence in 2–4 member communities despite similar prediction performance in the 10-member data (Figure 4B). For cases with low confidence, variations in the training data yielded different gLV model parameters, which, in turn, combined to generate large prediction uncertainty. Future efforts could leverage parameter and prediction uncertainty analysis to inform experimental designs for studying and engineering microbial communities. In sum, the analysis of gLV model uncertainty can provide insights into cases where the model predictions differ from experimental measurements.

## DISCUSSION

We currently lack a framework for designing interventions to precisely and predictably control microbial communities.[6] We demonstrate that despite their complexity, synthetic communities respond predictably to model-guided manipulation of media formulation and inoculation densities. Our results demonstrate that both media and inoculation densities are useful and complementary in steering communities to desired compositional states. Media factors determine the availability of metabolic niches, whereas inoculation densities can dictate the ability of species to secure the metabolic niches in a competitive environment either transiently or longer term. Data-driven dynamic and statistical modeling frameworks are developed for tuning these control points to optimize the endpoint Shannon diversity of a representative 10-member synthetic gut community to 91% of its maximum possible value (Figure 3F).

In each stage of the workflow, we exploit high-throughput, monoculture experiments to first characterize the "parts" of our microbial ecosystem and show that this information is useful for guiding community design. Based on the notion that monoculture growth is an influential variable in community assembly, we demonstrated that maximizing monoculture diversity as a function of media components substantially increased community diversity. The competition-based CSLE model informed by monoculture kinetics made useful predictions of inoculation densities that enabled efficient optimization of Shannon diversity using a DTL cycle (Figures 1I and S3J). Understanding what features of microbial communities can be predicted from monoculture information is a key scientific question in

the microbiome field. If monoculture information can forecast specific community-level properties, highly automated approaches for parallelized culturing could be exploited to inform the design of communities with desired functions. For example, synthetic human gut communities of ~100 members are now being studied *in vitro* and *in vivo*.[55] Although we selected a limited number of simple media control points, more complex resources such as fibers, peptones, and mucins have been shown to support high-richness communities from stool sample inocula. These media factors could be manipulated to expand the number of metabolic niches and tune species abundances in larger communities.[56,57] Central to our approach was mapping control inputs to outputs without the need for characterizing detailed biochemical mechanisms specific to a particular system (e.g., uptake and production kinetics of specific metabolites mediating interactions). Therefore, our model-guided strategy should generalize to a wide range of synthetic communities and media environments. Furthermore, although we focused on optimizing Shannon diversity as our objective function, our framework could be applied to design communities with tailored compositions or target functions (e.g., production of key health-relevant metabolites).

One limitation of our approach is that species with very low fitness in the community may not achieve a target abundance even with a high inoculation density. Future efforts could leverage the gLV model-guided design of multiple inoculation timings as an additional control point for endpoint community composition. For example, a species like BL that does not grow well in certain community contexts due to negative interactions could be given a "head start" by inoculating at an earlier time point.

As a proof of concept that inter-species interactions can be leveraged to design temporal behaviors, we used a data-driven gLV model to guide the design of communities with low variability of species composition over time (Figures 5B and 5C). Theoretical analyses of the gLV model tend to investigate qualitative long-term behaviors (e.g., exclusion, stable steady states, or limit cycles) to which many different initial conditions converge.[3] By contrast, the measured endpoint community composition of batch culture does not necessarily represent a system's stable steady state (e.g., a composition that does not change as a function of time in continuous culture or over multiple passages). This endpoint community composition measurement was nonetheless predicted quite accurately by fitting non-equilibrium trajectories of the model to experimental data (Figure 4B). Therefore, although our gLV model has constrained steady states that may not match the experimental system, it can still be used to design community compositions in batch culture as a function of initial conditions by exploiting the flexible transients of the model.

The merits of different mathematical modeling approaches for microbial communities have aptly garnered much attention.[14] With experimental systems like synthetic communities, a consideration of perhaps equal importance is "what data should be collected to inform the model?" The use of automated liquid handling to array synthetic communities according to optimal experimental designs (DoE) offers a practical approach for efficiently exploring these high-dimensional biological systems. Although the DoE framework has typically been implemented with linear models, this approach may broadly benefit parameter

inference for other models used to study microbial communities. Bayesian experimental design approaches could be used in future work to leverage parameter and prediction uncertainty to maximize the information from limited experiments using DTL cycles.[58] Furthermore, machine learning models like recurrent neural networks are flexible to various inputs and outputs and could be used to design community dynamics and functions using initial species abundances and key resources as simultaneous control points.[59]

Defined microbial communities hold great promise for many applications including sustainable agriculture, production of valuable compounds from renewable resources, and precision and personalized medicine.[60] Our systematic approaches for community control could be adapted as bioprocess engineering strategies to manufacture defined consortia as therapeutics in a scalable fashion. These methods could also be used to tune community member proportions and optimize key metabolite outputs for industrial bioprocessing in which metabolic pathways are distributed across community members to exploit the benefits of division of labor.[52,61] Eventually, the ability to identify and influence analogous control parameters for microbiome composition and function could be used to steer a patient's dysbiotic microflora toward a healthy state. Similar to media formulation, changes in diet are well documented to shape gut microbiome composition.[62] Dosage (analogous to inoculation density) was a critical factor in the successful redesign of the first phase three clinical trial of a donor-derived live bacterial therapeutic for treating recurrent *C. difficile* infection.[63] Overall, initial population densities (i.e., dosage), environmental resources, and inter-species interactions should be considered key control parameters for the model-guided design of microbial community dynamics and functions.

## STAR★METHODS

Detailed methods are provided in the online version of this paper and include the following:

- KEY RESOURCES TABLE
- RESOURCE AVAILABILITY
  - Lead contact
  - Materials availability
  - Data and code availability
- EXPERIMENTAL MODEL AND SUBJECT DETAILS
  - Species maintenance, precultures, and growth media
- METHOD DETAILS
  - Genomic DNA extraction, DNA library preparation, and sequencing
  - Monoculture media screening experimental procedures
  - Community inoculation density experimental procedures
  - Passaging experimental procedures
  - Scale up, CFU plating, determination of calculated viable profile, and plate scrapes
- QUANTIFICATION AND STATISTICAL ANALYSIS
  - Bioinformatic analysis of strain abundances
  - Quantification and modeling of monoculture growth for media optimization

- ○ Media optimization algorithm
- ○ Quantification of monoculture growth kinetics over a range of inoculation densities
- ○ Design algorithm for first community inoculation density experiment (DTL1)
- ○ Design algorithm for subsequent inoculation density experiments
- ○ Generalized Lotka-Volterra model training, validation, and supernatant growth effect comparison
- ○ Design algorithm for temporal variability subset communities
- ○ Bootstrap analysis of parameters and prediction uncertainty for the gLV model
- ○ Derivation of the Constrained System of Logistic Equations from Mass-action Type Kinetics
- ○ Data Exclusion

## SUPPLEMENTAL INFORMATION

## ACKNOWLEDGMENTS

We would like to thank Susan E. Hromada and Yili Qian for their helpful discussions. The research was sponsored by the Bill and Melinda Gates Foundation and was accomplished under grant number OPP1211893, National Institutes of Health under grant numbers R35GM124774 and R01EB030340, and the Army Research Office under grant number W911NF1910269.

## AUTHOR CONTRIBUTIONS

B.M.C., O.S.V., and R.L.C. conceived the study. B.M.C. carried out the experiments. B.M.C. implemented computational modeling. J.T. assisted with model development. B.M.C., O.S.V., and B.F.P. analyzed data. B.M.C. and O.S.V. wrote the paper, and all authors provided feedback on the manuscript. S.E. and R.L.C. assisted in experimental data collection. O.S.V. and B.F.P. secured funding.

## DECLARATION OF INTERESTS

B.M.C., O.S.V., and B.F.P. are inventors on a provisional patent application filed by the Wisconsin Alumni Research Foundation (WARF) with the US Patent and Trademark Office, which describes and claims concepts disclosed herein (application no. 63/306,691 Filing Date: 2/4/2022).

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

# STAR★METHODS

## KEY RESOURCES TABLE

| REAGENT or RESOURCE | SOURCE | IDENTIFIER |
|---|---|---|
| **Bacterial strains** | | |
| All bacterial strains are listed in Table S1 | N/A | N/A |
| **Chemicals** | | |
| Calcium chloride | Sigma-Aldrich | Cat# C5670-100G |
| Copper(II) sulfate | Sigma-Aldrich | Cat# 209198-100G |
| Pyridoxal hydrochloride | Sigma-Aldrich | Cat# P6155-5G |
| Thymidine | Sigma-Aldrich | Cat# T1895-1G |
| Xanthine sodium salt | Sigma-Aldrich | Cat# X3627-1G |
| Folic Acid | Alfa Aesar | Cat# J62937-06 |
| Orotic Acid potassium salt | Sigma-Aldrich | Cat# O2875-100G |
| Ca-D-pantothenate | TCI | Cat# P0012 |
| Cobalamin (B12) | TCI | Cat# C0449 |
| Pyridoxine HCl | DOT Scientific | Cat# DSP50240-50 |
| Riboflavin DOT | Scientific | Cat# DSR64040-100 |
| Tetrahydrofolic Acid | Cayman Chemical | Cat# 18263 |
| Thiamine HCl | Sigma-Aldrich | Cat# T1270-25G |
| p-Aminobenzoic Acid | DOT Scientific Inc | Cat# DSA20050-25 |
| Pyridoxamine | Chem Impex | Cat# 01461 |
| Ammonium Chloride | Fisher Scientific | Cat# A661-500 |
| Magnesium Chloride | Fisher Scientific | Cat# BP214–500 |
| Potassium phosphate monobasic | Alfa Aesar | Cat# 11594-1KG |
| Nicotinamide | Sigma-Aldrich | Cat# N0636-100G |
| Hematin | Santa Cruz | sc-207729B |
| Biotin | DOT Scientific | Cat# DSB40040-1 |
| L-alanine | DOT Scientific | Cat# DSA20060-100 |
| L-arginine | DOT Scientific | Cat# DSA50010-1000 |
| L-asparagine | DOT Scientific | Cat# DSA50030-100 |
| L-Aspartic acid | DOT Scientific | Cat# DSA50060-100 |
| L-Cysteine | DOT Scientific | Cat# DSC81020-500 |
| L-Glutamic acid | Sigma-Aldrich | Cat# G1251-500G |
| L-glutamine | Acros Organics | Cat# 119951000 |
| L-Glycine | Acros Organics | Cat# 120070010 |
| L-histidine | VWR | Cat# 1B1164-100G |
| L-isoleucine | DOT Scientific | Cat# DSI54020-25 |
| L-leucine | DOT Scientific | Cat# DSL22000-500 |
| L-lysine | DOT Scientific | Cat# DSL37040-500 |
| L-methionine | DOT Scientific | Cat# DSM22060-100 |
| L-phenylalanine | DOT Scientific | Cat# DSP2060-100 |
| L-proline | DOT Scientific | Cat# DSP50200-100 |
| L-serine | DOT Scientific | Cat# DSS22020-100 |
| L-threonine | DOT Scientific | Cat# DST21060-100 |
| L-tryptophan | DOT Scientific | Cat# DST60080-25 |
| L-valine | DOT Scientific | Cat# DSV42020-100 |
| L-tyrosine | Sigma-Aldrich | Cat# T3754-50G |
| MOPS | Fisher Scientific | Cat# BP308–500 |

*(Continued on next page)*

*Continued*

| REAGENT or RESOURCE | SOURCE | IDENTIFIER |
|---|---|---|
| Sodium Bicarbonate | Sigma-Aldrich | Cat# S5761-500G |
| Glucose | VWR | Cat# BDH9230-500G |
| Arabinose | CHEM-IMPEX | Cat# 01654 |
| Maltose | Sigma-Aldrich | M5885-100G |
| Yeast extract | Gibco | 288620 |
| 10x ACGU Solution | Teknova | M2103 |
| ATCC Mineral Supplement | ATCC | Cat# MD-TMS |
| Sodium acetate | Sigma-Aldrich | Cat# S5636-250G |
| Reinforced clostridial media | BD | Cat# 218081-500G |
| Anaerobic basal broth | Oxoid | Cat# CM0957 |
| Nitrogen | Airgas | Cat# NI200 |
| Mixed gas | Airgas | Cat# X03NI75C30057W3 |
| Anaerobe Basal Broth | Oxoid | Cat# CM0957 |
| Phusion High-Fidelity DNA Polymerase | Fisher Scientific | Cat# F530L |
| Lysozyme | Sigma-Aldrich | L6876-10G |
| Proteinase K | Meridian | BIO-37039 |
| PBS | MP Biomedicals | Cat# PBS10X02 |
| Ethanol | Koptec | Cat# 2716 |
| Molecular grade water | VWR | Cat# VWRL0201-0500 |
| **Critical commercial Assays** | | |
| Zymo DNA Clean & Concentrator kit | Zymo Research | Cat# D4013 |
| Quant-iT dsDNA Assay Kit | Invitrogen | Cat# Q33120 |
| MiSeq Reagent Nano Kit v2 | Illumina | Cat# MS-103-1003 |
| AL buffer | Qiagen | 19075 |
| AW1 buffer | Qiagen | 19081 |
| AW2 buffer | Qiagen | 19072 |
| AE buffer | Qiagen | 19077 |
| **Deposited Data** | | |
| Processed sequencing for relative abundance and optical density data | This paper | https://doi.org/10.5281/zenodo.8356547 |
| Raw 16S rRNA amplicon sequences for relative abundance | This paper | SRA:PRJNA1030060 or https://doi.org/10.5281/zenodo.8360495 |
| **Software and Algorithms** | | |
| Python 3.7 | Open-source | https://www.python.org |
| MATLAB R2020a | MathWorks | https://www.mathworks.com |
| JMP 15 | SAS | https://www.jmp.com/ |
| PEAR v0.9 | Zhang et al.[64] | PMID: 24142950 |
| mothur v1.40.5 | Schloss et al.[65] | PMID: 19801464 |
| Analysis and computational modeling conducted in this study | This paper | https://doi.org/10.5281/zenodo.8356547 |
| Generate maximally perceptually-distinct colors | Open-source | https://www.mathworks.com/matlabcentral/fileexchange/29702-generate-maximally-perceptually-distinct-colors |

## RESOURCE AVAILABILITY

### Lead contact

Further information and requests for resources and reagents should be directed to and will be fulfilled by the lead contact, Ophelia S. Venturelli (venturelli@wisc.edu).

CellPress

## Materials availability

This study did not generate new materials. All strains were obtained from ATCC or DSMZ as indicated in Table S1.

## Data and code availability

Raw 16S rRNA amplicon sequencing files have been deposited at SRA and Zenodo and are publicly available as of the date of the publication. DOIs are listed in the key resources table.

All processed data are available from Zenodo using the DOIs listed in the key resources table.

All original code has been deposited at Zenodo and is publicly available as of the date of publication. All original code is available from Zenodo using the DOIs listed in the key resources table.

Any additional information required to reanalyze the data reported in this paper is available from the lead contact upon request.

## EXPERIMENTAL MODEL AND SUBJECT DETAILS

### Species maintenance, precultures, and growth media

The following methods are adapted from Venturelli et al.,[18] Clark et al.,[24] and Venturelli et al.[66] All anaerobic culturing was carried out in a custom anaerobic chamber (Coy Laboratory Products, Inc) with an atmosphere of 2.5 ± 0.5% $H_2$, 15 ± 1% $CO_2$ and balance $N_2$. All prepared media, stock solutions, and materials were placed in the chamber at least overnight before use to equilibrate with the chamber atmosphere. The species used in this work were obtained from the sources listed in Table S1 and permanent stocks of each were stored in 25% glycerol at $-80\,^{\circ}C$. Batches of single-use glycerol stocks were produced for each species by first growing a culture from the permanent stock in anaerobic basal broth (ABB) media (HiMedia or Oxoid) to stationary phase, mixing the culture in an equal volume of 50% glycerol, and aliquoting 400 μL into Matrix Tubes (ThermoFisher) for storage at $-80\,^{\circ}C$. Quality control for each batch of single-use glycerol stocks included (1) plating a sample of the aliquoted mixture onto LB media (Sigma-Aldrich) for incubation at 37°C in ambient air to detect aerobic contaminants and (2) next-generation DNA sequencing of 16S rDNA isolated from pellets of the aliquoted mixture to verify the identity of the organism (Illumina). For each experiment, precultures of each species were prepared by thawing a single-use glycerol stock and combining the inoculation volume and media listed in Table S1 to a total volume of 5 mL for stationary incubation at 37°C. Incubation times are also listed in Table S1. Our data suggested that lot-to-lot variability between freezer stocks may have caused variation in community assembly (which was otherwise highly reproducible). Future efforts may consider subculturing of precultures for one passage prior to community inoculation to reduce potential lot-to-lot variation by allowing populations to synchronize, facilitating reproducible assembly of diverse and viable communities. Prior to inoculating starter cultures, the workspace and pipettes were cleaned with Spor-klenz (STERIS), and again with 70% ethanol between species inoculations.

## METHOD DETAILS

### Genomic DNA extraction, DNA library preparation, and sequencing

DNA extraction, library preparation, and sequencing were performed according to methods described in Hromada[24] and Clark et al.[66] Cell pellets from 150 uL of culture were stored at -80C following experiments. Genomic DNA was extracted using a 96-well plate adaption of the DNeasy protocol (Qiagen). Genomic DNA was normalized to 1 ng/uL in molecular grade water, and stored at -20C. Dual-indexed primers for multiplexed amplicon sequencing of the v3-v4 region of the 16S gene were designed as described previously, and arrayed in 96-well, skirted PCR plates (Thomas Scientific) using an acoustic liquid handling robot (Echo LabCyte). Genomic DNA and PCR master mix were added to primer plates and amplified prior to sequencing on an Illumina MiSeq platform using a MiSeq Nano Kit v2 (Illumina).

### Monoculture media screening experimental procedures

The media screening experiment was designed to improve monoculture-diversity (Equation 4) using DM38, a chemically defined medium developed in our laboratory, and referenced as the "baseline" medium in the text. Table S2 contains the medium and stock solution recipes referenced in this section. A four-factor, two-level half factorial screening design with appended center point condition was constructed in JMP 15 (SAS institute). "High" absolute design levels for sugar mixture, amino acid mixture, and pH variables (these are key components in DM38) were set at their respective DM38 concentrations. Yeast extract (sterile filtered, not autoclaved) was included to support monoculture growth of *F. prausnitzii*, as keenly observed by D'Hoe et al.[17] "Low" design levels were set at 0 g/L for sugars, amino acids, and yeast extract, and 5.7 for pH (according to generally reported ranges for the human large intestine).[40] Stock solutions of sugars, amino acids, and yeast extract were prepared at 20x v/v of their target "high" concentrations, and sterile filtered. The nine media were arrayed according to the experimental design in 2mL deep-well blocks (Nest), using a Tecan Evo liquid handling robot to aliquot the appropriate volume of 20x stocks into 1.4x base medium. The final concentration was brought to 1x using sterile water. The deep well blocks, containing ten sets of the media experimental design, were inoculated from the ten precultures to a 600nm optical density (OD600) value of 0.01. Optical density was measured using 200 uL of sample in a Tecan F200 plate reader in standard clear, flat bottom 96-well microplates (Grenier). Inoculation volumes were calculated as Volume$_{(inoc)}$ = Volume$_{(well)}$*0.01 OD600 / (Preculture OD600). Inoculation was performed from a sterile trough with a multichannel pipette. Four 200 uL replicates were mixed and aliquoted to sterile, clear, flat bottomed, 96-well microplates (Grenier), covered with a transparent

seal (Breath EZ, Diversified Biotech), and incubated at 37°C in the Tecan Evo incubator. Automated OD600 measurements were recorded every two hours for about 60 hours with a Tecan F200 plate reader.

### Community inoculation density experimental procedures

Experimental designs were arrayed with a Tecan Evoware liquid handling robot. Before inoculation, precultures were centrifuged at 4000 rpm, 7.5 minutes in a Sorvall ST 16R centrifuge (Thermo Scientific). Anaerobically, the supernatant was decanted, the pellet was vortexed, and resuspended in fresh optimized medium using a serological pipette (Drummond). Two 24-well blocks were used to array various densities of the precultures. The top row contained a high-density preculture, the second row contained a mid-density preculture, and the third row contained a low-density preculture. The concentration of the high-density preculture well for each species was calculated by finding the number of ten-fold dilutions of the measured preculture OD which resulted in the smallest inoculation volume greater than 7 uL. In other words, we calculated the lowest volume that can be accurately pipetted by the robot to inoculate the deep well block to its target "high" experimental level. For example, if species A grew to a preculture OD of .2 and was to be inoculated to a target "high" level of .0001 in a volume of 700 uL, then the high-density preculture well would contain a hundred-fold dilution of the preculture (.002 OD600), such that "high" experimental condition would be inoculated with V = .0001 OD * 700 uL / (.002) = 35 uL. This strategy was implemented because any volume less than 7 uL could not be pipetted accurately, while larger inoculum volumes would quickly accumulate and result in a scenario where the sum of all species inoculum volumes exceeds the target culture volume. The "mid" and "low" preculture wells were filled by diluting the "high" preculture well by the same x-fold ratio of the high to center point design levels (and equivalently the ratio between the center point and low levels). Two serial dilutions at this ratio were performed from high to mid, and mid to low preculture wells for each species, such that each species high, center point, and low design levels were inoculated with a constant volume from the high, mid, and low preculture wells, respectively. A 200 μL aliquot of the inoculated deep well block was transferred to a 200 μL microplate, covered with a breathable seal, and incubated in the Tecan F200 plater reader at 37°C. Labware and culture conditions were consistent between monoculture and coculture, as it should be noted that differences in labware geometries, particularly surface to volume ratios, can affect anaerobic microbial growth dynamics. Optical density measurements were recorded at 28 +/- 1 hour in the plate reader. Next, 150 μL of the endpoint culture was transferred to a sterile 1mL deep well block and centrifuged at 2400xg for 10 minutes. The supernatant was removed, and the pellet was stored at -80°C. Next, 20 μL of the supernatant was used to measure pH using a spectrophotometric phenol red assay, as described in Clark et al.[24]

### Passaging experimental procedures

A serial subculture is performed by mixing well and diluting 20 uL of the endpoint community culture into 500 uL of fresh medium (25-fold v/v). The new culture is then aliquoted (200 uL) into a microplate and incubated as previously described. This process was performed three times for the first inoculum design (DLT cycle 1) and once for DTL cycle 2.

### Scale up, CFU plating, determination of calculated viable profile, and plate scrapes

Cultures at 100 mL scale were performed in autoclaved 100 mL bottles. Inoculum density conditions for the cultures in Figure 3G corresponded to DTL3 center point (condition 13, Table S3), while the culture in Figure 3H corresponded to the inoculum density conditions that yielded the best diversity (DTL3 condition 6, Table S3). Cultures were inoculated and incubated anaerobically at 37°C without agitation. Preculture and total community CFU counts were determined on modified ABB and RCM agar (Table S2), which together supported the growth of all species on at least one agar type. Calculated viable profile in Figure 3H is determined for precultures by dividing preculture CFU by the dilution factor between preculture and main culture; in other words, it represents the inoculation density of the main culture in units of CFU/mL. The calculated viable profile for the 28-hour community culture was determined by multiplying the species relative abundance as determined by NGS by the total CFU count corresponding to the agar on which that species preculture had the higher colony count (Figure S4E). We first verified that liquid monocultures were not beyond stationary phase at this timepoint (Figure S1B), and that all species grew in monoculture CFU plates on these agar (Figure 3H light bars). We acknowledge the potential for substantial bias in this calculation of viable profile, due to variations in plating efficiency across species and potential genome extraction or sequencing bias.

   After counting, each CFU assay spread plate was suspended by adding 1 mL of PBS to the plate, scraped and resuspended with a disposable spreader, aspirated into an Eppendorf tube, and vortexed to resuspend. Each plate's homogenized suspension was sequenced as previously described, in order to determine whether viable cells of each species could be detected in the community culture.

## QUANTIFICATION AND STATISTICAL ANALYSIS

### Bioinformatic analysis of strain abundances

Sequencing data were analyzed as described in Venturelli et al.[18] Basespace Sequencing Hub's FastQ Generation demultiplexed the indices and generated FastQ files. Paired reads were merged using PEAR (Paired-End reAd mergeR) v0.9.[64] Reads were mapped to a reference database of species used in this study, using the mothur v1.40.5, and the Wang method.[65,67] Relative abundance was calculated by dividing the read counts mapped to each organism by the total reads in the sample. Estimated absolute abundance

was calculated by multiplying the relative abundance of an organism by the OD600 of the sample. Samples were excluded from further analysis if > 1% of the reads were assigned to a species not expected to be in the community (indicating contamination).

### Quantification and modeling of monoculture growth for media optimization

Model-guided optimization of community Shannon diversity (Equations 1 and 2) was performed by modeling monoculture growth response (Equation 3) on various media. "Monoculture diversity" (Equations 4 and 5) was used as a proxy function for Shannon diversity, enabling a monoculture-based approach for manipulating community Shannon diversity.

$$Shannon\ diversity: -\sum_{i=1}^{Species} X_{fr,i}\ \ln X_{fr,i} \qquad \text{(Equation 1)}$$

$X_{fr,i}$ — fractional abundance of species "$i$" in a community

$$Fractional\ Abundance: X_{fr,i} = \frac{X_i}{\sum_{i=1}^{species} X_i} \qquad \text{(Equation 2)}$$

$X_i$ — absolute abundance of species "$i$"

$$Logistic\ differential\ equation: \frac{dX}{dt} = \mu\left(1 - \frac{X}{K}\right) \qquad \text{(Equation 3)}$$

$dX/dt$ — rate of population growth

$\mu$ — specific growth rate parameter

$K$ — carrying capacity $(i.e., steady-state\ population\ size)$

$$Monoculture - Diversity - \sum_{i=1}^{Species} K_{fr,i}\ \ln K_{fr,i} \qquad \text{(Equation 4)}$$

$K_{fr,i}$ — Normalized carrying capacity of species "$i$"

$$Normalized\ carrying\ capacity\ of\ species\ "i": K_{fr,i} = \frac{K_i}{\sum_{j=1}^{species} K_j} \qquad \text{(Equation 5)}$$

$\sum K_j$ — Sum of logistic carrying capacities in a particular medium

Monoculture timeseries growth data from the media screening experiment was fit with logistic differential equations (Equation 3), and the carrying capacity parameter was used as a readout of growth response. Carrying capacity serves as a "smoothed," time independent maximum growth value. Smoothing is required because raw data may contain outlier values due to condensation on the transparent plate seal or other technical variability. If computational resources or expertise are limited, the growth response could also be taken as the maximum value of a smoothed timeseries (e.g. after applying a running average filter). The baseline of the OD600 timeseries data was computationally "blanked" (i.e. normalized) to the known inoculum density by subtracting the difference between the time-zero measured value and known inoculum from the entire timeseries. Each model fit was performed independently using bounded, nonlinear regression with MATLAB's "fmincon" function, which returns the logistic parameter set $(\mu, K)$ that minimizes the sum of squared errors between the model predictions and the experimental data. All timeseries were truncated to 30 hours. Outlier detection was performed by comparing the z-score of the mean OD600 across replicates, to omit replicates that did not grow.

Multivariate polynomial regression models (Equation 6) were fit to predict carrying capacity parameter of each species (growth response) as a function of the scaled media design matrix (predictors).

*Media Regression Models (MR):*

$$\hat{K}_i = \sum_{l=1}^{4} \beta_l^{M.E.} x_l + \sum_{l=1}^{4} \beta_l^{Q.E.} x_l^2 + \sum_{l=1}^{3}\sum_{m=l+1}^{4} \beta_p^{I.X.2} x_l x_m \ldots \qquad \text{(Equation 6)}$$

$$+ \sum_{l=1}^{2}\sum_{m=l+1}^{3}\sum_{n=m+1}^{4} \beta_q^{I.X.3} x_l x_m x_n$$

$$\widehat{K}_i - \text{predicted carrying capacity of species } "i"$$

$$\beta_I^{M.E.} - \text{main effects parameters}$$

$$x_I - \text{predictors (media component variables)}$$

$$\beta_I^{Q.E.} - \text{quadratic main effects parameters}$$

$$\beta_p^{I.X.2} - \text{interaction parameters, 2nd order}$$

$$\beta_q^{I.X.3} - \text{interaction parameters, 3rd order}$$

We note that although the model is a multivariate polynomial function of the design variables, the regression is linear with respect to the parameters, as the higher order predictors are treated as "new" variables whose value is calculated prior to regression. The polynomial structure (Equation 6) contained linear terms ($X_1$), quadratic terms ($X_1^2$), and both second and third order interaction terms ($X_1 * X_2$ and $X_1 * X_2 * X_3$). The double and triple sum terms in this equation represent the upper triangular matrix of unique two-factor interaction parameters and three-dimensional upper triangular matrix of third order interaction parameters ($X_1 * X_2 = X_2 * X_1$ so only one of these predictor terms should be included). The estimation of quadratic terms is contingent on the inclusion of a center point condition in the otherwise two-level experimental design. Because the models are data limited, elastic net regularization and nested cross validation were performed to reduce overfitting. The elastic net and regularization coefficient hyperparameters were selected using a "grid search" approach, and MATLAB's "lasso" function. For each species, the 9-condition dataset (9x16 predictor matrix and 9x1 growth response vector) was partitioned into all nine possible combinations of eight conditions (rows) using MATLAB's "cross-valind" function (first partitioning). The "lasso" function is called with the cross-validation argument, which performs a second round of leave-one-out cross validation to identify the regularization and elastic net coefficients (hyperparameters) that minimize the out-of-fold mean sum of squared errors for the "internal" cross validation sets. Only the hyperparameters, but not the regression parameters, are returned at this stage. The Lasso function is then called again without the cross-validation arguments, receiving the previously identified hyperparameters as arguments to find a best fit parameter set for the "first partitioning" of the original dataset. This is performed for each partition of the original dataset, such that each regression model is an ensemble model with nine parameter sets, each corresponding to one "leave-one-out" partitioning of the data. Each parameter set has its own, independently identified hyperparameters, such that none of the hyperparameters are biased by training on the entirety of the dataset. The models are validated by making "out-of-fold predictions", meaning using the parameters trained on each of the nine partitions of eight datapoints to predict the one datapoint that is not contained in that partition. When the models are called to make a new prediction (e.g. for the optimization script), the nine predictions of the "ensemble" are averaged to a scalar value. The repository path of the script implementing these methods is BC031/BC031_xval_opt_final.m''.

### Media optimization algorithm

A constrained optimization problem was solved using MATLAB's "fmincon" function to solve for the concentration profile of sugar mixture, amino acid mixture, yeast extract, and pH that maximized the monoculture-diversity (Equations 6, 7, and 8).

*Objective function for media optimization*:

$$maximize \left( -\sum_{i=1}^{Species} \widehat{K}_{fr,i} \ln \widehat{K}_{fr,i} \right) \qquad \text{(Equation 7)}$$

*Predicted normalized carrying capacity of species i* :

$$\widehat{K}_{fr,i} = \frac{\widehat{K}_i}{\sum\limits_{j=1}^{species} \widehat{K}_j} \qquad \text{(Equation 8)}$$

The upper and lower bound arguments to the "fmincon" function are set to constrain the solution within the original experimental design levels (sugars between 0 and 9.45 g/L, yeast extract between 0 and 2 g/L, amino acids between 0 and 10.7 g/L, and pH between 5.7 and 6.7). The function is initialized with a random guess of the sugars, amino acids, yeast extract, and pH concentrations. The "objective function" references the received concentration inputs and calls the linear regression models to make a prediction of each species carrying capacity from this set of media component concentrations. From these ten carrying capacity predictions, the

predicted monoculture diversity is calculated. The "fmincon" function then iteratively solves for the single concentration of the resources that maximizes the predicted monoculture diversity, using the default interior point algorithm. The repository path of the script implementing these methods is "code/BC031/BC031_xval_opt_final.m".

### Quantification of monoculture growth kinetics over a range of inoculation densities

Deep well blocks (96-well, 2mL, Nest) were filled with 1000uL of the optimized medium. Species were precultured and inoculated into each of the first ten wells of the first row of the block at a density of 0.01 OD600 as previously described. A multichannel pipet was used to mix and perform six 10-fold volume/volume serial dilutions of the first row down the rows of the plate. Three replicate 96-well microtiter plates with 200 uL in each well were aliquoted from the deep well block and covered with a transparent and breathable seal. Plates were incubated and timeseries OD600 was recorded as previously described.

Time series data from inoculum conditions that did not result in reproducible growth were omitted from the dataset, and data was normalized as previously described. The low inoculum densities resulted in growth curves that appeared to have a long lag phase, but were likely to be in exponential growth phase at a biomass density that was far below the limit of detection of the plate reader. The exponential and stationary phase data from individual species growth data was isolated as values greater than the assumed 0.05 lower limit of detection for the plate reader. The true limit of detection of the reader is .001, but data below ∼.05 has low signal-to-noise ratios for automated microbial growth. As such, the "measured" initial conditions were omitted from the dataset, as they generally reflected the low limit of detection of the plate reader. Nonlinear regression was used to solve for the single logistic parameter set $(\mu, K)$ and the set of initial conditions (one for each growth curve in the set) that minimized the sum of squared errors between the model predictions and the exponential phase data. A vector of two logistic parameters and one-to-six initial conditions (depending on how many dilutions grew reproducibly) was passed as variables to the "fmincon" solver. The objective function then parsed the vector into initial conditions and ODE parameters, then called an ODE solver to generate model predictions. The value of the objective function is the sum of mean squared errors between the model predictions and the exponential phase data for all growth curves in the set. The "fmincon" function returns the vector of parameters and initial conditions that minimize the objective function. The computationally fitted initial conditions were plotted in log-log space against the experimental initial conditions, and a first order linear regression was performed to map the log transformed experimental initial conditions to the log transformed, computationally fitted initial conditions, using sets of values that fell in the linear range of the plate reader.

An alternative to the previously described method entails simultaneously inferring all CSLE parameters from the set of monoculture timeseries data, obviating the empirical parameterization of $K_{comm}$, and the fitting and mapping of initial conditions (Figure S3I, code available in "CLSEpredictComms.m" script). The CSLE model parameters were in this case inferred from all monoculture experimental data using MATLAB's "fmincon" solver to minimize a cost function. The cost function consisted of the sum of squared errors between the model predictions and data, with an L1 regularization penalty to minimize overfitting, as previously described.[18] The upper bounds for growth rate terms $\mu_i$, species specific carrying capacity terms $K_i$, and $K_{comm}$, were 10, 10, and 5, respectively. The lower bounds for these parameters were all 0. The "MaxFunctionEvaluation" and "MaxIterations" arguments for "fmincon" were both set to "Inf" via the "optimoptions" function to allow the solver sufficient time to converge. The value of $K_{comm}$ was inferred as 0.76 OD600. The repository path of the script implementing these methods is "code/BC035/modeling/BC035_analysis_v4a.m".

### Design algorithm for first community inoculation density experiment (DTL1)

The experimental design chosen for the first inoculum screening was a nine-factor, three-level definitive screening design.[44] These designs have three levels for each variable, improving estimation of the quadratic effects that are likely important for approximating the endpoint of exponential microbial growth with a polynomial function. The scaled design matrix was constructed in JMP 15. Inoculum concentrations were assigned to the scaled experimental design levels using solutions from the constrained system of logistic equations model. The constrained system of logistic equations was simulated in MATLAB, using the growth rate and carrying capacity parameters as fitted to monoculture data (described in the previous section).

*Constrained System of Logistic Equations*:

$$\frac{dX_i}{dt} = f(\mathbf{X}) = \mu_i\left(1 - \frac{X_i}{K_i}\right)\left(1 - \frac{\sum X_j}{K_{comm}}\right)X_i \tag{Equation 9}$$

$$and : \widehat{X}_{F,i} = \int_{t0}^{tF} f(\mathbf{X})\, dt$$

$dX_i/dt$ — rate of change of species "i"

$\mu_i$ — specific growth rate

$K_i$ — logistic carrying capacity

$$K_{comm} - community\ carrying\ capacity\ (total\ growth\ constraint)$$

$$\widehat{X}_{F,i} - predicted\ endpoint\ abundance\ of\ species\ "i"$$

The community carrying capacity parameter $K_{comm}$ was taken as the maximum OD600 ($K_{comm} = 0.89$ OD600) of a full community culture inoculated from an even inoculum (all species inoculated to 0.001 OD600). To find the set of initial conditions that maximized the Shannon diversity of the CSLE model at steady state, a constrained optimization problem was solved with MATLAB's "fmincon" function. The variables optimized by the "fmincon" solver consisted of the set of all species initial conditions. The objective function internally maps these initial conditions to the computational space equivalent (using the linear regression functions previously described), and simulates community growth by calling a CSLE ODE function. The "fmincon" solver solves for the set of initial conditions that maximize the Shannon diversity (Equation 1) of the steady state population abundances using the default interior point algorithm.

*Objective Function Maximizing Shannon diversity of CSLE prediction*

$$maximize\left(-\sum_{i=1}^{Species}\widehat{X}_{F,fr,i}\ln\widehat{X}_{F,fr,i}\right) \tag{Equation 10}$$

$$\widehat{X}_{F,fr,i} - , fr, i(10)\ endpoint\ fractional\ abundance\ of\ species\ "i"\ by\ the\ CSLE$$
model

The initial condition solutions are constrained by lower bounds of the experimental inoculum conditions that did not grow, such that the solver does not return initial condition that are too low to use in practice (an issue that can arise when modeling populations as continuous numerical variables). The total inoculum is constrained using a linear inequality argument such that the sum of all initial conditions did not exceed 0.02 (*F. prausnitzii* was fixed at 0.01; the sum of the other nine species was constrained to below 0.01). The high inoculum level for each species was solved for by fixing all other species initial conditions at the maximum diversity solution (center point), then finding the initial condition for that species which yielded a 3.3-fold higher steady state abundance than the center point condition. Specifically, "fmincon" was called to minimize the squared error between the simulation and 3.3 times the steady state abundance of that species maximum diversity solution as a function of that species initial condition. This was iteratively performed to find all species "high" initial condition levels for the experimental design. The low levels were set symmetrically to the "high" levels in log space, (e.g. the center point was multiplied and divided by the same x-fold factor), such that a CSLE simulation of the experimental design conditions predicted maximum diversity at the center point, and a 10-fold range of steady state abundances of each species occurred between "high" and "low" design levels. This approach accounts for the fact that a species with a very fast exponential growth rate will likely need a much larger perturbation (in comparison to a species with a low exponential growth rate) to its initial condition to achieve a similar change in the endpoint growth. The repository path of the script implementing these methods is these methods is "code/BC035/modeling/BC035_analysis_v4a.m".

### Design algorithm for subsequent inoculation density experiments
Linear regression with interactions (Equation 11) was used to predict the abundance of each species in the community from the inoculum design matrix, using the nested cross validation approach detailed in the media design methods section.

*Inoculum Regression Models (IR):*

$$\widehat{X}_{F,i} = \sum_{l=1}^{10}\beta_l^{M.E.}X_{0,l} + \sum_{l=1}^{10}\beta_l^{Q.E.}X_{0,l}^2 + \sum_{l=1}^{9}\sum_{m=l+1}^{10}\beta_p^{I.X.2}X_{0,l}X_{0,m} \tag{Equation 11}$$

$$\widehat{X}_{F,i} - predicted\ endpoint\ abundance\ of\ species\ "i"\ in\ community\ culture$$

$$X_0 - predictors\ (designed\ inoculum\ values)$$

$$\beta_l^{M.E.} - main\ effects\ parameters$$

$$\beta_l^{Q.E.} - quadratic\ main\ effects\ parameters$$

$$\beta_p^{I.X.2} - interaction\ parameters$$

The inoculum design matrix was $\log_{10}$ transformed to scale the values prior to fitting. The models trained on DTL 1 and 2 community data were evaluated on withheld test data (5 of 59 total conditions) to asssess the model's predictive capability (Figure 3E). Replicates were averaged prior to fitting to avoid biasing test/validation data with conditions contained in training data. Validation predictions and Pearson correlation coefficients for both cycles' models are shown in Figures S2K–S2M. Models that were deemed predictive

were used in a multi-objective optimization problem (Equation 12) to predict an updated center point for the new experimental design. Any desired target composition (not only even endpoint, i.e. maximum diversity) can be designed with this approach by updating this target vector with the desired endpoint abundances. Species whose models were not deemed predictive were adjusted using a rational "frameshift" strategy (Figures S3D and S3E). The "frameshift" involves selection of new design level absolute setpoints as follows: if a species overgrew (saturated response) in the previous experiment, the new center point level is set at the previous low level. If a species undergrew (non-measurable or very low growth in comparison to other species), its updated center point inoculum level is set at the previous "high" level. These new center point levels were thus equivalent to the extrema of the previous design space, and could be used as inputs to the regression models (without forcing the models to extrapolate beyond the bounds of training data). We note that the DTL process could probably be carried out using only the "frameshift" strategy to approach a design goal. The magnitude of the levels (x-fold of center point) was maintained between cycles one and two, unless the total range between high and low exceeded two orders of magnitude, in which case it was constrained to two orders of magnitude. In cycle three, the experimental design was modified to a twelve-run Placket-Burman screening with center point, with levels set at two-fold above and below center point. This adjustment of the levels initially informed by the CSLE model (cycle 1 levels) is a qualitative decision that reflects the purpose of the designs. Cycle one had large magnitude levels because it was meant to explore a large design space. Cycle two levels were constrained to two orders of magnitude or less to balance searching the design space with the probability of finding a high diversity condition. Cycle three levels were constrained to only two-fold because the purpose of the design was to demonstrate the robustness of a high confidence prediction to small variations, rather than to explore the design space and gather data for further model training.

A constrained multi-objective optimization problem was solved to minimize the error between target abundances and regression model predictions. This objective function is a more strict definition of maximizing Shannon diversity at a particular total species abundance, and was chosen because maximizing the Shannon diversity can return very low total growth solutions. Additionally, it is also a more flexible approach, as it allows the user to define an exact target community composition. We targeted an even endpoint abundance for each organism of magnitude (average community OD600) / (# of species), where the average community OD600 was the average endpoint OD600 across all the conditions of the previous experiment. The repository path of a script implementing these methods is "code/BC037/ BC037_designBC038.m".

*Objective Function for inoculum optimization*:

$$minimize \sum_{i=1}^{species} \left( X_{Targ,i} - \widehat{X}_{F,i} \right)^2 \tag{Equation 12}$$

$X_{Targ,i}$ — *target endpoint absolute abundance of species "i" (set to even abundances in this work to maximize diversity)*

## Generalized Lotka-Volterra model training, validation, and supernatant growth effect comparison

The parameters of a generalized Lotka-Volterra (gLV) model were fit to monoculture timeseries data and 10-member community initial and stationary phase data.

*generalized Lotka − Volterra model*:

$$\frac{dX_i}{dt} = \left( \mu_i + \sum_{j=1}^{n} a_{ij}X_j \right) X_i \tag{Equation 13}$$

$dX_i/dt$ − *rate of change of species "i"*

$\mu_i$ − *specific growth rate*

$a_{ii}$ − *intra − species interaction paramter* (*note*, $a_{ii} = -\mu_i/K_i$)

$a_{ij,i \neq j}$ − *inter − species interaction terms*

The training data additionally included three passages of the first inoculum screening and one passage of the third. The passages were treated as independent experiments with initial conditions calculated from the previous culture's endpoint abundances divided by 25 (corresponding to the volumetric dilution performed to inoculate). The gLV model was fit to experimental data using MATLAB's "fmincon" solver to minimize a cost function. The cost function consisted of the sum of squared errors between the model predictions and data, with an L1 regularization penalty to minimize overfitting, as previously described.[18] The upper bounds for growth rate terms $\mu_i$, self interaction terms $a_{ii}$, and inter-species interaction terms $a_{ij,i \neq j}$, were 3, 0, and 10, respectively. The lower bounds for these quantities were 0, -10, and -10, respectively. Self-interaction terms must be non-positive and growth rate terms must be non-negative to avoid divergence and maintain biological meaning. The "MaxFunctionEvaluation" and "MaxIterations" arguments for "fmincon" were both set to "Inf" via the "optimoptions" function to allow the solver sufficient time to converge. The solver was initialized

**Cell Systems**
Article

with the monoculture growth rates, monoculture derived self-interaction terms, and zeros as respective initial guesses for the gLV growth rates, gLV self-interaction terms, and gLV inter-species interaction terms. Zero is a logical initial guess for unknown parameters subject to L1 regularization, which pushes poorly constrained parameters towards zero. The community data was randomly partitioned into test and training+validation datasets consisting of 10% and 90% of the data, respectively, using MATLAB's "randsample" function. Monoculture data was not included in validation or test sets because it is collected at high-resolution time intervals, and thus does not evaluate the model's predictive performance on community data. The regularization coefficient was found by first scanning a logarithmic range of 10 values from $10^{-8}$ to $10^1$ and identifying the value that corresponded to the lowest averaged sum of squared errors across out-of-fold predictions, which also corresponded to the highest mean correlation coefficient for out-of-fold predictions (5-fold cross validation on training+validation data partitioned using MATLAB's "crossvalind" function). A higher resolution scan was then performed with 13 logarithmically spaced values between $10^{-2.5}$ and $10^{-0.5}$, centered around the previous best value ($10^{-1}$). A best estimate for the parameter set was then inferred from the training+validation dataset using a regularization coefficient of $10^{-0.5}$ (best value from the higher resolution scan). The predictive capability of the model was evaluated on the randomly withheld test data (Figure 4B). This parameter set was used to design the temporal variability communities in Figures 5D and 5E. The repository path of the script implementing these methods is "code/BC037/modeling/BC037B_Fit_gLV.m".

To generate results in Figure S4J, the 5-fold cross validation procedure was applied to random samplings comprising 25% of the full community dataset, which contains 201 conditions, each the average of 3 biological replicates. For example, each "fold" for the 25% subsampling contained about 40 community samples (each of which contains abundances for all 10 species), or 80% of 25% of the full dataset. Regularization coefficients were sampled over a logarithmically spaced interval. In order to utilize the full dataset, median parameter values inferred across each "fold" for 100% of the data with a regularization coefficient of 0.1 are used in Figures 4C and 4D. The repository path of the script implementing these methods is "code/BC037/modeling/regularization_validation.m".

The "supernatant growth effect" described in Figure 4B was as follows: each species was cultured on each opposing species filter-sterile supernatant as well as fresh media (90 total conditions, 4 biological replicates each, with self-supernatant culture omitted). Time series growth curves were collected as previously described (40 hours). The area under the curve was calculated using MATLAB's "trapz" function. The supernatant growth effect was calculated as the median area under the curve of species i cultured on species j supernatant minus the median area under the curve of species i on fresh media. Both the supernatant growth effect and the inter-species interaction parameters were normalized between -1 and 1 by dividing by their maximum absolute value prior to plotting in Figure 5C. The purpose of normalization was so that an equivalent "small numerical threshold" around 0 could be applied when categorizing the sign of the parameters or supernatant effects as positive, negative, or zero. This threshold (5% of the maximum absolute value of each) was applied such that small numerical values that were effectively 0 would count as such, particularly in the case of the gLV parameters driven towards zero by L1 regularization. The repository path of the script implementing these methods is "code/BC058/analysis/BC058_analysis.m".

## Design algorithm for temporal variability subset communities

The gLV model was used to design communities with low temporal variability over the course of four simulated passages. For all possible 2-to-9-member subcommunities (i.e. sum of 10 choose k for k=2 to 9), a constrained optimization problem was solved to minimize an objective function as a function of the initial conditions of the species present in the subcommunity.

$$EuclideanDistance : \left( \sum_{i}^{n} \left( X_{p,i} - X_{(p-1),i} \right)^2 \right)^{1/2} \qquad \text{(Equation 14)}$$

$X_{p,i} -$ *fractional abundance of species "i" at (simulated) stationary phase of passage "p"*

$$maximize \ \frac{\sum_{p=1}^{4} Sd_p}{\sum_{p=2}^{4} Eu_p} \qquad \text{(Equation 15)}$$

$Sd_p -$ *Shannon diversity at (simulated) stationary phase of passage p*

$Eu_p -$ *Euclidean distance between (simulated) stationary phase compositions of passages "p" and "p - 1"*

Species absence in subcommunities were simulated by forcing both upper and lower bounds of the omitted species population sizes to zero. Initial condition solutions were bounded between zero and 0.01 simulated OD600 for species present in a subcommunity. The endpoint compositions resulting from all initial condition solutions were sorted into unique results using MATLAB's "uniquetol" function (within a numerical tolerance of 0.05 for each species). Nine of these communities were chosen for experimental validation on the qualitative criteria of having all species present in the set of subcommunities. These nine communities were of size 2-4 members. As a comparison, we designed four-member high temporal variability subcommunities by maximizing the

product, rather than the ratio, of the diversity and distance terms in Equation 15. The nine low temporal variability subcommunities and three high temporal subcommunities were inoculated at densities according to the computational predictions. These inoculum conditions spanned orders of magnitude with no symmetry between conditions. The following strategy was used to inoculate these conditions: an "inoculum" 96-well 2 mL deep well block was prepared in which each species preculture material was diluted to 0.1 in row one. Tenfold serial dilutions were then performed such that preculture material was available for pipetting at a range of 0.1 to $10^{-5}$ OD600. The liquid handling robot was assigned to aspirate from whichever well would result in the smallest aspiration volume greater than 7 $\mu$L, for each species in each condition. The culture was incubated, passaged, and sampled as previously described. Statistical evaluation of temporal variability for Figure 5B was performed by calculating the coefficient of variation of the mean relative abundances (of 3 biological replicates) across the 4 passages for each species in each designed community. For example, if species "i" had mean relative abundances across passages in community "j" of (0.2, 0.1, 0.1, 0.1) then $CV_{ij}$ would be 40%. The repository path of the script implementing these methods is "code/BC040/modeling/BC040_design_temporal_variability.m"

The ensemble null model referenced in Figure 5C was generated by randomly shuffling interspecies interaction terms 100 times (i.e. non-diagonal elements of the aij parameter matrix of the gLV model). As MATLAB's "randsample" function, which was used to reindex the parameters, can yield indices consistent with the original order (e.g. randsample([1 2 3 4),4) can yield an output of [**1** 3 4 2]), only samplings in which all non-diagonal parameters had been assigned to new locations in the shuffled aij matrix were used (or 37 of the original 100). The repository path of the script implementing these methods is "code/BC040/modeling/BC040_analysis_v2.m"

## Bootstrap analysis of parameters and prediction uncertainty for the gLV model

Sampling with replacement was performed 100 times to resample the "training+validation" data on which the original gLV model was trained (same 90% of full dataset, same 10% still withheld for testing). A parameter set was inferred for each resampling (each of which contained around 120 unique samples) using the previously identified best regularization coefficient of $10^{-1.5}$. These interaction parameter distributions are shown in Figure S5A, while selected predictive distributions on the designed "temporal variability" initial conditions made using these parameter sets are shown in Figures S5B–S5E. The repository path of the script implementing these methods is "code/BC037/ Bootstrap_FitgLV_predictTempVar_v3.m".

## Derivation of the Constrained System of Logistic Equations from Mass-action Type Kinetics

We aim to predict the dynamics of community growth given an understanding of monoculture growth kinetics, and an empirical community total growth threshold. This is a null-model, given that it will operate on the null-hypothesis that inter-species interactions are competitive and equivalent in magnitude.

Consider a closed system of bacteria and resources. Let us designate two categories of resource: niche resources and global resources. Each niche resource is uniquely accessed by a single organism. Niche resources are likely to represent resources used for energy generation, for example arginine consumption by *E. lenta* or sulfate reduction by *D. piger*. Alternately, a global resource is consumed by all species in a community, at an equivalent yield. This resource designation is likely to reflect nitrogen, phosphorous, trace minerals, etc., which are used by all species to produce biomass, a "substance" whose formulation is quite consistent.

$$X_1 + \frac{1}{Y_1^{X/R}}R_{N,1} + \frac{1}{Y^{X/G}}R_G \xrightarrow{\mu_1} 2X_1$$

$$X_2 + \frac{1}{Y_2^{X/R}}R_{N,2} + \frac{1}{Y^{X/G}} R_G \xrightarrow{\mu_2} 2X_2$$

And in the *n* species, *m* resource scenario:

$$X_i + \frac{1}{Y_i^{X/R}}R_{N,i} + \frac{1}{Y^{X/G}}R_G \xrightarrow{\mu_i} 2X_i$$

$$i = 1, 2, \ldots n$$

$$X_i - species\ i$$

$$R_{N,i} - niche\ resource\ i$$

$$R_{G} - global\ resource$$

$$\mu_i - growth\ rate\ of species\ i\ on\ niche\ resource\ i$$

$$Y_{X/R,(i,j)} - growth\ yield\ of species\ i\ on\ niche\ resource\ i$$

$$Y\_\{X/G\} \;-\; \textit{growth yield of all species the global resource}$$

The species' growth rates will then be multiplicative functions of each resource availability (as in mass-action Multiplicative Monod Models):

$$\frac{dX_1}{dt} \;=\; \mu_1 R_1^N R^G X_1$$

$$\frac{dX_2}{dt} \;=\; \mu_2 R_2^N R^G X_2$$

...for *n* species and *m* resources:

$$\frac{dX_i}{dt} \;=\; \mu_i R_i^N R^G X_i$$

Niche Resources:

$$\frac{dR_i^N}{dt} = -\frac{1}{Y_i^{X/R}}\frac{dX_i}{dt}$$

Global Resource:

$$\frac{dR^G}{dt} = -\frac{1}{Y^{X/G}}\sum_{i=1}^{n}\left(\frac{dX_i}{dt}\right)$$

Mass Balance on Niche Resource 1:

$$R_1^N(t) \;=\; R_{1,0}^N \;-\; \frac{1}{Y_1^{X/R}}\left(X_1(t) \;-\; X_{1,0}\right)$$

$$X_{i,0} - \textit{species i initial condition}$$

$$R_{i,0}^N - \textit{niche resource i initialcondition}$$

Assuming $X_i(t) \gg X_{i,0}$ for simplicity (inocula are usually much lower than total overall growth):

$$R_i^N(t) \;=\; R_{i,0}^N \;-\; \frac{1}{Y_i^{X/N}}X_i(t)$$

Mass Balance on Global Resource:

$$R^G \;=\; R_0^G \;-\; \frac{1}{Y^{X/G}}\sum_{i=1}^{n}X_i$$

The yield term $Y_{X/R}$ the amount of biomass a species can produce per unit of resource $R$.
In Monoculture, according to the Monod Model:

$$Y^{X/R} \;=\; \frac{\Delta X}{\Delta R}$$

For batch culture where all resource is consumed:

$$Y^{X/R} \;=\; \frac{X_f - X_0}{R_0}$$

For a batch community where all species have a constant yield on a common (global) resource, all of which is consumed, we can say:

$$Y^{X/G} \;=\; \frac{\sum f_j \Delta X_j}{\Delta R^G} = \frac{\Delta X_{comm.}}{R_0^G}$$

Where $f_j$ is the fraction of the the resource that has been allocated to species j. The sum of all fractions is 1, and $\Delta X$ is the total biomass of any species type that can be produced from an initial quantity of resource, $R_0^G$

Substituting this expression into the mass balance for $R^G$:

$$R^G(t) = R_0^G \left( 1 - \frac{\sum\limits_{j=1}^{n} X_j(t)}{\Delta X_{comm.}} \right)$$

Substituting the above mass balances into the species' ODE:

$$\frac{dX_i}{dt} = \mu_i R_i^N(t) R^G(t) X_i(t)$$

$$= \mu_i \left( R_{i,0}^N - \frac{1}{Y_i^{X/R}} X_i(t) \right) \left( R_0^G \left( 1 - \frac{\sum\limits_{i=j}^{n} X_j(t)}{\Delta X_{comm.}} \right) \right) X_i(t)$$

$$= \mu_i R_{i,0}^N R_0^G \left( 1 - \frac{X_i(t)}{Y_i^{X/R} R_{i,0}^N} \right) \left( 1 - \frac{\sum\limits_{i=j}^{n} X_j(t)}{\Delta X_{comm.}} \right) X_i(t)$$

### Data Exclusion

The following replicates were omitted from NGS analysis due to cross-contamination of >1% of total reads and/or low total sequencing reads <10% of average: Figure 5E.ii passage 2 replicate 1 and passage 4 replicate 3, Figure S5E.vii replicate 2 passages 2-4. The following growth curve replicates were omitted from logistic analysis in Figure 1B due to lack of growth or suspected contamination, using a z-score threshold of 1.5: BH M5 r1, BH M8 r4, BL M9 r4, BU M3 r1, CA M1 r1, EL M1 r1, ER M1 r1, ER M2 r1, ER M3 r1, FP M3 r4, FP M6 r1, and PC M8 r4. In total, 12 of the 360 replicates across 10 species, 9 media, and 4 replicates were omitted, no more than one replicate was omitted per species/media condition.

