## [Document S2. Transparent peer review records for Connors et al. · Cell Systems]

Control points for design of taxonomic composition in synthetic human gut communities

Bryce M. Connors, Jaron Thompson, Sarah Ertmer, Ryan L. Clark, Brian F. Pflieger, Ophelia S. Venturelli

Summary

Initial Submission: Received August 03, 2022
Preprint: <https://doi.org/10.1101/2022.03.14.484355>

Scientific editor: Ernesto Andrianantoandro, Ph.D.

First round of review: Number of reviewers: Three
Three confidential, Zero signed
Revision invited October 20, 2022
Major changes anticipated
Revision received June 22, 2023

Second round of review: Number of reviewers: Three
Three original, Zero new
Three confidential, Zero signed
Accepted November 20, 2023

This Transparent Peer Review Record is not systematically proofread, type-set, or edited. Special characters, formatting, and equations may fail to render properly. Standard procedural text within the editor's letters has been deleted for the sake of brevity, but all official correspondence specific to the manuscript has been preserved.

Editorial decision letter with reviewers' comments, first round of review

Dear Dr. Venturelli,

I hope this email finds you well. The reviews of your manuscript are back and I've appended them below. On balance, the reviewers appreciate the goals of the work presented here; they've provided constructive comments that are aligned with our hopes for the paper. Accordingly, we're happy to invite a revision.

To help guide this revision, here are a few points that I see as being potentially make-or-break and must be addressed as we move forward:

1. Clearer articulation of the driving problem and improved framing of the study
2. Improving statistical rigor and presentation of the data.
3. Clarifying the role of the gLV model and the rationale for its connection with the rest of the framework, and ensuring the validity of its inferences.

Reviewer #2 has some particularly thorough and detailed comments and suggestions for addressing these points and thus provides a good template for revision. I'd also like to be explicit about an almost philosophical stance that we take at Cell Systems.

- We believe that understanding how approaches fail is fundamentally interesting: it provides critical insight into understanding how they work. We also believe that all approaches do fail and that it's unreasonable, even misleading, to expect otherwise. Accordingly, when papers are transparent and forthright about the limitations and crucial contingencies of their approaches, we consider that to be a great strength, not a weakness.
- We believe that the figures are the scientific backbone of the paper. Currently, it's not possible to understand the manuscript's conceptual advance from figures presented. Similarly, it's not possible to understand where your approach gets its analytical power. These things need to be demonstrated with data and analysis, in the form of clear figures with their legends or mathematical argumentation, and then supported with explanatory text.

Please keep these in mind as you address the reviewer concerns. I would also like to note, in regard to dealing with the density of the figures - please keep in mind you can have up to 7 main figures, so it may be beneficial to expand or split up existing figures and/or bring important supplemental figure to the main text. I hope you find this feedback helpful. If you have any questions or concerns, I'm always happy to talk, either over email or by Zoom. More technical information and advice about resubmission can be found below my signature. Please read it carefully, as it can save substantial time and effort later.

I look forward to seeing your revised manuscript.

All the best,

Ernesto Andrianantoandro, Ph.D.
Scientific Editor, Cell Systems

Reviewers' comments:

Reviewer #1: You will find below my review of the paper by Connors and colleagues with the title: "Control points for design of taxonomic composition in synthetic human gut communities".

I read this paper with great pleasure. The way I understand this paper, the authors propose three different methods to achieve a given end-point bacterial population, using bacteria from the gut as an example community. The proposed set point is to get even bacteria communities maximizing the Shannon diversity index.

- 1) First, the authors seek to optimize media composition in order to obtain an even growth of each bacteria in monoculture. They use monoculture data fitting and manage to increase the "monoculture-diversity".
- 2) As a second step, the aim of the authors is to use the newly designed media to find the optimal inoculating strategy to maximize the Shannon diversity in a coculture experiment. They use a dynamic population model to predict the effect of inoculation values on endpoint proportions. The model is then refined manually based on initial results to force new initial inoculation values. A last step, excluding one of the species yielded even better results.
- 3) Finally, the authors propose to use acquired experimental data to predict interspecies interactions fitted by a Lotka-Volterra model, with the aim of being able to propagate an endpoint culture to fresh media with stable results.

First of all, I believe that the proposed, interesting approaches will probably be a good seeding point for further studies. I find the data and analysis to be convincing, the conclusion well formulated and the text nice to read. Although I am probably not aware of all the literature in the field, I believe this approach to be new. For these reasons I would recommend the editor for the publication in Cell Systems.

I propose to describe my main interrogations about the paper in this section, and I will describe minor points below.

I have two main comments for this paper.

The first one is that while the main ideas and claims seem solid, some smaller claims in the paper are not sustained by a proper statistical analysis or clear-cut data. As an example (but there may be others in the paper that need to be attended to), the claim in line 144 feels counterintuitive from the data: it is stated that there is a positive interaction between amino acids (AA) and sugar for the BH bacteria, while the blue line in 1e seems thinner for AA+sugar than for AA alone. Fig S2a, as I read it, seems to even show that AA alone is better than everything else... Other statistical testing could be added to the paper, although I would leave it up to the authors since some observations seem obvious. For instance, line 163 claims a gain in the method (fig 1i). This can be seen from the figure, some may like to know how statistically relevant this is. In the same line, I didn't get what was tested was in line 283.

The second comment is on the modularity of the 3 approaches. An effort was made to combine media and inoculation optimizations, but the gLV part felt more independent to me. While looping the application of the gLV to design media composition in order to e.g favor or impair some bacteria is probably out of the scope of the study, I would have liked more convincing discussion on what could technically be made (or not) to combine the 3 methods in order to obtain a stable, Shannon-maximised, complete community. Part of my feeling may come from the introduction to the gLV part, were I got a bit lost on the goal (maybe start by mentioning 4a in the Line313-331 paragraph?). I also didn't see much discussion on the fact that most stable networks are 3-fold networks (plus some 2 and 4). How does this matter for the methods?

Minor comments and interrogations:

- Line 153-158: this is impressive results. Could you comment on what kind compositions can or cannot be targeted with this method?
- Line 160: typo: "than"?
- Line 201: CSLE is spelled out in figure, I would suggest having it also in main text (see what is the editor's policy)
- Lag time is discussed two or three time in the paper (per se, about BL and in discussion I believe). Given that lag distributions e.g. increase when cells wait in stationary phase, how would this affect the stability of cycling? It was not clear to me: did you try subculturing BL first before inoculation? You mention a lag time issue with the freezer stocks (L298): was it evolution of shorter lag time or just due to the priming with the other 9 members?
- Line 218: I first got lost while reading this and the figure, thinking you may have been using old cultures to inoculate the new in the DTL cycles. Maybe make sure other readers don't get lost as well.
- L230: why did you choose to cap the inoculation values? If there is no cap, how high would the CA, EL, PC levels need to be? Already at end values?
- L249: I don't see it in the figure? In SI? (see main point 1)
- L254: It seems FP and BH have higher inoculation in Fig3b
- L261: I do not understand the point of using other metrics here.
- L284-288: I had a hard time understanding the reasoning in this section. I think the prior I was missing is that selective plating is not possible, so the authors had to resort to proxy. I didn't see why the first dilution process was useful (L284-288): doesn't it only prove that one/some bacteria of each species is alive since inoculate size doesn't predict end point very well? The second part (L289-302) was more convincing to me, although this was the part I had a hard time understanding on first read. I think here a plain explanation of the design before presenting the results would help.
- On L289-302, one overkill possibility coming to mind would be to add a colony size to CFU observation first in monoculture to correct the pooled colony read...
- L295: if PC has the only small colonies, wouldn't an enumeration of small colonies compared to the liquid prediction be more convincing that this hand waving argument?
- L307 (to link to main comment 2): fig 4a missing from main text, fig 2d,e inexistent. I got lost.
- L320: I felt like a graphical representation of the 10 species network of interactions like in Fig 4d would have helped me
- L345: how close to stability can the full community or some of the 9-members ones be? We see it is very unstable in the DL results, but I would have liked a comparison.

I don't have any comments on the discussion. I would like to thank the authors for this interesting read! I

also would like to apologize for sending my comments 2 days later than the initial editor's deadline.

Reviewer #2: The authors have done impressive work and made progress in understanding how to design community media for a target composition. The experiments are overall well done and the results are compelling. The paper could be framed better and the inference of the gLV model faces what I see as serious issues. That said, I find the work to be of high quality and support its eventual publication in cell systems.

Major

(1) I don't find the framing of the problem as especially compelling and I don't think that the current framing has broad interest. The argument is that we want to be able to grow communities with defined composition to be able to scale-up production of probiotics. It isn't totally clear to me that this is a serious issues (yet - it might be down the line?). There are two reasons -- (1) it seems like a distant problem relative to the work proposed here which isn't explicitly looking at scaling up production. Further, in the experiments the authors have to start from monocultures, mix with well-defined initial conditions and then the designed communities grow through one cycle to a designed shannon diversity. then they tune these initial abundances. It seems to solve the applied problem that is a motivation for the paper as it stands one would want to be able to mix species together in uncontrolled relative abundances and assemble a community with some final set of abundances (otherwise, you still have to grow the monocultures of each strain!). (2) Studies with synthetic communities of gut microbiota assembled by growing mono-cultures and mixing strains show high reproducibility of community structure in vivo ([https://www.cell.com/cell/fulltext/S0092-8674\(22\)00990-4](https://www.cell.com/cell/fulltext/S0092-8674(22)00990-4)) and in vitro. Note in these experiments the initial relative abundances of each strain are not carefully controlled. So it appears that the final abundances of a probiotic in the host may not even depend that much the initial relative abundances. Given these two arguments it's harder to see the motivation for the study presented here.

I think this is a problem from a narrative perspective (drawing the reader in) and from an intellectual perspective (why do we care?). That said, I think the solution is to broaden the perspective of the introduction to ask if and how we can control the abundances of organisms in a community. The authors briefly seem to touch on this in the paragraph starting on line 48... but I think this should be significantly expanded. There is a question of how controllable the abundances are and then the question of why you might want to control them. One way to think about the work here is see it as an answer to the question -- can we control the abundances of species in a community to put them in a desired state? When and why would we expect this to be possible and how could we do it? The authors have made some progress to this end in this paper, and so I think it is a framing that would be more broadly compelling. I am open to suggestions from the authors on how to broaden the scope of the introduction as well. But I do think it is important to think carefully about this.

(2) Figure 1 is packed with information, some of it redundant and this takes away from the reader getting the gist of what the figure is trying to communicate. Can you try to simplify? Here are some specific comments on Figure 1. Do we need to see the tree for all the species you used? I like including panel (b) -- one question here: why does the PC strain grow when all nutrients are zero? It would seem that it

should not grow at all. (D) -- it is cluttering to write out the equation for shannon diversity twice in the figure. Remove this for a text label? (E) is a really important panel information wise but I cannot get anything out of the line thicknesses. Can you actually plot these coefficients? I am finding myself trying to do the linear regression by eye from panel (c). I'm not sure I need panel (i) if we already have panel (f). Same for panel (j) -- little information here, it could be included simply as a line in the text and this panel could be removed.

(3) The paper is awash with modeling formalisms and going from one to the next is confusing and doesn't always seem necessary. For example, the statistical approach to designing media is reasonable and it seems to work ok. I understand the motive, but could the authors articulate why they don't use a more explicit model like a consumer-resource formalism? For example, it seems like the data in Figure 1 could be used to specify growth rates and carrying capacities for each species on each type of carbon (but of course not pH). (Including pH in the CRM formalism is a challenge). I'm asking for an elaboration of the motivation of using a statistical approach here. It seems an important question, because a CRM would give a more explicit characterization of the negative interspecies interactions (line 195) and could significantly help in the objectives set forth in Figs 2 and 3. I am left wondering if a model that accounted even a bit more explicitly for the nutrients might not perform significantly better. This might also improve predictions, especially for strains experiencing strong competition (e.g. BL line 254). After reading the paper I feel deluged with different statistical and dynamical models. I'm left wondering if a single formalism might not be able to make all of these predictions under one rubric. I recognize that this is too much of a renovation for this manuscript, so here I ask the authors to comment on why not a CRM. I think the multiplicity of formalisms introduced weakens the work overall.

(4). The authors choose to optimize communities for shannon diversity, which peaks when all strains are at equal abundances. Any idea how well this approach would work for target compositions with large differences between strain relative abundances?

(5) I'm not convinced of the inference procedure for the gLV model or of its utility for predicting dynamics. There are several issues that I see here:

(5.0) (line 842) I'm confused about exactly what training data was used for this inference. I can see how monoculture OD time series would be used, but for the "10-member community initial and stationary phase data." Are the authors using sequencing measurements of relative abundances to parameterize this model? If so, how are they converting from relative abundances to absolute abundances (as the model is written in absolute abundances)? I am concerned that this inference will be subjected to wild errors due to difficulties mentioned in comment (1) (moderate). I'm also a little unclear on how trustworthy the L1 regularization approach is for these nonlinear ODE models. When doing regression problems L1 regularization can be very sensitive to the structure of the training data in terms of the predictors that get selected, especially if those predictors are correlated. How does this play out in this non-linear ODE case? It seems that the penalty term could result in models that are good predictors but poorly reflect the underlying ecological interactions. I imagine the authors have thought about this in the past, but a mention should be made either in the text or the SI.

(5.1) Examining Figure S10 amplifies my concerns with respect to the inference of this model. There is no

clear minimum in the MSE as a function of the hyperparameter, and the error bars from different test sets are large. Similar statement holds for ρ . Further, neither the MSE or the ρ change much over the range of hyperparameter scanned. This suggests to me that the model might not be changing much (e.g. a plot of parameter values vs. λ would have small slopes -- note that these plots typically show smooth parameter variation with the hyperparameter, is this happening here?). What is going on here? Is it that the many parameters that were initialized at zero are stuck there and the other parameters are not changing much? Perhaps these parameters already provide a reasonable test set prediction. Is it clear that the regularization is even necessary? What happens at $\lambda \rightarrow 0$? It may also have to do with the way the cross validation is being done. If there is always a sample in the training set that is similar (e.g. a replicate) of a sample in the test set then the apparent out of sample prediction will look good, but the ability of the model to predict dynamics in say a community of 5 of the 10 species may be very poor. In some cases it does appear that the model is doing a pretty bad job with true out of sample communities  (Fig. 4D -- 2nd, 3rd, 4th, 5th, columns).

(5.2) Can the authors show that their inference procedure is actually improving model predictions for the data in Figure 4 relative to some reasonable null model? For example, if you pick interaction terms by some heuristic is the optimization of these LV interaction coefficients actually improving predictions relative to an educated guess? It isn't clear to me. I think this is important because the claim being made is that the inference of this gLV model is useful for designing communities with some specific dynamics, but I don't believe that it has been shown that this is the case. For example, what if you use the CSLE model. Does that perform quantitatively worse than the gLV model?

(5.3). Is there any indication that the inferred interactions are correct? For example, if the framework infers a large interaction coefficient, can the authors validate a strong interaction in a pairculture experiment? The authors point to other studies which are qualitatively consistent with inferred interactions -- but this isn't so convincing because there is not direct evidence.

(5.4) Is there an estimation of gLV prediction errors? Is there any systematic trend in these errors? The authors call out specific examples where the model works (paragraph line 350), but I find this unsatisfying (confirmation bias). Does the model do better or worse for particular species? Is the model able to predict final OD and do these predictions work? Does the model perform better for some species than others or communities of high/low temporal variability? These questions should be engaged with statistically if at all possible.

(5.5) Euclidean distances on compositional data as defined by Equation 14 don't make sense because compositional data lie on a simplex (not a metric space). For distances in relative abundances to satisfy the constraints of a metric space it is necessary to measure distances after a CLR or ALR transform. These distances will in general not be the same as those measured by equation 14. So I think the authors need to compute these distances after such a transform or provide a different metric of temporal variability that respects the compositional nature of the data.

Moderate:

(1) Paragraph that starts line 289: given that sequencing measurements are a poor reflection of true

relative abundances (e.g. <https://elifesciences.org/articles/46923>) I find it a bit dubious to multiply the total CFUs by the relative abundance of each strain to get the fraction of viable cells. Another serious issue with this is that plating efficiency can be highly variable across species. Given these two (hard to characterize) sources of bias I wouldn't be surprised if these estimates of viability were off by an order of magnitude or more. Can the authors address these issues at least in explaining these results.

(2) It appears that DTL 3 was designed by using data from DTL 1 and 2 to train a model and infer conditions for DTL 3. This approach neglects the fact that the conditions for DTL 2 were chosen based on DTL 1. The "right" way to do this is to use a bayesian framework where posterior distributions from a previous DTL are used as priors in the next. could the authors comment on why they did not take this approach and how it might impact their results?

Minor

-- In the abstract the 91% and 53% are confusing numbers that are given without much context. Can you make more general statements here about the success of your approach?

-- For the elastic net hyperparameter selection -- can you show the MSE (for held out data) as a function of the values of the hyperparameters (a surface in this case)? It's not immediately clear to me that you aren't still overfitting. Overfitting is sometimes indicated when the CV curves do not have a clear minimum. How many datapoints and how many parameters are there? (But -- Fig S3 does look quite good so probably ok!)

-- line 152 and Figure 1 panel (f) -- make "rho" the actual greek letter ρ . realize that this is present throughout. Similarly, p-values typically use lower-case p.

-- The p-values in Figure 4b are astronomically low, but I think this reflects the fact that the authors are using a parametric test which assumes gaussian distributions. If one does a non-parametric test what are these p-values? (e.g. permutation) alternatively, does the data adhere to the assumptions of the t-test? Squinting it doesn't really look like it.

-- could the freezer stock related issues with BL have been mitigated if the authors used multiple rounds of preculturing before doing experiments? e.g. growing each strain overnight a couple of times?

Reviewer #3: The manuscript by Connors et al describes a two-stage, model-driven framework for optimizing the composition of a defined community of gut commensals. Manipulation of media composition and inoculation density were the two control points used to optimize diversity. The paper is well written and clearly constructed, and reflects a quantitative, engineering mindset that is sorely needed in the microbiome field. In particular, the idea that guiding the medium composition to promote diversity is very important, as is the focus on exploiting isolate growth behaviors to guide community engineering. My main questions focus around clarifying the motivation for several parts of the manuscript, as well as some pairwise experiments that would help to solidify some of the main conclusions.

- Line 70: when I first read this, I was surprised at the idea that inoculation densities would have a strong effect, since I was picturing experiments carried out with repeated passaging or in a chemostat. The authors discuss the differences between a single passage vs. steady-state compositions later on, but it would be helpful to set the stage earlier on.

- Speaking of which, I was interested to see what happens to the community at steady state in the optimized medium - I don't think it has to maintain diversity for it to be interesting, but I think a comparison

between the optimized and initial media would be very interesting.

- The idea of maximizing/evening carrying capacity is one that was also introduced in Tramontano et al Nat Micro 2018, where they find that mGAM is the complex medium in which carrying capacity best reproduces typical abundances in gut microbiomes. It would be very interesting to compare the growth of their community in mGAM to their optimized medium.
- Fig. 1b: I'm confused as to what is being plotted. Can you clarify in the legend and/or text?
- They find that sugars vs. amino acids etc have variable importance depending on the species. How do these findings map to a recent Cell paper by Feng et al about nutritional preferences of gut microbes?
- Their modeling is well done and is based on the central assumption from line 196 that species with higher growth rates and larger carrying capacities would outcompete lower fitness species. Nonetheless, I would like to see more comparisons to simpler predictors such as growth rate to see what one would expect in terms of various community compositions, to see to what extent their more complex modeling is critical.
- Why do they only passage for 28 h; I would have thought some species are still growing after this point?
- Line 279, I was confused about the statement that the process is robust to variations in species inocula: isn't the point that inoculum density DOES matter, and hence there is not robustness?
- Line 282: even though things are similar when you scale up, I presume that the initial densities were maintained? Is the same true if you scale all densities down by 500-fold (so there are more generations of growth)?
- Line 298: I was confused by the point about different freezer stocks of BL - can you elaborate on the differences?
- For the interactions, these should be validated in co-cultures -e.g. BL experiencing many negative interactions as described on line 329.
- Line 350: I'm a bit confused as to the motivation for focusing on high and low temporal variability. Also what is going on in the communities that exhibit low variability when predicted to have high variability (Fig. 4b)? Cases where predictions don't work may be more interesting as they presumably highlight unappreciated interactions...
- One plus that the authors should emphasize is that you're starting from a point where species abundance is even due to the optimization.

Authors' response to the reviewers' first round comments

Attached.

Editorial decision letter with reviewers' comments, second round of review

Dear Dr. Venturelli,

I'm very pleased to let you know that the reviews of your revised manuscript are back, the peer-review process is complete, and only a few minor, editorially-guided changes are needed to move forward towards publication.

In addition to the final comments from the reviewers, I've made some suggestions about your manuscript within the "Editorial Notes" section, below. Please consider my editorial suggestions carefully, ask any questions of me that you need, make all warranted changes, and then upload your final files into Editorial Manager.

I'm looking forward to going through these last steps with you. Although we ask that our editorially-guided changes be your primary focus for the moment, you may wish to consult our FAQ (final formatting checks tab) to make the final steps to publication go more smoothly. More technical information can be found below my signature, and please let me know if you have any questions.

All the best,

Ernesto Andrianantoandro, Ph.D.
Scientific Editor, Cell Systems

Editorial Notes

Transparent Peer Review:

Thank you for electing to make your manuscript's peer review process transparent. As part of our approach to Transparent Peer Review, we ask that you add the following sentence to the end of your abstract: "A record of this paper's Transparent Peer Review process is included in the Supplemental Information." Note that this **doesn't** count towards your 150 word total!

Also, if you've deposited your work on a preprint server, that's great! Please drop me a quick email with your preprint's DOI and I'll make sure it's properly credited within your Transparent Peer Review record.

Abstract:

The Abstract is missing detail on what your approach actually entails and on why other approaches are lacking. Please add a sentence or two on these. Please also specify the composition of the microbial community - "...a human gut community" is too vague. It's better to say "10-member synthetic human gut

microbial community” as you have in the Graphical Abstract. It is also not clear what “desired temporal properties” means - please elaborate. You can have up to 150 words in the abstract, so I encourage you to make use of them. I also recommend you condense or eliminate the concluding sentence (it takes up a lot of space, which can be better used to add crucial details on your approach and results).

Manuscript Text:

- We do not support supplementary Methods or Results. Please put the derivation of the CSLE model in the STAR Methods.

Also:

- Please only use the word "significantly" in the statistical sense.

Figures and Legends:

Please look over your figures keeping the following in mind:

- When color scales are used, please define them, noting units or indicating "arbitrary units," and specify whether the scale is linear or log.
- Bar graphs are not acceptable because they obscure important information about the distributions of the underlying data. Please display individual points within your graphs unless their large number obscures the graph's interpretation. In that case, box-and-whisker plots are a good alternative.
- Please ensure that every time you have used a graph, you have defined "n's" specifically and listed statistical tests within your figure legend.
- Please ensure that all figures included in your point-by-point response to the reviewers' comments are present within the final version of the paper, either within the main text or within the Supplemental Information.

STAR Methods: Note that Cell Press has recently changed the way it approaches "availability" statements for the sake of ease and clarity. Please revise the first section of your STAR Methods as follows, noting that the particular examples used might not pertain to your study. Please consult the STAR Methods guidelines for additional information.

RESOURCE AVAILABILITY

Lead Contact: Further information and requests for resources and reagents should be directed to and will be fulfilled by the Lead Contact, Jane Doe (janedoe@qwerty.com).

Materials Availability: This study did not generate new materials. *-OR-* Plasmids generated in this study have been deposited at [Addgene, name and catalog number]. *-OR-* etc.

Data and Code Availability:

- **Source data statement** (described below)
- **Code statement** (described below)
- Any additional information required to reanalyze the data reported in this paper is available from the lead contact upon request.

Data and Code Availability statements **have three parts and each part must be present. Each part should be listed as a bullet point, as indicated above.**

Instructions for section 1: Data. The statements below may be used in any number or combination, but at least one must be present. They can be edited to suit your circumstance. Please ensure that all datatypes reported in your paper are represented in section 1. For more information, please consult this list of standardized datatypes and repositories recommended by Cell Press.

- [Standardized datatype] data have been deposited at [datatype-specific repository] and are publicly available as of the date of publication. Accession numbers are listed in the key resources table.
- [Adjective] data have been deposited at [general-purpose repository] and are publicly available as of the date of publication. DOIs are listed in the key resources table.
- [De-identified human/patient standardized datatype] data have been deposited at [datatype-specific repository]. They are publicly available as of the date of publication until [date or delete “until”]. Accession numbers are listed in the key resources table.
- [De-identified human/patient standardized datatype] data have been deposited at [datatype-specific repository], and accession numbers are listed in the key resources table. They are available upon request until [date or delete “until”] if access is granted. To request access, contact [insert name of governing body and instructions for requesting access]. [Insert the following when applicable] In addition, [summary statistics describing these data/processed datasets derived from these data] have been deposited at [datatype-specific repository] and are publicly available as of the date of publication. These accession numbers are also listed in the key resources table.
- Raw [standardized datatype] data derived from human samples have been deposited at [datatype-specific repository], and accession numbers are listed in the key resources table. Local law prohibits depositing raw [standardized datatype] datasets derived from human samples outside of the country of origin. Prior to publication, the authors officially requested that the raw [adjective] datasets reported in this paper be made publicly accessible. To request access, contact [insert name of governing body and instructions for requesting access]. [Insert the following when

applicable] In addition, [summary statistics describing these data/processed datasets derived from these data] have been deposited at [datatype-specific repository] and are publicly available as of the date of publication. These accession numbers are also listed in the key resources table.

- The [adjective] data reported in this study cannot be deposited in a public repository because [reason]. To request access, contact [insert name of governing body and instructions for requesting access]. [Insert the following when applicable] In addition, [summary statistics describing these data/processed datasets derived from these data] have been deposited at [datatype-specific or general-purpose repository] and are publicly available as of the date of publication. [Accession numbers or DOIs] are listed in the key resources table.

- This paper analyzes existing, publicly available data. These accession numbers for the datasets are listed in the key resources table.

- [Adjective or all] data reported in this paper will be shared by the lead contact upon request.

Instructions for section 2: Code. The statements below may be used in any number or combination, but at least one must be present. They can be edited to suit your circumstance. ***If you are using GitHub, please follow the instructions here to archive a “version of record” of your GitHub repo at Zenodo, then report the resulting DOI in the Key Resources Table. Additionally, please note that the Cell Systems strongly recommends that you also include an explicit reference to any scripts you may have used throughout your analysis or to generate your figures within section 2.***

- All original code has been deposited at [repository] and is publicly available as of the date of publication. DOIs are listed in the key resources table.

- All original code is available in this paper’s supplemental information.

- This paper does not report original code.

Instructions for section 3. Section 3 consists of the following statement: "Any additional information required to reanalyze the data reported in this paper is available from the lead contact upon request."

Thank you!

Reviewer comments:

Reviewer #1: I have read with interest the answer to reviewers comments. I didn't have major issues with the manuscript, and do not have more now.

All questions were answered in many details, and I thank the reviewers for that. I understand that the manuscript should reflect the authors style rather than the reviewer's, although I would have liked some of the clarifications to be integrated to the paper itself, especially when some clarity comments came from several reviewers. I would think that some of the reviewer figures could integrate the supplementary data.

This being said, I congratulate the authors for the nice piece of work!

Best regards

Reviewer #2: Thanks to the authors for their careful and thorough work addressing my comments. I hope the authors feel that the process improved the paper. I am enthusiastic about the final revision and I have no further questions at this time.

Reviewer #3: I apologize to the authors for the delay in my review. They have addressed all of my questions, and are to be commended on an excellent piece of work.

Reviewer #1: You will find below my review of the paper by Connors and colleagues with the title: "Control points for design of taxonomic composition in synthetic human gut communities".

I read this paper with great pleasure. The way I understand this paper, the authors propose three different methods to achieve a given end-point bacterial population, using bacteria from the gut as an example community. The proposed set point is to get even bacteria communities maximizing the Shannon diversity index.

- 1) First, the authors seek to optimize media composition in order to obtain an even growth of each bacteria in monoculture. They use monoculture data fitting and manage to increase the "monoculture-diversity".
- 2) As a second step, the aim of the authors is to use the newly designed media to find the optimal inoculating strategy to maximize the Shannon diversity in a coculture experiment. They use a dynamic population model to predict the effect of inoculation values on endpoint proportions. The model is then refined manually based on initial results to force new initial inoculation values. A last step, excluding one of the species yielded even better results.
- 3) Finally, the authors propose to use acquired experimental data to predict interspecies interactions fitted by a Lotka-Volterra model, with the aim of being able to propagate an endpoint culture to fresh media with stable results.

First of all, I believe that the proposed, interesting approaches will probably be a good seeding point for further studies. I find the data and analysis to be convincing, the conclusion well formulated and the text nice to read. Although I am probably not aware of all the literature in the field, I believe this approach to be new. For these reasons I would recommend the editor for the publication in Cell Systems.

I propose to describe my main interrogations about the paper in this section, and I will describe minor points below.

I have two main comments for this paper.

The first one is that while the main ideas and claims seem solid, some smaller claims in the paper are not sustained by a proper statistical analysis or clear-cut data. As an example (but there may be others in the paper that need to be attended to), the claim in line 144 feels counterintuitive from the data: it is stated that there is a positive interaction between amino acids (AA) and sugar for the BH bacteria, while the blue line in 1e seems thinner for AA+sugar than for AA alone. Fig S2a, as I read it, seems to even show that AA alone is better than everything else...

Thank you for drawing attention to the incorrect description of interaction parameters in what was previously line 144; we have revised this paragraph accordingly (**Lines 152-159**). As noted, the linear term of (AA) alone is slightly stronger than the interaction parameter of (AA multiplied by sugars). What we had intended to communicate was that this interaction effect was substantial, highlighting a meaningful interdependency of these media variables on the growth of BH. The caption detailing interpretation of regression parameters in **Fig. S2** has been updated including a description of how the data was scaled, interpretation of regression parameters as sensitivity of specie's growth to media variables, and providing the example "*a large "sugar" parameter indicates that this metabolite has a strong inferred effect on the growth of this species.*"

Other statistical testing could be added to the paper, although I would leave it up to the authors since some observations seem obvious. For instance, line 163 claims a gain in the method (fig 1i). This can be seen from the figure, some may like to know how statistically relevant this is.

We thank the reviewer for this observation and have made several updates detailed in the following response. We have added a statistical test (ANOVA) to demonstrate that all improvements to Shannon diversity were significantly significant (**Fig. 1i and 3f**).

In the same line, I didn't get what was tested was in line 283.

Thank you for highlighting this ambiguity. We revised the phrasing of the previous line 283, in which the statistical test compares community compositions at 200 uL and 100 mL volumetric scales (from identical inoculation densities) (**Fig. 3g**). We clarify that the Pearson correlation is between the sets of relative abundances for communities at the two scales (now **Lines 304-307**).

Towards substantially improving our statistical rigor, we have also added a statistical comparison of the gLV model and two null models (**Fig. 5c**), uncertainty analysis of inferred gLV parameters (**Fig. S5a**), and corrected the statistical test in **Fig. 5b** (previously 4b) to a non-parametric Mann-Whitney U test (per a similarly themed comment from Reviewer #2).

The second comment is on the modularity of the 3 approaches. An effort was made to combine media and inoculation optimizations, but the gLV part felt more independent to me.

We thank the reviewer for this question and have revised the motivation behind the gLV model, which now occupies its own section entitled “**A dynamic computational model of community assembly from designed inoculation conditions**” (**Lines 334-404**). We agree that the gLV section is more independent and have framed it as such.

While looping the application of the gLV to design media composition in order to e.g favor or impair some bacteria is probably out of the scope of the study, I would have liked more convincing discussion on what could technically be made (or not) to combine the 3 methods in order to obtain a stable, Shannon-maximised, complete community.

We understand the reviewer to be asking why the gLV model was not leveraged to further improve the Shannon diversity of the full community. We thank the reviewer for the feedback and agree that the gLV model has a strong potential for design of community composition community. Using a linear regression-based design-test-learn cycle, we achieved a Shannon diversity of 91% of the maximum possible value. We therefore decided that further improvements to the Shannon diversity of the full community using the gLV model would be relatively marginal in comparison to what had already been demonstrated towards this goal using the DTL approach. Instead, we chose to use the dynamic gLV ODE model towards the proof of concept of model-guided design of community temporal dynamics over passages. In addition, the inferred inter-species interaction parameters of the gLV model provide deeper insights into microbial interactions in our community. Our discussion has also been edited to suggest that as a future direction a flexible machine learning model and design-test-learn cycle driven by Bayesian experimental design could combine core concepts from the three approaches demonstrated herein in order to design a stable, diverse community.

Part of my feeling may come from the introduction to the gLV part, were I got a bit lost on the goal (maybe start by mentioning 4a in the Line313-331 paragraph?).

Thank you for this feedback, we have amended the missing text reference to Fig. 4a and substantially added to the introduction of the gLV model in the new section “**A dynamic computational model of community assembly from designed inoculation conditions**” (**Lines 334-404**).”

I also didn't see much discussion on the fact that most stable networks are 3-fold networks (plus some 2 and 4). How does this matter for the methods?

We expect that the algorithm returned smaller communities of 2-4 members because they had fewer species competing for a limited number of ecological niches than larger communities, therefore enabling persistence of lower fitness organisms. Future work could leverage methods presented in the media optimization section combined with community passaging experiments to identify and optimize sets of resources that support a diverse community that stably coexists across many passages.

Minor comments and interrogations:

- Line 153-158: this is impressive results. Could you comment on what kind compositions can or cannot be targeted with this method?

Thank you! If we are understanding correctly, the reviewer is referencing the improvement towards an even species composition achieved by manipulating media variables, and asking whether other target compositions can be attained with the same method. We see no reason that the approach could not be adapted to favor a non-even species composition by adjusting the objective function in the optimization scripts. In this work, we maximized "predicted monoculture diversity" as a function of media components, but one could imagine minimizing the error between predicted and target monoculture carrying capacity for all species (similar to the objective function used for the inoculation optimization, Methods). This may require more accurate media regression models, which could probably be achieved with a higher resolution experimental design (e.g., 3-level full factorial). We have revised our discussion section to mention the possibility of using our framework to design any community composition of interest in the future.

More generally, we see the media optimization step as complimentary to the inoculation tuning, as it yields a medium that "favors" the desired composition, which is later achieved by fine tuning inoculation densities. What we are really doing in this case is maximizing the potential for all species to grow in the media, which we expect facilitates the ability to later tune composition as a function of inoculation density.

- Line 160: typo: "than"?

Edited, thank you.

- Line 201: CSLE is spelled out in figure, I would suggest having it also in main text (see what is the editor's policy)

Agreed, edited as suggested.

- Lag time is discussed two or three time in the paper (per se, about BL and in discussion I believe). Given that lag distributions e.g. increase when cells wait in stationary phase, how would this affect the stability of cycling? It was not clear to me: did you try subculturing BL first before inoculation? You mention a lag time issue with the freezer stocks (L298): was it evolution of shorter lag time or just due to the priming with the other 9 members?

We have removed mention of lag time as it was a hypothesis based on data not included in the manuscript and mentioned the possibility of subculturing precultures to reduce lot-to-lot variation. We will certainly try this culturing approach in the future. Thank you for the suggestion.

- Line 218: I first got lost while reading this and the figure, thinking you may have been using old cultures to inoculate the new in the DTL cycles. Maybe make sure other readers don't get lost as well.

We are a bit unclear as to the question, which seems to be asking if the DTL cycles were inoculated as passages (e.g. if DTL 2 is sub-cultured from DTL1). Line 218 "*Guided by the CSLE model, we selected a range of inoculation densities for each species to efficiently explore the design space around the best predicted inoculum (Fig. 2, Methods)*" describes how the CSLE model was used to inform target inoculation densities for the initial DTL experiment. The remainder of the paragraph mentions "designed" inoculation densities, which we think is sufficient to explain that the subsequent DTL cycle was not inoculated from a previous DTL cycle. We do not introduce passaging until later in the manuscript (previously line 287), which we hope clarifies that the DTL cycle data shown in Fig. 3 does not include passaging. To further clarify, we now mentioned that the DTL cycle involves designing "subsequent independent experiments" when introducing these experiments (**Line 231**).

- L230: why did you choose to cap the inoculation values? If there is no cap, how high would the CA, EL, PC levels need to be? Already at end values?

Exactly, we want to make sure the specie's target inoculation densities are much smaller than the specie's carrying capacities. With a measured endpoint OD600 of around 1 and total inoculation density around 0.01 we enforce total population expansion by approximately two orders of magnitude (Methods). If there were no limit to the total inoculation density, the algorithms would likely return a trivial solution of mixing precultures at a high density to the desired endpoint composition just as the reviewer has suggested, yielding cultures with very low cell viability.

- L249: I don't see it in the figure? In SI? (see main point 1)

Apologies for the confusion and thank you for the noticing, the panel reference has been corrected from Fig. 3d to Fig. 3e.

- L254: It seems FP and BH have higher inoculation in Fig3b

We understand the reviewer to be asking why FP and BH have larger inoculation densities than BL in Fig. 3b when the previous line 254 states that BL has the largest inoculation density (in DTL cycle 3). This is a bit of a subtlety but this sentence references DTL3 specifically (largest circles connected by dashed line in **Fig. 3b**). The FP and BH inoculation densities in question correspond to cycles one and two (small and medium circles). The sentence states that BL has the largest inoculation density in DTL3, which is consistent with the data shown.

- L261: I do not understand the point of using other metrics here.

Thank you for requesting additional motivation, we have amended this paragraph to provide our rationale for comparing Shannon Diversity to the Inverse Simpson Diversity Index (**Lines 286-289**). In brief, different diversity metrics weight species evenness differently and the Inverse Simpson Diversity Metric has a greater weight on evenness than presence/absence. Optimizing the evenness of a community may be important for certain applications. Therefore, we considered this index as well but our results show that the Shannon diversity and Inverse Simpson Index are linearly related within our regime and thus does not change our results (e.g. communities with high Inverse Simpson also have high Shannon diversity in this regime).

- L284-288: I had a hard time understanding the reasoning in this section. I think the prior I was missing is that selective plating is not possible, so the authors had to resort to proxy. I didn't see why the first dilution process was useful (L284-288): doesn't it only prove that one/some bacteria of each species is alive since inoculate size doesn't predict end point very well? The second part (L289-302) was more convincing to me, although this was the part I had a hard time understanding on first read. I think here a plain explanation of the design before presenting the results would help.

We included these data since species viability is important for many applications, for example microbial community therapeutics. One approach is to use selective plating for each species in the community. However, selective plating is possible for communities of fewer members with distinct metabolic niches or well-established antibiotic resistances but infeasible for our 10-member community of gut isolates. An alternative approach is to plate on media that supports the growth of as many species as possible and then pick individual colonies into 96-well plates containing liquid media, culture these colonies and then perform 16S rRNA gene sequencing on the purified DNA. The species likely display different fitness levels on agar plates than in liquid media due to differences in inter-species interactions. Therefore, we would likely need a large number of colonies to provide a statistically robust estimate of the number of CFU mL⁻¹ for each species in the 10-member community.

Due to the challenges in implementing these methods, we resorted to different proxy as the reviewer suggests. By plating the culture and pooling the colonies, we could perform 16S rRNA gene sequencing on a large number pooled colonies to determine whether colonies of all 10 species observed in the endpoint culture. We have revised this section to include a motivation for these different approximate methods to provide information into cell viability in the end point community cultures. We edited this section following these suggestions and combined the results around the three experiments concerning viability into one paragraph (**Lines 308-324**), and included the following motivation (**Lines 298-300, 309-312**):

“Biomanufacturing of microbial communities in a real-world setting requires (1)... (3) viability of organisms harvested at the endpoint. ...Selective plating of each species in the community is not feasible. Therefore, we used different approaches to provide information about the viability of species in the endpoint community cultures.”

- On L289-302, one overkill possibility coming to mind would be to add a colony size to CFU observation first in monoculture to correct the pooled colony read...

We thank the reviewer for this suggestion. This is an interesting idea, but colony size can vary across a plate due to competition for resources and thus the colony size can depend on density of colonies (intra-strain and inter-species interactions) (Chacón, Möbius and Harcombe, 2018). In addition, colonies of different species may also exhibit inter-species interactions on agar plates, which contributes to the variability in colony size (Huang *et al.*, 2023). The colonies also have different heights as well as diameters, likely necessitating the quantification of colony volume as a correction factor. Therefore, we think this potential “correction factor” may have potential biases and thus may not be a reliable metric to normalize for species-specific colony growth variation.

- L295: if PC has the only small colonies, wouldn't an enumeration of small colonies compared to the liquid prediction be more convincing than this hand waving argument?

We thank the reviewer for drawing attention to this less than convincing argument. The previous version of the manuscript speculated as to why PC was not identified by 16S sequencing in resuspended CFU plate scrapes of the full community (an experiment estimating viability of the endpoint populations in community culture). The rationale regarding the lack of detection in the CFU plate scrape despite substantial relative abundance in the sequenced liquid culture was the following: we observed PC to have very small, slow growing colonies on monoculture CFU plates and was thus less likely to be detected than species with faster growing, larger colonies. An alternative explanation is that the substantial relative abundance of PC measured in liquid culture corresponded to non-viable cells. However, we expect this to be less likely given that monocultures of PC were not in death phase at the 28-hour harvest point of the community culture (Fig. S1a). However, we agree with the reviewer that without having provided data (e.g. comparative measurements of PC colony size) the argument was hand waving. We have removed the argument, as it is not a critical to the control strategies that are the main subject of the manuscript.

- L307 (to link to main comment 2): fig 4a missing from main text, fig 2d,e inexistent. I got lost.

Thank you for bringing up these issues, they have been resolved.

- L320: I felt like a graphical representation of the 10 species network of interactions like in Fig 4d would have helped me

Thanks for the suggestion. We debated including **Fig. R1** in the manuscript but favored the heatmap (**Fig. 4c**) as the network is largely comprised of small negative interactions that we found to be too dense to interpret easily.

Figure R1. Network representation of inter-species interactions in the 10-member community based on the inferred inter-species interaction coefficient in the generalized Lotka Volterra (gLV) model. Edge scaling represents magnitude of interspecies interaction parameter value with green indicating positive and red indicating negative interactions.

- L345: how close to stability can the full community or some of the 9-members ones be? We see it is very unstable in the DL results, but I would have liked a comparison.

We are unsure of what is meant by “the DL results” (referencing *Dorea longicatena* or perhaps a typo of “DTL”?). Community compositions for passages of the full community are shown in **Fig. S4f-h** and indicate that the composition is highly variable, seeming to trend towards competitive exclusion of all other species by BL. These results suggest that stable coexistence of all species over time is challenging in the full community. We do not have a direct comparison of the extent of variability to other communities since the 10-member community passaging experiment was inoculated with different initial species proportions as opposed to a single inoculum for the lower richness communities in Figure 5.

I don't have any comments on the discussion. I would like to thank the authors for this interesting read! I also would like to apologize for sending my comments 2 days later than the initial editor's deadline.

Thank you for taking the time to provide thoughtful, detailed feedback. We hope the manuscript has been substantially improved as a result.

Reviewer #2: The authors have done impressive work and made progress in understanding how to design community media for a target composition. The experiments are overall well done and the results are compelling. The paper could be framed better and the inference of the gLV model faces what I see as serious issues. That said, I find the work to be of high quality and support its eventual publication in cell systems.

Major

(1) I don't find the framing of the problem as especially compelling and I don't think that the current framing has broad interest.

Thank you taking the time to consider how we could meaningfully improve the quality and reception of the paper. After consideration, we thoroughly agree. Framing this study as an answer to the question “*can we control the abundances of species in a community to put them in a desired state?*” is both truer to the spirit of the work and of broader interest to the academic community. Towards this end, we have substantially revised the Introduction and Discussion sections by adding text and references concerning the fundamental question of controlling communities, as well as reducing emphasis on the applied problem.

The argument is that we want to be able to grow communities with defined composition to be able to scale-up production of probiotics. It isn't totally clear to me that this is a serious issues (yet - it might be down the line?). There are two reasons -- (1) it seems like a distant problem relative to the work proposed here which isn't explicitly looking at scaling up production. Further, in the experiments the authors have to start from monocultures, mix with well-defined initial conditions and then the designed communities grow through one cycle to a designed shannon diversity. then they tune these initial abundances. It seems to solve the applied problem that is a motivation for the paper as it stands one would want to be able to mix species together in uncontrolled relative abundances and assemble a community with some final set of abundances (otherwise, you still have to grow the monocultures of each strain!).

We thank the reviewer for this justification of reframing the scope of the study. We understand the rationale as follows: the ability to grow communities with defined composition towards scalable production of probiotics is not a serious issue because it is a distant problem and to solve this problem a target community composition should be achieved from uncontrolled initial abundances (to avoid monoculture at any point in the process). While we agree that reframing the study with emphasis on the fundamental question would greatly benefit the paper, we maintain that community culture production of microbial therapeutics is a contemporary and relevant problem towards which our work makes important progress. The crux of the applied problem is improving production efficiency via community culture to meet a dosage quota, e.g., the back-of-the-envelope calculations from the Gates Foundation grant posit that we need to produce $\sim 10,000$ doses at $\sim 10^9$ bacteria per dose every week at $\sim \$0.10$ production cost per dose. In particular, our work was funded by the Bill and Melinda Gates Foundation under the “Global Grand Challenges: New Approaches for Manufacturing Gut Microbial Biotherapeutics” grant, which notes that fermenting individual commercial probiotic strains of *Lactobacillus* or *Bifidobacterium* is commonly done at scale and with low cost. However, to restore the microbial diversity of the healthy adult microbiome, it is likely that a consortium of a substantially larger number of difficult to grow commensal strains is required. Currently these difficult to grow commensal gut isolates (such as those composing our synthetic community) are grown individually and mixed to a final formulation, a process that is complex, costly, and not scalable. Therefore, a major portion of the applied problem is bioprocess design, which occurs at bench scale, similar to our work. Once designed at small scale, this process can then be scaled up to production volumes using conventional engineering approaches. In the industrial sector many microbiome companies (Seres, Vedanta, Finch, Microbiotica, Federation, and more) have defined consortia in their drug development pipelines. Manufacturing is commonly cited as a major challenge (Eisenstein, 2020). We highlight that an independent intellectual property review team at the Wisconsin Alumni Research Foundation found our approaches to have commercial potential for this application, leading to a provisional patent (Application No. 63/306,691).

Finally, an opinion piece published in *Nature Microbiology* states that manufacturing presents a significant challenge to the large number of start-up companies aiming to develop novel therapeutics based on anaerobic gut commensals and that many of the commercially successful probiotics that currently dominate the marketplace were selected, in large part, based on their technological robustness (O’Toole, Marchesi and Hill, 2017). In fact, this type of “technologically robust” probiotics (i.e., *Lactobacilli* and *Bifidobacterium*) have even been demonstrated to impair post-antibiotic microbiome recovery (Suez *et al.*, 2018). These references highlight how the applied problem currently limits the variety of effective bacterial therapeutics that are available to patients.

(2) Studies with synthetic communities of gut microbiota assembled by growing mono-cultures and mixing strains show high reproducibility of community structure in vivo ([https://www.cell.com/cell/fulltext/S0092-8674\(22\)00990-4](https://www.cell.com/cell/fulltext/S0092-8674(22)00990-4)) and in vitro. Note in these experiments the initial relative abundances of each strain are not carefully controlled. So it appears that the final abundances of a probiotic in the host may not even depend that much on the initial relative abundances. Given these two arguments it’s harder to see the motivation for the study presented here.

We thank the reviewer for the question about the contribution of initial species proportions on the assembled community in the mammalian gut. Although mouse-gut colonization of the community members in the Cheng AG *et al.* Cell 2022 paper was not highly sensitive to the initial species proportions, initial species proportions and/or total bacterial therapeutic density plays an important role in determining the outcome of the therapeutic treatment in humans. For example, total dosage was demonstrated to be a critical factor in two human clinical trials for live microbial

community therapeutics, including the first phase 3 trial for a microbial community therapeutic (Garber, 2020; Dsouza *et al.*, 2022). We propose to reconcile these findings with the observation of reproducible community assembly from uncontrolled initial relative abundances in murine models by hypothesizing a “critical threshold dosage” is required to achieve engraftment for each species, above which relative proportions may not have a large impact on community composition. This becomes even more complicated for humans or mice already colonized with a gut community. In this case, invader species may not be able to engraft into niches that are already filled by a resident community member, leading to variation in colonization ability (Maldonado-Gómez *et al.*, 2016).

Notably, Cheng AG *et al.* Cell 2022 concerns colonization in a germ-free gut, while human therapeutics must invade a resident community with substantial inter-individual and intra-individual variability. Therefore, it is difficult to compare dosage dependence of colonization in a germ-free mouse gut to that of a human gut with resident flora, as a much larger dose may be required to overcome priority effects, which are an important factor in community assembly (Sprockett, Fukami and Relman, 2018). Further, mice can be dosed by oral gavage and are typically cohoused in a group, and coprophagic. These factors together may reduce the dependence of the final community composition on specie’s inoculum densities.

I think this is a problem from a narrative perspective (drawing the reader in) and from an intellectual perspective (why do we care?). That said, I think the solution is to broaden the perspective of the introduction to ask if and how we can control the abundances of organisms in a community. The authors briefly seem to touch on this in the paragraph starting on line 48... but I think this should be significantly expanded. There is a question of how controllable the abundances are and then the question of why you might want to control them. One way to think about the work here is see it as an answer to the question -- can we control the abundances of species in a community to put them in a desired state? When and why would we expect this to be possible and how could we do it? The authors have made some progress to this end in this paper, and so I think it is a framing that would be more broadly compelling. I am open to suggestions from the authors on how to broaden the scope of the introduction as well. But I do think it is important to think carefully about this.

We thank the reviewer for this suggestion. We agree that our work is better framed as an answer to the fundamental question, “can the abundances of a synthetic community be controlled towards a desired composition” and have substantially revised our Introduction and Discussion sections to this end. We have reduced emphasis of the applied problem by consolidating the first two paragraphs of the previous version into a single paragraph. We reframing the applied problem not as being directly solved by our efforts but as an example of where these control strategies could be applied in the future (**Lines 26-43**). We have added an introductory paragraph specifically highlighting the fundamental question from which we provide this excerpt:

A key challenge towards realizing this potential is developing the capability to steer communities towards desired states (Inda et al., 2019). Microbial communities are complex and dynamic systems, shaped by nonlinear interactions and feedbacks (Coyte, Schluter and Foster, 2015; Strogatz, 2018). Control of such nonlinear dynamical systems towards desired states is a fundamental question that lies at the heart of many problems encountered in the field of engineering and is critical to harness the functional capabilities of microbiomes for a wide range of potential applications (Haddad and Chellaboina, 2008). Precise control of microbiomes requires the ability to elucidate influential control parameters and predict temporal behavior as a function of these different inputs (Eng and Borenstein, 2019).

(2) Figure 1 is packed with information, some of it redundant and this takes away from the reader getting the gist of what the figure is trying to communicate. Can you try to simplify?

We thank the reviewer for the suggestions regarding the clarity and density of the information presented in Figure 1. It is challenging to balance detail and clarity for data from many experiments with 10-member synthetic communities and we appreciate the feedback that the density of the information interfered with its communication. As detailed in responses to the specific comments below, we have made efforts to simplify what we agree were redundancies while providing rationale for changes we chose not to implement.

Here are some specific comments on Figure 1. Do we need to see the tree for all the species you used?

The phylogenetic tree in **Fig. 1a** is the first species color legend in the manuscript and serves to associate the color schemes in the largely mathematical representations of the data with the phylogenetically diverse set of fastidious gut anaerobes that these colors represent. We are of the opinion that it improves the manuscript by bringing the underlying microbial ecology to the forefront.

I like including panel (b) -- one question here: why does the PC strain grow when all nutrients are zero? It would seem that it should not grow at all.

The slight growth (mean OD600 of 0.2) of PC in medium 5 (**Fig. 1c**) likely occurs due to consumption of resources in the base medium such as acetate and tyrosine (which was not included in the amino acid mixture due to solubility, Supplementary Data File 2). **Fig. 1b** shows only the variable media components involved in the experimental design but not all the media components in the baseline media.

(D) -- it is cluttering to write out the equation for shannon diversity twice in the figure. Remove this for a text label?

We agree and thank the reviewer for this suggestion; we have removed the “predicted monoculture diversity” equation from above **Fig. 1h**.

(E) is a really important panel information wise but I cannot get anything out of the line thicknesses. Can you actually plot these coefficients? I am finding myself trying to do the linear regression by eye from panel (c).

We understand the reviewer to be saying that it is difficult to interpret the line thicknesses in **Fig. 1e** as regression parameter values. We have included bar plots of all regression model parameters as requested (edges in **Fig. 1e**) in **Fig. S2a-j**. Previously 3 examples were shown. We have also added an explanation that **Fig. 1e** is meant to give a higher-level overview of the resource preferences network, while **Fig. S2a-j** shows specific regression terms in more detail.

I'm not sure I need panel (i) if we already have panel (f).

Fig. 1f shows the predicted vs. measured “monoculture diversity”, while **Fig. 1i** shows the community compositions and community Shannon diversity. The result that maximization of monoculture diversity yields a concomitant improvement in community Shannon diversity is a central result of the section, so we think it is important to show both plots.

Same for panel (j) -- little information here, it could be included simply as a line in the text and this panel could be removed.

We appreciate the perspective and have considered it, but we find the qualitative change in pH profile shown in **Fig. 1j** to be an interesting result of our model-guided media optimization since it corresponds with an increase in the community diversity and provides deeper mechanistic insights.

(3) The paper is awash with modeling formalisms and going from one to the next is confusing and doesn't always seem necessary. For example, the statistical approach to designing media is reasonable and it seems to work ok. I understand the motive, but could the authors articulate why they don't use a more explicit model like a consumer-resource formalism? For example, it seems like the data in Figure 1 could be used to specify growth rates and carrying capacities for each species on each type of carbon (but of course not pH). (Including pH in the CRM formalism is a challenge). I'm asking for an elaboration of the motivation of using a statistical approach here. It seems an important question, because a CRM would give a more explicit characterization of the negative interspecies interactions (line 195) and could significantly help in the objectives set forth in Figs 2 and 3. I am left wondering if a model that accounted even a bit more explicitly for the nutrients might not perform significantly better. This might also improve predictions, especially for strains experiencing strong competition (e.g. BL line 254). After reading the paper I feel deluged with different statistical and dynamical models. I'm left wondering if a single formalism might not be able to make all of these predictions under one rubric. I recognize that this is too much of a renovation for this manuscript, so here I ask the authors to comment on why not a CRM. I think the multiplicity of formalisms introduced weakens the work overall.

We thank the reviewer for the suggestions. We have added a paragraph to the introduction (**Lines 44-54**) explaining why we considered but did not use a consumer-resource model, and further elaborate as follows: The D'Hoe 2017 article referenced in the new introduction highlights some of the practical difficulties with using consumer-resource type models on even small communities of fastidious anaerobes in complex media. For example, the authors discovered that an unknown, heat sensitive compound in yeast extract was growth limiting for *F. prausnitzii*. Further, they had to assume an additional unknown chemical species to represent apparent cross feeding between two species. These examples highlight how it is in practice quite difficult to identify all rate limiting resources in even small communities of fastidious anaerobes in a single complex media. By contrast, our design of experiments allowed us to quantify their input-output relationship with growth via linear regression. Consumer resource models work well in theoretical applications or in well-characterized experimental systems where rate limiting resources are known, for example minimal media with glucose as a sole carbon source or engineered *E. coli* auxotroph communities, but are difficult to implement in practice for fastidious anaerobes on complex media (D'hoel *et al.*, 2018; Liao *et al.*, 2020).

Regarding the suggestion to reduce the multiplicity of formalisms, we are working in the future to design simple methods for designing microbial communities with desired functions. In this manuscript, we approached the problem in two stages (media optimization and then inoculum density manipulation), which was successful in substantially enhancing the Shannon diversity of the community. While the two-stage method we used is not necessarily the optimal approach to solve this problem, our experimental results demonstrate that it is effective for steering communities to desired states. Changing our approach at this stage beyond the scope of the paper as the reviewer points out.

(4). The authors choose to optimize communities for shannon diversity, which peaks when all

strains are at equal abundances. Any idea how well this approach would work for target compositions with large differences between strain relative abundances?

We clarify that while the media optimization (**Fig. 1**) optimizes for monoculture diversity, the inoculation optimization (**Fig. 3**) is performed by finding the inoculation densities that minimize the error between the model predicted composition and a “target composition vector” (Methods). The target abundance profile could be edited to any desired composition beyond the even composition that we focused on in our study. Therefore, our method should be able to identify media/inoculum conditions that steer the community to a wide range of compositions. We have amended our discussion to discuss the possibility of targeting other compositions in the future.

(5) I'm not convinced of the inference procedure for the gLV model or of its utility for predicting dynamics. There are several issues that I see here:

We very much appreciate the feedback on the gLV model. We found this feedback to be insightful, intentional in scope, and of earnest goodwill towards improvement of the manuscript. In the process of addressing these issues we have developed a much better understanding of these results and have added an additional main figure (**Fig. 4**), along with to **Fig. 5b,c** (previously **Fig. 4**). The corresponding results sections are entitled “**A dynamic computational model of community assembly from designed inoculation conditions**” (Lines 334-404) and “**Model-guided design of microbial community dynamics**” (Lines 407-469), and have been written and revised, respectively, in accordance with the feedback provided in comment 5.0-5.5.

(5.0) (line 842) I'm confused about exactly what training data was used for this inference. I can see how monoculture OD time series would be used, but for the "10-member community initial and stationary phase data." Are the authors using sequencing measurements of relative abundances to parameterize this model? If so, how are they converting from relative abundances to absolute abundances (as the model is written in absolute abundances)? I am concerned that this inference will be subjected to wild errors due to difficulties mentioned in comment (1) (moderate).

A description of our community training data preprocessing was missing (and the cartoon of the previous **Fig. 4a** may have been misleading in that it implied the model was trained on relative abundance data). We have updated **Fig. 4a** and added the following clarifications to our new results section (Lines 344-352). Our model's predictive capability was very similar using either relative or estimated absolute abundance (**Fig. R2**):

Figure R2. Comparison of gLV model test predictions of (a) relative abundance or (b) estimated absolute abundance. Predicted relative abundances is calculated analogously to experimental relative abundance: each species predicted absolute abundance is divided by the sum of predicted absolute abundances of all species for that condition. The axes are very similar in scale because total OD600 for most communities is close to 1 (which is within the linear range of the plate reader).

I'm also a little unclear on how trustworthy the L1 regularization approach is for these nonlinear ODE models. When doing regression problems L1 regularization can be very sensitive to the structure of the training data in terms of the predictors that get selected, especially if those predictors are correlated. How does this play out in this non-linear ODE case? It seems that the penalty term could result in models that are good predictors but poorly reflect the underlying ecological interactions. I imagine the authors have thought about this in the past, but a mention should be made either in the text or the SI.

(5.1) Examining Figure S10 amplifies my concerns with respect to the inference of this model. There is no clear minimum in the MSE as a function of the hyperparameter, and the error bars from different test sets are large. Similar statement holds for rho. Further, neither the MSE or the rho change much over the range of hyperparameter scanned. This suggests to me that the model might not be changing much (e.g. a plot of parameter values vs. lambda would have small slopes -- note that these plots typically show smooth parameter variation with the hyperparameter, is this happening here?). What is going on here? Is it that the many parameters that were initialized at zero are stuck there and the other parameters are not changing much? Perhaps these parameters already provide a reasonable test set prediction. Is it clear that the regularization is even necessary? What happens at lambda \rightarrow 0? It may also have to do with the way the cross validation is being done. If there is always a sample in the training set that is similar (e.g. a replicate) of a sample in the test set then the apparent out of sample prediction will look good, but the ability of the model to predict dynamics in say a community of 5 of the 10 species may be very poor. In some cases it does appear that the model is doing a pretty bad job with true out of sample communities \rightarrow (Fig. 4D -- 2nd, 3rd, 4th, 5th, columns).

We thank the reviewer for the questions. We will first clarify that the regularization coefficient versus correlation/error in question was a “higher resolution” sampling of regularization coefficient values around the best estimated value of a previous, larger range/lower resolution sampling, which was not included in the manuscript. To clarify this, we now include both plots in **Fig. S4i**. Nonetheless, the observation that model performance is not changing substantially as a function

of variation in small values of the regularization coefficient is correct. A peak in the correlation coefficient vs. regularization coefficient curve (or minimum in the MSE) results from a strong bias-variance tradeoff, wherein poor out-of-fold predictions arise from either overfitting (high variance error) or underfitting/over-regularization (high bias error), while at intermediate coefficient values the model is appropriately complex and displays high predictive ability (Bishop and Nasrabadi, 2006b). Since our out-of-fold and test predictions seemed to suffer from bias but not variance error, i.e., prediction accuracy was high but not changing much as a function of regularization coefficient until it became very large, we hypothesize that our design of experiments generated sufficient information that minimized overfitting. The community dataset originally used for training (90% of the full dataset, randomly sampled) contained 180 community samples after averaging biological replicates, each of which has abundance data for ten species, meaning that the 90 interspecies-interaction parameters were being inferred from 1800 datapoints. Further, these datapoints were systematically sampled within the design space using design of experiments. From this perspective, it is more intuitive that the predictive ability of a model trained on these data would not be sensitive to variations in small values of regularization coefficients.

We can test this hypothesis by reducing the amount of training data to see how the sensitivity to the regularization coefficient changes. To this end, we performed parameter inference on a smaller training set comprised of a randomly selected 25% of the full community dataset. We observed a strong bias-variance tradeoff, i.e. clear maximum of the out of fold correlation as a function of the regularization coefficient as shown in **Fig. S4j** and described in **Lines 357-372**. This supports the notion that our training data reduced the sensitivity of the model's predictive ability to small regularization coefficients. Since regularization yields strong improvements for smaller training data sizes, we justify its usage for larger datasets as best practice, given that the tuned coefficient value provides a small improvement in the model's predictive ability. A bootstrapping analysis of parameter uncertainty is provided in **Fig. S5a** and corresponding methods section, and suggests that parameters are for the most part well constrained.

(5.2) Can the authors show that their inference procedure is actually improving model predictions for the data in Figure 4 relative to some reasonable null model? For example, if you pick interaction terms by some heuristic is the optimization of these LV interaction coefficients actually improving predictions relative to an educated guess? It isn't clear to me. I think this is important because the claim being made is that the inference of this gLV model is useful for designing communities with some specific dynamics, but I don't believe that it has been shown that this is the case. For example, what if you use the CSLE model. Does that perform quantitatively worse than the gLV model?

Thank you for the suggestion. We agree that a comparison of the gLV predictions to a reasonable null model provides important evidence that our model and parameter inference procedure are improving predictions. We have compared the gLV model to the CSLE model (as suggested) and a gLV model in which inter-species interaction terms are randomly shuffled to new locations within the a_{ij} matrix (according to the suggestion of picking interaction terms by some heuristic). These results are shown in **Fig. 5c**, with corresponding amendments to the results (**Lines 421-423**) and methods sections. In brief, although the gLV model inferred from monoculture and 10-member community data is not always highly predictive of the designed 2-4 member communities ($\rho=0.55$, $p=1e-13$), it substantially outperforms the CLSE and shuffled gLV null models, ($\rho=.27$, $p=6e-4$ and median $\rho=.15$, median $p=4e-2$). The CLSE null model assumes that all species compete (all negative interactions) and as expected performs worse than the gLV model (infers negative and positive interactions from data). However, the CLSE model performs substantially better than the shuffled inter-species interaction gLV null model ensemble. Future work would likely improve model prediction accuracy by collecting training data with communities

of similar size (number of species added to the community at the beginning of the experiment) to the design goal, as a recent study demonstrated that prediction of community function for various sizes benefitted from including training data with similar sized communities (Clark *et al.*, 2021b).

(5.3). Is there any indication that the inferred interactions are correct? For example, if the framework infers a large interaction coefficient, can the authors validate a strong interaction in a pairculture experiment? The authors point to other studies which are qualitatively consistent with inferred interactions -- but this isn't so convincing because there is not direct evidence.

We agree this is an important question and we have carried out new experiments towards addressing it, with corresponding amendment to the results (**Lines 371-388**), (**Fig. 4d**), and methods sections in the updated manuscript. We would first like to discuss the notion of parameters being “correct” or “validated” in pairwise culture. We previously demonstrated that the model is highly predictive on withheld test data, which is a widely accepted method of model validation. We see the question of whether the parameters inferred from 10-member data reflect the results of pairwise cultures as asking whether the inferred parameters “generalize” or “extrapolate” rather than if they are “validated” or “correct”. For example, it is possible that higher-order interactions can shape community assembly and are implicitly captured in the inferred pairwise inter-species interaction coefficients informed by multi-species community measurements (Ludington, 2022). Supporting this notion, a recent study showed that accurate prediction of communities containing a particular number of members required training data comprised of communities of a similar number of members (Clark *et al.*, 2021b).

We performed a new experiment to explore the effect of growing all species in monoculture on all other species spent media (sterilized supernatant) (Venturelli *et al.*, 2018; Clark *et al.*, 2021b). A “supernatant growth effect” was calculated as the difference between the area under curve (AUC) of species *i* in species *j* supernatant and species *i* on fresh media. For example, the effect would be negative if species *i* grows less on the supernatant than the fresh media, mirroring a competitive interspecies-interaction term. The supernatant growth effects are compared to the gLV inter-species interaction terms inferred from 10-member community data, designating each as positive, negative or zero (within a small threshold). Overall, 69% of the interactions had the same sign, and 95% of the negative interactions had the same sign, demonstrating that the gLV interactions often generalize to new biological contexts and are not likely purely an artifact of parameter inference. (**Fig. 4d**). As supporting evidence, we found a highly significant correlation between the inferred gLV interaction terms and the supernatant growth effect, demonstrating an informative relationship between these inter-species interaction metrics (**Fig. 4d**).

(5.4) Is there an estimation of gLV prediction errors? Is there any systematic trend in these errors? The authors call out specific examples where the model works (paragraph line 350), but I find this unsatisfying (confirmation bias). Does the model do better or worse for particular species? Is the model able to predict final OD and do these predictions work? Does the model perform better for some species than others or communities of high/low temporal variability? These questions should be engaged with statistically if at all possible.

We agree that a more comprehensive evaluation of model error is critical. We have amended the manuscript with new analyses in **Fig. 4b** and **Fig. S5a-e** with corresponding changes to the results (**Lines 373-376, 418-469**) and methods text. In brief, we first analyzed the accuracy of the gLV model on test data with respect to each species and determined that the model displayed good predictive performance (Pearson correlation of 0.74-0.99) for all species except FP (**Fig. 4b** legend). This likely explains the poor predictions of two of the designed communities referenced in comment 5.1 containing FP (now labeled **Fig. 5e**, vii and ix). We hypothesize that model accuracy suffered because FP was not included as a design variable in the inoculation

experimental designs (but rather inoculated at a constant high level in all a conditions due to its inability to grow at lower densities in monoculture).

The other two designed communities with poor predictions referenced in 5.1 (now **Fig. 5e**, vi and viii) both predict persistence of PJ while this organism did not persist over time in the experiment. We carried out a bootstrap analysis to understand if parameter and prediction uncertainty could provide insights towards poor model predictions in communities vi-ix. In brief, bootstrapping involves generating many data subsets by randomly sampling the original dataset with replacement. A gLV parameter set is then inferred from each of the 100 randomly resampled datasets. The parameter distributions reflect parameter uncertainty, and this ensemble of models is used to generate a distribution of predictions on new data that reflect prediction uncertainty. Parameter uncertainty for the two species that were poorly predicted multiple times in communities vi-ix was comparable to parameter uncertainty for well-predicted species. This suggests that these poor predictions were not a result of disproportionate parameter uncertainty in comparison to the other species.

Prediction uncertainty for FP in the poorly predicted communities (vii and ix) was high using the parameter sets inferred using the bootstrapping approach (**Fig. S5b,c**), in agreement with the poor accuracy of the model with respect to this species on 10-member data (as described in the previous paragraph). Notably, this analysis reveals high model prediction uncertainty for PJ in the poorly predicted communities vi and vii (**Fig. S5d,e**), in contrast to the model's high prediction accuracy for PJ in held-out 10-member data (**Fig. 4b**). To demonstrate that high prediction uncertainty is not simply a common feature of designed 2-4 member communities, we include an example of CA-DL (**Fig. 5e** community iv). The bootstrapping analysis revealed that model predictive uncertainty was much lower for this accurately predicted community, evidenced by the low variance of the predicted abundance distribution.

(5.5) Euclidean distances on compositional data as defined by Equation 14 don't make sense because compositional data lie on a simplex (not a metric space). For distances in relative abundances to satisfy the constraints of a metric space it is necessary to measure distances after a CLR or ALR transform. These distances will in general not be the same as those measured by equation 14. So I think the authors need to compute these distances after such a transform or provide a different metric of temporal variability that respects the compositional nature of the data.

We thank the reviewer for this insight, they are correct that Euclidean distances should not be calculated from compositional data. We used the centered log ratio (CLR) transformation as suggested. To avoid taking a logarithm of zeros in the dataset, we followed the method of adding a small value ($1e-6$) to the relative abundances before the CLR transform (**Fig. R3b**) (Leite and Kuramae, 2020). We note that using a different small value ($1e-10$) results in a substantial change in the distributions (**Fig. R3c**). Statistical significance between conditions is similar in all cases (**Fig. R3**).

Figure R3 Alternate analyses of Euclidean distances between passages. **a** Boxplot of distributions of Euclidean distances of community relative abundances between passages for low and high temporal variability communities. A Mann-Whitney U Test is applied to determine statistical significance, with p-value indicated above the boxplots. **b** Boxplot of distributions of Euclidean distances of centered log ratio transformed community relative abundances adding $1e-6$ as a small value to eliminate taking the logarithm of zero. **c** Boxplot of distributions of Euclidean distances of centered log ratio transformed community relative abundances adding $1e-10$ as a small value to eliminate taking the logarithm of zero.

We have no doubt that centered log ratio transform is a useful method, but it seems to have some caveats for our small dataset. Therefore, we have instead calculated the coefficient of variation of each specie's relative abundance across the four passages of each community as a metric of temporal variability (**Fig. 5c**, Methods). This analysis quantifies temporal variability and avoids calculating Euclidean distance from compositional data. We have also replaced the statistical test with a non-parametric Mann-Whitney U test as suggested in the minor comments.

Moderate:

(1) Paragraph that starts line 289: given that sequencing measurements are a poor reflection of true relative abundances (e.g. <https://elifesciences.org/articles/46923>) I find it a bit dubious to multiply the total CFUs by the relative abundance of each strain to get the fraction of viable cells. Another serious issue with this is that plating efficiency can be highly variable across species. Given these two (hard to characterize) sources of bias I wouldn't be surprised if these estimates of viability were off by an order of magnitude or more. Can the authors address these issues at least in explaining these results.

We have changed the name of the metric to “calculated viable profile” to remove implication that it is “absolute” and revised our text to acknowledge the potential for substantial bias owing to plating efficiency and sequencing (**Lines 319-321**). It seems to us that the potential biases present in our metric are arguably comparable to the potential biases in a paper commonly cited as the gold-standard for “quantitative microbiome profiling,” in which 16S read fractions are normalized by copy number then multiplied by flow cytometry counts (Vandeputte *et al.*, 2017). Neither adjust for extraction or sequencing bias, and neither address species biases within a single total population abundance measurement. We do not intend this point as a criticism of this paper, but simply to highlight that potential biases are present in many widely accepted microbiological

population metrics. We also first verified that liquid monocultures were not beyond stationary phase at this timepoint (to maximize cell viability) based on the monoculture growth curve data (**Fig. S1b**), and that all species plated as monocultures exhibited growth (CFU) on the plates (**Fig. 3h** light bars).

(2) It appears that DTL 3 was designed by using data from DTL 1 and 2 to train a model and infer conditions for DTL 3. This approach neglects the fact that the conditions for DTL 2 were chosen based on DTL 1. The "right" way to do this is to use a bayesian framework where posterior distributions from a previous DTL are used as priors in the next. could the authors comment on why they did not take this approach and how it might impact their results?

We thank the reviewer for the suggestion. While we highlight Bayesian experimental design as a possible future direction in the discussion, we chose to take an approach based on an established experimental design procedure called response surface methodology (Box and Wilson, 1951). The premise of response surface methodology is to use a surrogate model (typically a linear regression model with quadratic effects) to characterize the curvature of an objective (i.e. the response surface). Using the surrogate model, new experimental conditions are proposed by first determining the point that maximizes the predicted response, and then using standard design of experiment approaches (e.g. fractional factorial designs as in the case of this manuscript) to select additional experimental conditions that explore the region around the optimized point (Box and Wilson, 1951; Box, 1954). Response surface methodology was first conceptualized by G.E. Box in 1951 and predates the conception of Bayesian experimental design (Lindley, 1956). Response surface methodology continues to be widely used in both academia and industry (Campaña *et al.*, 1997; Anagnostopoulou *et al.*, 2022). We expect that response surface methodology and Bayesian experimental design would both be effective methods to design experiments with the goal of optimizing community diversity. Finally, we would like to highlight that our method was successful in substantially increasing the community diversity.

Minor

-- In the abstract the 91% and 53% are confusing numbers that are given without much context. Can you make more general statements here about the success of your approach?

Thank you for highlighting this ambiguity, we have updated the sentence with a more general statement as follows:

We develop a two-stage approach wherein media components and initial species abundances are precisely manipulated to achieve a target taxonomic composition. Using this approach, we demonstrate the ability to predict and design the diversity of a human gut community.

-- For the elastic net hyperparameter selection -- can you show the MSE (for held out data) as a function of the values of the hyperparameters (a surface in this case)? It's not immediately clear to me that you aren't still overfitting. Overfitting is sometimes indicated when the CV curves do not have a clear minimum. How many datapoints and how many parameters are there? (But -- Fig S3 does look quite good so probably ok!)

a

Figure R4 Heatmaps of average MSE as a function of hyperparameter grid search for media regression models. a Heatmaps of average mean squared error across out-of-fold predictions with leave-one-out cross validation for elastic net coefficient and lasso regularization coefficient hyperparameter tuning of media regression models via grid search. The unique lowest MSE value is indicated with a gray star in each plot.

We thank the reviewer for this comment; we understand that the reviewer is concerned that the models are overfit and is requesting surface plots of model predictive capability vs. regularization coefficient and elastic net hyperparameters to determine if MSE is changing as a function of the hyperparameter values. We understand the reviewer to be concerned that if the MSE is not changing as a function of the hyperparameter value the models may be overfit. Figure R4 provides the requested plots, with the lowest MSE value as a function of hyperparameter values indicated by a gray star. In each case, there is a unique set of hyperparameter corresponding to the lowest average MSE for out-of-fold predictions using leave-one-out cross validation. This indicates that the average MSE is changing as a function of hyper parameter value, and therefore that the parameter inference procedure can reduce overfitting by selecting appropriate hyperparameters. Elastic net regularization was implemented to reduce overfitting as our media regression models have 16 parameters fit to nine data points.

Further, as the reviewer notes, **Fig. S1k** (previously Fig. S3) shows that the model's predictive capability across all species' test data is quite good ($\rho=0.77$, $p=7e-19$). These models were used for model-guided optimization to design a new medium with high monoculture diversity (**Fig. 1h**). The mean monoculture diversity of the optimized medium was 2.166, as compared to 1.97 in the baseline medium (**Fig. 1d** medium 7 vs. **Fig. 1h** medium 10). This improvement as compared between sets of 3 biological replicates was statistically significant via a two-sample t-test (p -value $4e-07$). The maximum possible monoculture diversity of a set of 10-species is 2.30. Therefore, our models were used for an optimization process that attained 98.6% of the maximum possible value of monoculture diversity (vs. 85.6% of maximum in the initial medium). These results suggest that our models were sufficiently predictive to achieve nearly perfect optimization results, and suggesting they are unlikely to be overfit in general.

-- line 152 and Figure 1 panel (f) -- make "rho" the actual greek letter ρ . realize that this is present throughout. Similarly, p -values typically use lower-case p .

We decided to spell out "rho" because the Greek symbol looks similar to a lower case "p," though we will certainly defer to the journal when finalizing formatting.

-- The p -values in Figure 4b are astronomically low, but I think this reflects the fact that the authors are using a parametric test which assumes gaussian distributions. If one does a non-parametric test what are these p -values? (e.g. permutation) alternatively, does the data adhere to the assumptions of the t-test? Squinting it doesn't really look like it.

Thank you for the observation, we agree that the distributions are not Gaussian and that the t-test should be replaced with a non-parametric test. We have replaced the statistical test with a Mann-Whitney U test. We have also changed the quantification of temporal variability in response to major comment 5.5.

-- could the freezer stock related issues with BL have been mitigated if the authors used multiple rounds of preculturing before doing experiments? e.g. growing each strain overnight a couple of times?

We thank the reviewer for the suggestion. We are curious to know if this is common practice and have added its mention using this procedure in the future in the methods text (**Line 603**).

Reviewer #3: The manuscript by Connors et al describes a two-stage, model-driven framework for optimizing the composition of a defined community of gut commensals. Manipulation of media composition and inoculation density were the two control points used to optimize diversity. The paper is well written and clearly constructed, and reflects a quantitative, engineering mindset that is sorely needed in the microbiome field. In particular, the idea that guiding the medium composition to promote diversity is very important, as is the focus on exploiting isolate growth behaviors to guide community engineering. My main questions focus around clarifying the motivation for several parts of the manuscript, as well as some pairwise experiments that would help to solidify some of the main conclusions.

- Line 70: when I first read this, I was surprised at the idea that inoculation densities would have a strong effect, since I was picturing experiments carried out with repeated passaging or in a chemostat. The authors discuss the differences between a single passage vs. steady-state compositions later on, but it would be helpful to set the stage earlier on.

We appreciate the perspective and have clarified “batch culture” in the revised introduction (**Line 73**) to address the reviewer’s comment.

- Speaking of which, I was interested to see what happens to the community at steady state in the optimized medium - I don't think it has to maintain diversity for it to be interesting, but I think a comparison between the optimized and initial media would be very interesting.

Results for passaging the full community from various inoculation densities in DTL 1 (optimized media) are provided in **Fig. S4f-h**, and suggest that the various initial conditions all trend towards exclusion of the other nine species by BL. This strongly suggests that the community is unstable, even in the optimized medium. Given that initial medium supports substantially lower diversity (fewer coexisting species) in a single batch culture than the optimized medium (**Fig. 1i**), it is likely that a community in the initial medium would also be unstable over passages. Optimization of resources to promote long term stability is a very interesting future direction, which we mention in our discussion.

- The idea of maximizing/evening carrying capacity is one that was also introduced in Tramontano et al Nat Micro 2018, where they find that mGAM is the complex medium in which carrying capacity best reproduces typical abundances in gut microbiomes. It would be very interesting to compare the growth of their community in mGAM to their optimized medium.

This is an interesting question but not directly relevant and thus beyond the scope of our study. Our study develops and applies a framework for controlling a particular community/media system towards a desired community composition. We show a strong improvement over our baseline condition, which is a sufficient comparison to support our claims. It is likely that the mGAM media, which contains many rich, complex components (soybean meal, protease peptone, digested serum, yeast extract, beef extract, liver extract) would support a highly diverse community, given the sheer number of resource niches present. It would not be a fair comparison to compare diversity in this medium to ours, in which we control a limited number of mostly defined components (except yeast extract at 2 g/L).

- Fig. 1b: I'm confused as to what is being plotted. Can you clarify in the legend and/or text?

Thank you for highlighting this ambiguity, the caption has been clarified to indicate that **Fig. 1b** is a heatmap of a fractional factorial experimental design matrix varying key components in a common base medium.

- They find that sugars vs. amino acids etc have variable importance depending on the species. How do these findings map to a recent Cell paper by Feng et al about nutritional preferences of gut microbes?

Feng 2022, Cell Host Microbe focuses on the utilization of complex polysaccharides by *Bacteroides uniformis* and how deletion of the polysaccharide utilization loci (PULs) influence fitness, mammalian gut colonization and inter-species interactions with butyrate producing bacteria in pairwise co-culture. By contrast, our study here focuses on controllability of communities via manipulation of sugars, amino acids, pH and yeast extract. It is difficult to draw a direct comparison since our media did not include complex polysaccharides.

- Their modeling is well done and is based on the central assumption from line 196 that species with higher growth rates and larger carrying capacities would outcompete lower fitness species. Nonetheless, I would like to see more comparisons to simpler predictors such as growth rate to see what one would expect in terms of various community compositions, to see to what extent their more complex modeling is critical.

The complexity of community growth dynamics, which is well established in the literature, is difficult to explain with simple metrics such as growth rate alone (Kumar *et al.*, 2019). This holds true for our data, as the predictive capability of the model suffers substantially when only using the growth rate and carrying capacity terms of the gLV model (**Fig. R5**). This result demonstrates that our community dynamics are not simply driven simple metrics such as growth rate and carrying capacity, and supports the use of mathematical models that capture various inter-species interactions.

Figure R5. a Scatter plot of gLV model predictions of community relative abundance in which inter-species interaction terms are set to zero. Predictions are therefore based on growth rate and carrying capacity (intra-species interaction terms) alone.

- Why do they only passage for 28 h; I would have thought some species are still growing after this point?

Thank you for this question, experimental design variables such as passaging time are of course critical for collecting meaningful data. We understand the reviewer to be asking whether our community cultures had reached stationary phase prior to passaging (in other words, are we passaging too early?). We rationally selected our 28 \pm 4 hour passaging time based on monoculture growth kinetics in the optimized medium (**Fig. S1b**). This plot shows that all organisms had reached stationary phase by 28 hours. Based on intuition alone, we also would have expected a longer culture time for gut anaerobes to reach stationary phase; however, after examining the data, we expect that this relatively fast growth (relative to gut anaerobes) was a result of having optimized the medium specifically for this community. It can indeed be seen that in the media screening experiment, species often grew slower in media 1-9 than in the optimized medium 10 (**Fig. S1a**).

- Line 279, I was confused about the statement that the process is robust to variations in species inocula: isn't the point that inoculum density DOES matter, and hence there is not robustness?

We thank the reviewer for requesting additional clarification. In DTL3, the magnitude over which the inoculation densities vary is 4-fold. In DTL 1 and 2, which aimed to explore a large design space, the inoculation densities varied 10-1000-fold. We refer to this in the sentence that follows the line in question: "*This demonstrates that our process was robust to moderate variations in specie's inocula, while changing inoculum densities over orders of magnitude was a viable control point.*"

- Line 282: even though things are similar when you scale up, I presume that the initial densities were maintained? Is the same true if you scale all densities down by 500-fold (so there are more generations of growth)?

The reviewer is correct. The densities were maintained during volumetric scale up. We would expect that scaling inoculation densities by 500-fold would affect endpoint composition (in contrast to scaling volume by 500-fold and maintaining inoculation density). Like the reviewer says, there would be more generations of growth in this case. We assume that over more generations, faster growing species would overtake slower growing species, resulting in faster growing species being overrepresented in the endpoint composition. However, performing this experiment is beyond the scope of this study.

- Line 298: I was confused by the point about different freezer stocks of BL - can you elaborate on the differences?

Thank you for highlighting ambiguity in the description of the differences in growth behavior between lots of BL freezer stocks. We prepare batches of 96 single-use glycerol stocks from colony picks derived from a permanent freezer stock (Methods). There can be variation in growth kinetics from lot to lot (for example harvest time affecting viable cell density). The experiments for the entire study, except for those shown in **Fig. 3h**, were performed from "Lot 1" of BL. BL showed a strong lag phase but high growth rate in monoculture (**Fig. S1b**) and did not grow well in community experiments despite high inoculation densities (DTL1, 2 and 3). In contrast to poor growth in initial community experiments, BL entirely outcompeted all other strains in later

passages of this experiment, demonstrating that it is indeed a high-fitness organism that for some reason did not grow well in initial community experiments.

The results shown in **Fig. 3h** were performed approximately one year after all other experiments, at which point a new lot of BL single use glycerol stocks was in use (Lot 2). At the same high inoculation density that previously resulted in very low growth in community experiments (from Lot 1), BL grew well in this initial (i.e., not passaged) community experiment (Lot 2). The growth behavior of BL from Lot 2 in the initial community experiment displayed qualitatively similar, robust growth to BL from Lot 1 during the later passages of DTL1. An extensive characterization of lot-to-lot variation of freezer stocks is beyond the scope of this study. We hypothesize that the transient difference in growth performance during the initial community culture is due to history dependence of the preculture resulting from the characteristics of the freezer stock lots. We received the suggestion from reviewers 1 and 2 that sub-culturing our precultures prior to inoculation of the community culture could solve this problem. We have therefore edited the methods text (**Line 603**) to suggest that future efforts could minimize lot-to-lot effects could by sub-culturing of precultures.

- For the interactions, these should be validated in co-cultures -e.g. BL experiencing many negative interactions as described on line 329.

We agree this is an important question and we have carried out new experiments towards addressing it, with corresponding amendment to the results (**Lines 381-400**), (**Fig. 4d**), and methods sections in the updated manuscript. We would first like to discuss the notion of parameters being “correct” or “validated” in pairwise culture. We previously demonstrated that the model is highly predictive on withheld test data, which is a widely accepted method of model validation. We see the question of whether the parameters inferred from 10-member data reflect the results of pairwise cultures as asking whether the inferred parameters “generalize” or “extrapolate” rather than if they are “validated” or “correct”. For example, it is possible that higher-order interactions can shape community assembly and are implicitly captured in the inferred pairwise inter-species interaction coefficients informed by multi-species community measurements. Supporting this notion, a recent study showed that accurate prediction of communities containing a particular number of members required training data comprised of communities of a similar number of members (Clark *et al.*, 2021b).

We performed a new experiment to explore the effect of growing all species in monoculture on all other species spent media (supernatant) (Venturelli *et al.*, 2018; Clark *et al.*, 2021b). A “supernatant growth effect” was calculated as the difference between the area under curve (AUC) of species *i* in species *j* supernatant and species *i* on fresh media. For example, the effect would be negative if species *i* grows less on the supernatant than the fresh media, mirroring a competitive inter-species interaction term. The supernatant growth effects are compared to the gLV inter-species interaction terms inferred from 10-member community data, designating each as positive, negative or zero (within a small threshold). Overall, 69% of the interactions had the same sign, and 95% of the negative interactions had the same sign, demonstrating that the gLV interactions often generalize to new biological contexts and are not likely purely an artifact of parameter inference. (**Fig. 4d**). As supporting evidence, we found a highly significant correlation between the inferred gLV interaction terms and the supernatant growth effect, demonstrating an overall relationship the magnitude of these metrics (**Fig. 4d**).

With regards to the specific situation of BL receiving many negative interactions as described in the previous line 329, we observed that all supernatant effects received by BL were indeed negative in sign (**Fig. R6a**). We tested the strongest negative interaction received by BL (from FP) in pairwise coculture and found the results to be consistent with the large, negative interspecies interaction term inferred from 10-member community data (**Fig. R6b**).

Figure R6 exploration of negative interactions received by *B. longum*. **a** Supernatant effects received by BL. All supernatant effects received by BL were negative, with 8/10 agreeing in sign with gLV parameters inferred from the 10-member community. Edge color indicates species receiving interaction (BL) and marker color indicates species donating interaction. **b** Bar plot of monoculture and pairwise coculture absolute abundance of BL and FP. Bar height indicates mean of n=3 biological replicates, which are shown as overlaid circles. Monoculture logistic growth rate (Fig. S1c) is indicated above the bar plots. BL has a larger monoculture logistic growth rate, yet FP strongly outcompetes BL from an even inoculation density. This result agrees with the very large negative inter-species interaction term inferred from 10-member community data (bottom of Fig. R6a, brown circle with red edge).

- Line 350: I'm a bit confused as to the motivation for focusing on high and low temporal variability.

Thank you for the question regarding the motivation of this section. We have added the following introductory sentence (**Line 407**):

Maintaining species coexistence over time can be influential in achieving a target community function for industrial and therapeutic applications (Peng, Gilmore and O'Malley, 2016).

The first three results sections address designing the endpoint composition of a synthetic community. The long-term dynamics are not characterized (i.e., what happens if we passage the community several times?) It would be desirable to design compositions that are maintained over passages. With the “low temporal variability” communities, we aim to design communities with an approximately constant species composition over several passaging timepoints. Communities with this characteristic would be useful in many industrial applications, for example increasing production efficiency of microbial community therapeutics or maintaining a constant function for valuable chemical production. The “high” temporal variability communities are included to demonstrate the model can differentiate between low and high temporal variability.

Also what is going on in the communities that exhibit low variability when predicted to have high variability (Fig. 4b)? Cases where predictions don't work may be more interesting as they presumably highlight unappreciated interactions...

We thank the reviewer for highlighting that in several cases the gLV model was not successful in designing the desired temporal dynamics. We agree that a more comprehensive evaluation is important towards understanding cases where the model performed poorly for designing low and high temporal variability communities. We have amended the manuscript with new analyses in **Fig. 4b (new)** and **Fig. S5a-e** with corresponding changes to the results (**Lines 364-368, 408-451**) and methods text. In brief, we first analyzed the accuracy of the gLV model on test data with respect to each species and determined that the model displayed good predictive performance

(Pearson correlation of 0.74-0.99) for all species except FP (**Fig. 4b** legend). This likely explains the poor predictions of two of the designed communities referenced in comment 5.1 containing FP (now labeled **Fig. 5e**, vii and ix). We hypothesize that model accuracy suffered because FP was not included as a design variable in the inoculation experimental designs (but rather inoculated at a constant high level in all a conditions due to its inability to grow at lower densities in monoculture).

The other two designed communities with poor predictions referenced in 5.1 (now **Fig. 5e**, vi and viii) both predict persistence of PJ while this organism did not persist in the experiment. We carried out a bootstrap analysis to understand if parameter and prediction uncertainty could provide insights towards poor model predictions in communities vi-ix. In brief, bootstrapping involves generating many data subsets by randomly sampling the original dataset with replacement. A gLV parameter set is then inferred from each of the 100 randomly resampled datasets. The parameter distributions reflect parameter uncertainty, and this ensemble of models is used to generate a distribution of predictions on new data that reflect prediction uncertainty. Parameter uncertainty for the two species that were poorly predicted multiple times in communities vi-ix was comparable to parameter uncertainty for well-predicted species. This suggests that these poor predictions were not a result of disproportionate parameter uncertainty in comparison to the other species.

Prediction uncertainty for FP in the poorly predicted communities (vii and ix) was high using the inferred parameter sets based on the bootstrapping approach (**Fig. S5b,c**). This is in agreement with the poor accuracy of the model with respect to this species on 10-member data (as described in the previous paragraph). Notably, our uncertainty analysis reveals high model prediction uncertainty for PJ in the poorly predicted communities vi and vii (**Fig. S5d,e**) in contrast to the model's high accuracy for PJ in 10-member data (**Fig. 4b**). To demonstrate that high prediction uncertainty is not simply a common feature of designed 2-4 member communities, we include an example of CA-DL (**Fig. 5e** community iv). Bootstrapping revealed that model predictive uncertainty was much lower for this well-predicted community, evidenced by a single mode and low variance distribution.

- One plus that the authors should emphasize is that you're starting from a point where species abundance is even due to the optimization.

We apologize but we do not fully understand the comment. Given that the language does not imply a major deficiency in the manuscript, we hope it will be acceptable to refrain from addressing.

References

Anagnostopoulou, C. *et al.* (2022) 'Valorization of household food wastes to lactic acid production: A response surface methodology approach to optimize fermentation process', *Chemosphere*, 296, p. 133871. Available at:

<https://doi.org/https://doi.org/10.1016/j.chemosphere.2022.133871>.

Bishop, C.M. and Nasrabadi, N.M. (2006) *Pattern recognition and machine learning*. Springer.

Box, G.E.P. (1954) 'The Exploration and Exploitation of Response Surfaces: Some General Considerations and Examples', *Biometrics*, 10(1), pp. 16–60. Available at:

<https://doi.org/10.2307/3001663>.

Box, G.E.P. and Wilson, K.B. (1951) 'On the Experimental Attainment of Optimum Conditions', *Journal of the Royal Statistical Society. Series B (Methodological)*, 13(1), pp. 1–45. Available at: <http://www.jstor.org/stable/2983966>.

Campaña, A.M.G. *et al.* (1997) 'Sequential response surface methodology for multioptimization in analytical chemistry with three-variable Doehlert designs', *Analytica Chimica Acta*, 348(1), pp. 237–246. Available at: [https://doi.org/https://doi.org/10.1016/S0003-2670\(97\)00155-4](https://doi.org/https://doi.org/10.1016/S0003-2670(97)00155-4).

Chacón, J.M., Möbius, W. and Harcombe, W.R. (2018) 'The spatial and metabolic basis of colony size variation', *The ISME Journal*, 12(3), pp. 669–680. Available at: <https://doi.org/10.1038/s41396-017-0038-0>.

Clark, R.L. *et al.* (2021a) 'Design of synthetic human gut microbiome assembly and butyrate production.', *Nature communications*, 12(1), p. 3254. Available at: <https://doi.org/10.1038/s41467-021-22938-y>.

Clark, R.L. *et al.* (2021b) 'Design of synthetic human gut microbiome assembly and butyrate production.', *Nature communications*, 12(1), p. 3254. Available at: <https://doi.org/10.1038/s41467-021-22938-y>.

Coyte, K.Z., Schluter, J. and Foster, K.R. (2015) 'The ecology of the microbiome: Networks, competition, and stability.', *Science (New York, N.Y.)*, 350(6261), pp. 663–666. Available at: <https://doi.org/10.1126/science.aad2602>.

D'hoë, K. *et al.* (2018) 'Integrated culturing, modeling and transcriptomics uncovers complex interactions and emergent behavior in a three-species synthetic gut community.', *eLife*, 7. Available at: <https://doi.org/10.7554/eLife.37090>.

Dsouza, M. *et al.* (2022) 'Colonization of the live biotherapeutic product VE303 and modulation of the microbiota and metabolites in healthy volunteers', *Cell Host & Microbe*, 30(4), pp. 583-598.e8. Available at: <https://doi.org/https://doi.org/10.1016/j.chom.2022.03.016>.

Eisenstein, M. (2020) 'Early investments powering the ascent of microbiome therapeutics'. Available at: <https://doi.org/https://doi.org/10.1038/d43747-020-01178-x>.

Eng, A. and Borenstein, E. (2019) 'Microbial community design: methods, applications, and opportunities.', *Current opinion in biotechnology*, 58, pp. 117–128. Available at: <https://doi.org/10.1016/j.copbio.2019.03.002>.

Garber, K. (2020) 'First microbiome-based drug clears phase III, in clinical trial turnaround.', *Nature reviews. Drug discovery*, pp. 655–656. Available at: <https://doi.org/10.1038/d41573-020-00163-4>.

Haddad, W.M. and Chellaboina, V. (2008) *Nonlinear Dynamical Systems and Control*. Princeton University Press. Available at: <https://doi.org/10.2307/j.ctvc4hws>.

Huang, Y. *et al.* (2023) 'High-throughput microbial culturomics using automation and machine learning', *Nature Biotechnology* [Preprint]. Available at: <https://doi.org/10.1038/s41587-023-01674-2>.

Inda, M.E. *et al.* (2019) 'Emerging Frontiers in Microbiome Engineering', *Trends in Immunology*, 40(10), pp. 952–973. Available at: <https://doi.org/https://doi.org/10.1016/j.it.2019.08.007>.

Kumar, M. *et al.* (2019) 'Modelling approaches for studying the microbiome', *Nature Microbiology*, 4(8), pp. 1253–1267. Available at: <https://doi.org/10.1038/s41564-019-0491-9>.

Leite, M.F.A. and Kuramae, E.E. (2020) 'You must choose, but choose wisely: Model-based approaches for microbial community analysis', *Soil Biology and Biochemistry*, 151, p. 108042. Available at: <https://doi.org/https://doi.org/10.1016/j.soilbio.2020.108042>.

Liao, C. *et al.* (2020) 'Modeling microbial cross-feeding at intermediate scale portrays community dynamics and species coexistence.', *PLoS computational biology*, 16(8), pp. e1008135–e1008135. Available at: <https://doi.org/10.1371/journal.pcbi.1008135>.

Lindley, D. V (1956) 'On a Measure of the Information Provided by an Experiment', *The Annals of Mathematical Statistics*, 27(4), pp. 986–1005. Available at: <http://www.jstor.org/stable/2237191>.

Ludington, W.B. (2022) 'Higher-order microbiome interactions and how to find them', *Trends in Microbiology*, 30(7), pp. 618–621. Available at: <https://doi.org/https://doi.org/10.1016/j.tim.2022.03.011>.

Maldonado-Gómez, M.X. *et al.* (2016) 'Stable Engraftment of *Bifidobacterium longum* AH1206 in the Human Gut Depends on Individualized Features of the Resident Microbiome', *Cell Host & Microbe*, 20(4), pp. 515–526. Available at: <https://doi.org/https://doi.org/10.1016/j.chom.2016.09.001>.

O'Toole, P.W., Marchesi, J.R. and Hill, C. (2017) 'Next-generation probiotics: the spectrum from probiotics to live biotherapeutics', *Nature Microbiology*, 2(5), p. 17057. Available at: <https://doi.org/10.1038/nmicrobiol.2017.57>.

Peng, X. "Nick", Gilmore, S.P. and O'Malley, M.A. (2016) 'Microbial communities for bioprocessing: lessons learned from nature', *Current Opinion in Chemical Engineering*, 14, pp. 103–109. Available at: <https://doi.org/https://doi.org/10.1016/j.coche.2016.09.003>.

Sprockett, D., Fukami, T. and Relman, D.A. (2018) 'Role of priority effects in the early-life assembly of the gut microbiota', *Nature Reviews Gastroenterology & Hepatology*, 15(4), pp. 197–205. Available at: <https://doi.org/10.1038/nrgastro.2017.173>.

Strogatz, S.H. (2018) *Nonlinear dynamics and chaos: with applications to physics, biology, chemistry, and engineering*. CRC press.

Suez, J. *et al.* (2018) 'Post-Antibiotic Gut Mucosal Microbiome Reconstitution Is Impaired by Probiotics and Improved by Autologous FMT.', *Cell*, 174(6), pp. 1406-1423.e16. Available at: <https://doi.org/10.1016/j.cell.2018.08.047>.

Vandeputte, D. *et al.* (2017) 'Quantitative microbiome profiling links gut community variation to microbial load', *Nature*, 551(7681), pp. 507–511. Available at: <https://doi.org/10.1038/nature24460>.

Venturelli, O.S. *et al.* (2018) 'Deciphering microbial interactions in synthetic human gut microbiome communities.', *Molecular systems biology*, 14(6), pp. e8157–e8157. Available at: <https://doi.org/10.15252/msb.20178157>.